# Privately Fine-Tuned LLMs Preserve Temporal Dynamics in Tabular Data

Lucas Rosenblatt [1]   Peihan Liu [2]   Ryan McKenna [3]   Natalia Ponomareva [3]

## Abstract

Research on differentially private synthetic tabular data has largely focused on independent and identically distributed rows where each record corresponds to a unique individual. This perspective neglects the temporal complexity in longitudinal datasets, such as electronic health records, where a user contributes an entire (sub) table of sequential events. While practitioners might attempt to model such data by flattening user histories into high-dimensional vectors for use with standard marginal-based mechanisms, we demonstrate that this strategy is insufficient. Flattening fails to preserve temporal coherence even when it maintains valid marginal distributions. We introduce PATH, a novel generative framework that treats the full table as the unit of synthesis and leverages the autoregressive capabilities of privately fine-tuned large language models. Extensive evaluations show that PATH effectively captures long-range dependencies that traditional methods miss. Empirically, our method reduces the distributional distance to real trajectories by over 60% and reduces state transition errors by nearly 50% compared to leading marginal mechanisms while achieving similar marginal fidelity.

## 1. Introduction

Structured, tabular data is a fundamentally human method of organizing information. It is at least as old as c. 2500 BCE, when Mesopotamian scribes detailed agricultural management and ration lists in tables split by horizontal and vertical lines (Robson, 2004). In the contemporary era, the ability to model such data has become a cornerstone of machine learning, with Transformer-based large language models (LLMs) demonstrating impressive fidelity in learning tabular distributions (Borisov et al., 2022; Fang et al., 2024).

* Work done at Google as part of student researcher engagement. [1]Williams College [2]Columbia University [3]Google Research. Correspondence to: Lucas Rosenblatt <lr21@williams.edu>.

*Proceedings of the $43^{rd}$ International Conference on Machine Learning*, Seoul, South Korea. PMLR 306, 2026. Copyright 2026 by the author(s).

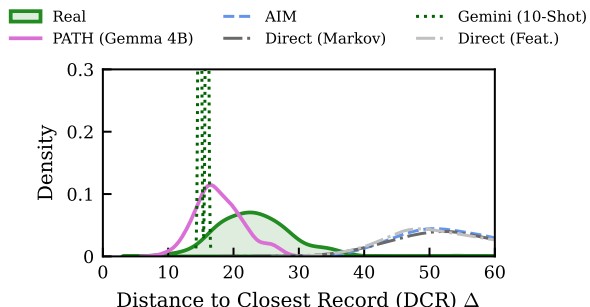

*Figure 1.* **Table-wise Distance to Closest Record (TDCR) Distribution Analysis** ($\varepsilon = 2.0$)**.** We visualize the distribution of distances from generated tables to their nearest neighbor in the real training set for MIMIC-IV Vitalsigns data using our Dynamic Time Warping-based metric. The green shaded area represents the "Real (Held-out)" baseline, showing the distances of real test tables to the training set; a high-fidelity synthetic method should produce a distribution that closely overlaps this baseline. Our proposed method, PATH (Gemma 4B), demonstrates superior overlap with the real data manifold compared to flattened marginal baselines (AIM, Direct) and non-private few-shot prompting (Gemini 2.5 Flash-Lite), which either mode-collapse (tight distribution on the left) or fail to capture the data support (too far right).

However, modern tabular data is frequently sensitive, containing personal medical histories, financial transactions, or private behavioral traces. To mitigate privacy risks, differential privacy (DP) (Dwork et al., 2006) has emerged as the gold standard for data release. A common approach to data release with DP is through the generation of *differentially private synthetic data* (DPSD) that is statistically similar to the original data. High quality DPSD preserves the statistical utility of the original private data while providing formal privacy guarantees. While there are powerful tabular DPSD methods based on marginal measurements, e.g. AIM (McKenna et al., 2024) or GEM (Liu et al., 2021b), these approaches predominantly rely on the assumption that the data consists of independent, identically distributed (i.i.d.) rows, where a single individual corresponds to a single row, and a full table represents a collection of many individuals.

This assumption breaks down in the presence of *longitudinal* or *temporal* tabular data. Consider electronic health records (EHR), such as the MIMIC dataset (Johnson et al., 2020), where a patient's health data is not a single static row, but a number of rows representing a trajectory of vital signs, lab results, and hospital events recorded over time. In this (and

other) longitudinal settings, the unit of ownership is not a row, but an entire *table* (or sequence of events); i.e. each user owns a full table of data with a temporal component present between the rows in multiple columns, and the full dataset is a collection of such tables.

Were we to apply traditional marginal-based DPSD methods to this data, one could simply concatenate all user tables into a single table, using group privacy and assuming maximum number of rows per user or sampling or aggregating each user's rows into a single row per user. This is, however, clearly undesirable if one wants to model temporal patterns in the data. Additionally, group-level privacy is too strong of a notion for this problem because it protects the privacy of every group of rows, even when the rows belong to different users. To preserve temporal trends, one would thus be forced to "flatten" all of the user trajectories into single, high-dimensional vectors that could be combined into one table. In Section 4 we define this flattening transformation formally; flattening can be catastrophic for utility, as we demonstrate later, in that it explodes the dimensionality of the domain and introduces artificial sparsity, making it difficult for marginal-based mechanisms to capture the complex, sequential dependencies inherent to the data.

To address this, we propose Private Autoregressive Trajectory Histories (PATH). Instead of synthesizing one table as a collection of rows, we seek to synthesize a full set of tables, each of which has the temporal component preserved. Here, the dataset is not a matrix of values, but is a collection of sub-matrices $\mathbf{D} = (D^{(1)}, \ldots, D^{(n)})$, where each $D^{(i)}$ represents the full history of user $i$. While we assume throughout this work that all $D^{(i)}$ share a common schema, we note that our text-based framework could naturally accommodate a heterogeneous schema, where columns may vary between users. We leave a complete exploration of this alternative application to future work. We leverage the autoregressive capabilities of pre-trained LLMs (specifically, the Gemma 3 (Team et al., 2024) family of models) to learn the distribution of these user-level tables directly. By fine-tuning LLMs with differential privacy (DP-SGD) (Abadi et al., 2016), we can generate high-fidelity synthetic collections of tables that preserve intricate temporal dynamics, such as a varied pattern of heart rates across patients.

**Our contributions.** We formalize the synthesis of user-level tabular data and contribute the following:

- **Privacy unit as full table.** To the best of our knowledge, we are the first to consider a full table as the privacy unit for differentially private data synthesis and the first to identify a practical use case for this framing, specifically for preserving temporal trajectories where a user owns the entire table. This possibility was previously hinted at by (Ponomareva et al., 2025) but lacked the application to temporal dynamics.

- **A novel framework for DP tabular data synthesis that preserves temporal trends.** Drawing inspiration from GReaT (Borisov et al., 2022), we introduce a methodology for fine-tuning LLMs to generate structured, multi-row tabular data. We propose a novel two-stage generation process that begins with an autoregressive row-generation technique conditioning on previous context to preserve temporal dependencies (e.g., state transitions in vital signs) followed by a post-processing private selection step.

- **Novel metrics for table families.** Evaluating generative quality for families of tables requires more than standard row-wise metrics. We introduce *Table-wise Distance to Closest Record (TDCR)*, a metric adapted from Borisov et al. (2022) to assess the *distributional* fidelity of generated user trajectories against real held-out data using Dynamic Time Warping (DTW). We additionally explore and report many other domain-specific time-series metrics (Section 3).

- **Extensive empirical evaluations.** We conduct extensive evaluations on a synthetic HMM dataset to control variable dependencies as well as on real-world MIMIC-IV vital signs (Johnson et al., 2020) and NYC 311 service requests (NYC Dept. of Info Tech, 2026). We demonstrate that traditional marginal-based methods scale poorly to the high dimensionality of flattened longitudinal data and fail to preserve temporal trends. Empirically, our method reduces the TDCR by over 60% compared to a state-of-the-art marginal baseline (AIM (McKenna et al., 2024)), effectively capturing temporal dynamics that previous approaches miss.

**Conflict of Interest Disclosure.** The authors RM and NP are employed by Google Research, which is involved in the development of the Gemma family of models, which was among the models evaluated in this paper. Gemma is an open-source model.

## 2. Problem Formulation

Let $\mathcal{U} = \{u_1, \ldots, u_n\}$ be a set of $n$ users. Each user $u_i \in \mathcal{U}$ is associated with a single table $D^{(i)}$ (e.g. trajectory of timesteps, recorded as rows), with $d$ columns $\mathcal{A} = \{A_1, \ldots, A_d\}$. Each attribute $A_j$ is associated with a domain $\mathcal{V}_j$. Thus, a user $u_i$'s table $D^{(i)}$ is an ordered sequence of $T_i$ records (rows), or,

$$D^{(i)} = (\mathbf{x}_1^{(i)}, \mathbf{x}_2^{(i)}, \ldots, \mathbf{x}_{T_i}^{(i)}), \qquad (1)$$

where each record (row) $\mathbf{x}_t^{(i)} = (v_{t,1}^{(i)}, \ldots, v_{t,d}^{(i)})$ is a vector in the product space $\mathcal{X} = \mathcal{V}_1 \times \cdots \times \mathcal{V}_d$. Crucially, the sequence length $T_i$ is a random variable, and the sequence represents a temporal trajectory where the value of row $\mathbf{x}_t^{(i)}$ depends on the history of previous rows $\mathbf{x}_{<t}^{(i)} = (\mathbf{x}_1^{(i)}, \ldots, \mathbf{x}_{t-1}^{(i)})$.

**User-Level Differential Privacy.** We explicitly define the privacy unit as the full user table $D^{(i)}$, representing the complete history of a single user. Based on this unit, we adopt the add/remove definition of adjacency to ensure the entirety of an individual's trajectory is protected. Formally, two collections of user tables $\mathbf{D}, \mathbf{D}'$ are **neighboring** ($\mathbf{D} \simeq \mathbf{D}'$) if $\mathbf{D}'$ can be obtained by adding or removing exactly one user $u_i$ and their associated table $D^{(i)}$.

A mechanism $\mathcal{M}$ satisfies $(\varepsilon, \delta)$-DP if for all neighboring $\mathbf{D}, \mathbf{D}'$ and all measurable sets $S \subseteq \text{Range}(\mathcal{M})$ it holds that,

$$\mathbb{P}[\mathcal{M}(\mathbf{D}) \in S] \leq e^{\varepsilon}\mathbb{P}[\mathcal{M}(\mathbf{D}') \in S] + \delta. \quad (2)$$

**Generative Goal.** Our objective is to learn a model $f_\theta$ that approximates the density of the collection $\mathbf{D}$. During synthesis, the model must generate a synthetic collection $\mathbf{D}^* = \{D^{(1)*}, \ldots, D^{(n)*}\}$ that matches the source distribution, ensuring both the joint distribution of columns (intra-row correlations) and the temporal dependencies (inter-row correlations) match the source distribution.

## 3. Metrics

Evaluating the quality of synthetic longitudinal data requires assessing how realistic individual rows look (**marginal fidelity**), whether the *trajectories* from $D^{(i)}$ maintain coherent temporal structures (**temporal integrity**), and if the collection $\mathbf{D}^*$ covers the support of the real distribution without memorizing it (**distributional variety**).

To evaluate marginal fidelity, we examine the global distributions of individual columns by comparing their densities. To ensure temporal integrity, we measure the preservation of sequential dynamics through state transition matrices and Hidden Markov Model (HMM) likelihoods. Finally, to assess distributional variety and manifold coverage, we utilize embedding-based metrics such as MAUVE and our proposed Table-wise Distance to Closest Record (TDCR). See detailed formal definitions in Appendix Section D.

**Distributional Divergence (MAUVE).** To measure the gap between the manifolds of the real collection $\mathbf{D}$ and the synthetic collection $\mathbf{D}^*$, we utilize MAUVE (Pillutla et al., 2021), which measures the information divergence between continuously embedded distributions (a score of 1.0 indicates perfect overlap). We map each variable-length table $D^{(i)}$ to a fixed-length vector $\mathbf{e}_i \in \mathbb{R}^k$ using Gecko embeddings (Lee et al., 2024), concatenating schema and mean row embeddings. Averaging row embeddings discards temporal sequencing information; thus, we utilize MAUVE to evaluate the overall distributional similarity of the tabular content rather than temporal fidelity (Section D.3).

**Table-wise Distance to Closest Record (TDCR).** Standard Euclidean distance is ill-defined for comparing tables of differing lengths. To evaluate fidelity, we must assess the *variety* of generated tables, ensuring the model covers the support of the real data rather than duplicating tables from a specific region. We introduce *Table-wise Distance to Closest Record* (TDCR). This metric extends the standard *Distance to Closest Record* (DCR) score (Borisov et al., 2022) by utilizing Dynamic Time Warping (DTW) (Kruskal, 1983) to compute a robust distance $\Delta(D^{(a)}, D^{(b)})$ between two user tables $D^{(a)}$ and $D^{(b)}$. To compute DCR between two tables, we sum DTW distances of their corresponding attributes $A_j$ and applying appropriate normalizations. This is crucial for longitudinal data, as it implicitly penalizes synthetic tables that fail to replicate the temporal shape and evolution of real trajectories, even if their aggregate marginals are correct. See Appendix Section D.2 for complete definition and Figure 7 for visual intuition.

Then, to evaluate the synthetic against real collections of tables, we compute the distance from every synthetic table $D^{(i)*} \in \mathbf{D}^*$ to its nearest neighbor in the real training set. We perform the same operation for a held-out real test set to establish a baseline distribution of "real-to-real" distances. Finally, we quantify the similarity between the synthetic and test distance distributions using the Jensen-Shannon Distance (JSD). We privilege JSD over transport-based metrics like Wasserstein for this specific comparison because it robustly compares the shape of the distributions, effectively penalizing mode collapse where synthetic data might cluster in a high-density region without capturing the full variance of the manifold; see Section D.2 for more details.

**Temporal Integrity (State Transitions, HMM).** To assess the preservation of Markovian dynamics, we evaluate the *state* transition likelihoods, where state here is a quantile of the distribution. This metric is computed per feature, measuring the univariate conditional probability of a feature value at row $j + 1$ given its value at row $j$, and does not capture cross-feature dependencies (e.g., how heart rate at $t$ might affect blood pressure at $t + 1$). We implement this by discretizing numerical features of all synthetic and real tables into quantiles to form transition matrices that estimate the probability of transitioning from one quantile to another. We distill the result into a single scalar by computing the Frobenius norm of the difference between the real and synthetic transition matrices for each feature, and report the average across all features (Section D.5).

For the case of "real" data that is completely synthetic, (where the ground truth data generation process is known (Section 6), we can evaluate long-range coherence by applying the true Gaussian HMM parameters to the private generated data (Jurafsky & Martin). We compare log-likelihoods of held-out real trajectories to DP synthetic trajectories.

**Classifier Indistinguishability.** We posit that a marker of

high-quality synthetic tables is that they are indistinguishable from real tables. Following the "classifier two-sample test" paradigm (Lopez-Paz & Oquab, 2016), we train classifiers (Logistic Regression, Random Forest, XGBoost) to discriminate between real and synthetic table embeddings (labeled binary, 0/1, and where worse predictive performance indicates better synthetic data). We utilize the same mean-pooled Gecko embeddings described in the MAUVE section, which capture the global semantic content of the table. Details on the classifier metric are in Section D.4.

**Categorical/Geospatial Metrics.** The NYC 311 dataset presents unique spatiotemporal and categorical challenges that require specialized evaluation beyond the previously discussed metric. For example, we assess temporal fidelity by extracting the "Hour of Day" from timestamps and computing the Wasserstein-1 distance between the real and synthetic diurnal distributions to ensure the daily rhythm of requests is preserved. To evaluate the geospatial features, we compare the joint distributions of latitude and longitude via hexbin density maps, allowing us to visually verify that the synthetic data respects the physical topology of the target distribution. Finally, to capture the semantic progression of user issues, we model complaint types as a Markov process. We construct a state transition matrix for the top-$k$ most frequent complaint categories, estimating the conditional probability $P(\text{type}_t \mid \text{type}_{t-1})$, and report the Frobenius norm of the difference between the real and synthetic matrices.

## 4. "Flattening" Longitudinal Data

To adapt traditional DP tabular synthesizers while attempting to retain the user-level trajectory structure, we apply a *flattening transformation* $\Phi$ to map a table $D^{(i)}$ of variable length $T_i$ into a single fixed-width vector. Unfortunately, the distribution of trajectory lengths $T_i$ could vary across users, forcing the use of extensive padding.

**Definition 4.1** (Flattening Transformation). Let $\mathcal{V}'_j = \mathcal{V}_j \cup \{\texttt{NULL}\}$ be the augmented domain for a feature $j$. Let $\mathbf{x}_\perp = (\texttt{NULL}, \ldots, \texttt{NULL})$ be a padding vector of dimension $d$, and let $L = \max_i T_i$ be the maximum sequence length across all users (or a truncation limit). The flattening transformation $\Phi : D^{(i)} \to \mathbf{y}_i$ maps a variable-length table to a fixed-size sequence in $\mathcal{Y} = (\mathcal{V}'_1 \times \cdots \times \mathcal{V}'_d)^L$ by appending $L - T_i$ padding vectors, or,

$$\mathbf{y}_i = \big(\mathbf{x}_1^{(i)}, \ldots, \mathbf{x}_{T_i}^{(i)}, \underbrace{\mathbf{x}_\perp, \ldots, \mathbf{x}_\perp}_{L-T_i}\big). \tag{3}$$

While applying $\Phi$ enables the use of standard row-based DPSD mechanisms (e.g., AIM), it introduces a critical utility bottleneck rooted in the privacy budget. State-of-the-art tabular data synthesis methods measure low-degree marginals, as these are the least expensive, highest impact measurements in terms of privacy budget (Liu et al., 2021b;

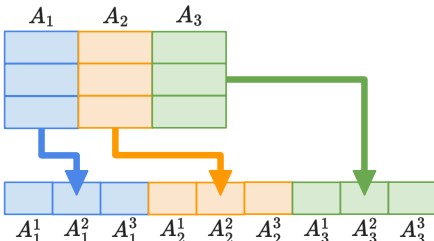

*Figure 2.* **The Flattening Transformation ($\Phi$).** Visualizing how a user's longitudinal history (left) is concatenated into a single high-dimensional vector (right). Each timestep becomes a distinct set of attributes (e.g., $A_1^1, A_1^2, \ldots$), effectively multiplying the domain dimensionality by the sequence length $L$.

McKenna et al., 2024; Rosenblatt et al., 2025). In this high-dimensional flattened space, these mechanisms measure low-order, local marginals (e.g., correlations between adjacent time steps $\mathbf{x}_t$ and $\mathbf{x}_{t+1}$). They do not, by default, measure all global dependencies; there are exponentially many of these, and it would be prohibitively expensive from both a computational and privacy budget perspective. We show an example in Appendix E (Proposition E.2) of how, even if these methods preserve local step-wise plausibility, they could fail to maintain global consistency across the entire timeline. Consequently, if the true data contains distinct user "types" (e.g. healthy adults compared with adults requiring extensive medical visits) where the state at time $t$ depends on the initial state at time 1, a locally-constrained mechanism would "mix" these trajectories, hallucinating spurious paths that are locally plausible but globally invalid. Additionally, we note that a trajectory that looks locally correct but is actually globally incoherent is a failure mode that is difficult to detect with standard marginal metrics, but is precisely what our TDCR metric is designed to penalize.

## 5. Proposing the `PATH` Generative Framework

We present `PATH`, a framework that captures the complex, time-dependent structure of user-level tabular data in an autoregressive manner. In contrast to flattening methods that attempt to model the joint distribution of a fixed-width vector, we treat the generation of a user's trajectory as a conditional sequence generation task.

**Serialization.** Our deterministic serialization schema $\Psi : D^{(i)} \to \mathbf{s}_i$, maps a user's table to a token sequence. As shown in Figure 3, the sequence begins with a schema definition, followed by the row-by-row encoding. We include row-delimiting tokens (e.g., `[Row i]`) to encourage the model to distinguish between temporal steps.

The serialization format is designed to be invertible; any valid output string generated by the model can be parsed back into a structured table $D^{(i)*}$. We seek to encode the "history" of the user, implicitly capturing temporal dependencies via the language model's context window rather

than explicitly modeling the state-space.

```
Columns:   charttime, heartrate, ...
[Row 1]:   charttime is 2180-07-22
16:36:00, heartrate is 83.0, ...
[Row 2]:   charttime is 2180-09-22
16:43:00, heartrate is 85.0, ...
```

*Figure 3.* The serialization format transforms longitudinal tables into a token sequence. The model learns to predict the next row conditioned on the schema and the history of previous rows.

**Data Construction and Privacy Unit.** Our framework strictly adheres to our *privacy unit*; to ensure that the DP guarantee protects the *entire* trajectory of a user rather than just isolated rows, we enforce that each user $u_i$ contributes exactly one example to the gradient update per epoch.

To ensure robust autoregressive generation capabilities at any stage of the timeline, we employ a dynamic windowing strategy. For each user $u_i$ at each epoch, we construct a single training example $(\mathbf{c}_i, \mathbf{t}_i)$ by sampling a split point $k \in [0, T_i - 1]$. The model is provided with the context $\mathbf{c}_i = \Psi(\mathbf{x}_{1:k}^{(i)})$ and trained to generate the target continuation $\mathbf{t}_i = \Psi(\mathbf{x}_{k+1:T_i}^{(i)})$. We then resample $k$ independently at every epoch. This strategy allows the model to eventually learn from the full diversity of the user's history and transition dynamics across the training process while maintaining a bounded sensitivity of 1 per user per step. We also want to train the model to iteratively generate a table *one row at a time*, which allows us to validate each row, and thus enforce coherence for the full generated table.

To ensure the model learns to generate trajectories at various stages of completion, we employ what we call a "density-based" sampling strategy. Rather than selecting the split point $k$ uniformly, we sample $k$ based on the empirical distribution of table lengths in the training set.[1] This prevents the model from being biased toward short contexts if the data contains a heavy tail of long trajectories. Specifically, with probability $p_{\text{start}}$, we select $k = 0$ (generating the first row from scratch); otherwise, we sample $k$ from the pool of all valid historical indices observed in the training corpus. We chose density-based sampling over uniform split-point selection or fixed splits because it naturally adapts to the distribution of trajectory lengths: uniform sampling over-represents late split points for long trajectories, while fixed splits prevent the model from seeing the full diversity of context lengths.

**Architecture and DP Fine-tuning.** We fine-tune models from the Gemma 3 family (Team et al., 2025), specifically the 1B and 4B parameter variants (using the pre-trained checkpoints). We utilize Low-Rank Adaptation (LoRA) (Hu et al., 2021) with a rank of $r = 128$, which allows us

[1]We assume this information to be public.

to update a small fraction of parameters while freezing the pre-trained weights. This is particularly effective for DP training, as it reduces the dimensionality of the gradient updates that must be noised (Kurakin et al., 2024). For more details on procedure and hyperparameters (which we fix for all datasets), see Section A.

**Inference, Parsing, and Private Selection** At inference time, we generate synthetic tables autoregressively, conditioning the generation of row $\mathbf{x}_t$ on the schema and the history of previously generated rows $\mathbf{x}_1, \ldots, \mathbf{x}_{t-1}$. Because DP fine-tuned models can yield malformed strings, we implement a robust parsing strategy to maximize data yield. For each generated step, we sequentially attempt to: (1) parse the output as structured key-value pairs (e.g., "col is val"); (2) fall back to parsing as comma-separated values matching the schema order if the first method fails; and (3) apply partial infilling to repair missing static identifiers (e.g., `subject_id`) by propagating values from the synthetic user's existing history. If a row fails all validation checks, such as containing non-numeric values in numeric columns or invalid dates, the generation for that specific table is terminated early. This is fine, as we are able to "over-generate" table examples without consuming additional privacy budget. Finally, to ensure the final dataset $\mathbf{D}^*$ has high utility, we employ a private selection strategy to post-process and filter a large, over-generated candidate pool (Ponomareva et al., 2025). See Appendix Section A for more details on the private selection procedure and parameters.

### 5.1. Baseline Approaches

**Real Data Subsampling (Reference).** To contextualize our metric scores, we evaluate a non-synthetic reference baseline consisting of real user tables uniformly subsampled from the training set (this is obviously not DP). We evaluate these subsamples at three distinct scales: 100, 1,000, and 10,000 unique tables. This comparison is important for interpreting distributional metrics like TDCR and MAUVE, as it establishes the "natural" variance and divergence one would expect simply due to finite sampling.

**Marginal-based Methods (on Flattened Data).** As discussed in Section 4, traditional DP tabular methods are designed to preserve relationships between columns of the data but do not generally preserve relationships between rows owned by the same user. To apply them to longitudinal data $\mathbf{D}$, we first apply the flattening transformation $\Phi$ (Section 4). Note that for the flattened baselines we enforce a strict fixed-width window to ensure consistent marginal measurements. We filter the cohort to users with at least $L$ timesteps (e.g., $L = 10$ for the MIMIC dataset) and truncate all trajectories to exactly this length. This results in a "wide" schema where each user contributes a single row with $d \times L$ columns. Since marginal methods must dis-

cretize the domain, a single outlier can force an artificially wide bin that disproportionately assigns probability mass to implausible values. To correct for this, we heuristically clip all generated values to the [min, 99th percentile] range of the real private data. This is technically a violation of differential privacy (a DP version of this same approach would be strictly worse); we disregard this as it artificially strengthens the baselines, ensuring they are not penalized for susceptibility to outliers but rather evaluated on their core temporal modeling capacity.

**Direct Mechanism.** We implemented a baseline following the *Select-Measure-Estimate* paradigm (McKenna et al., 2019) with a fixed, heuristic selection strategy designed to capture temporal dynamics manually. To make this baseline exceptionally competitive (at the cost of using prior domain knowledge), we explicitly selected all 1-way marginals to capture feature distributions at each timestep and a specific set of 2-way marginals between adjacent time steps for every feature (e.g., measuring the joint distribution of $(\texttt{heartrate}_t, \texttt{heartrate}_{t+1})$). In total, for a sequence length of $L = 10$ and $d = 9$ attributes (as is the case for MIMIC), this results in 90 1-way marginals and 81 temporal 2-way marginals. We used the Gaussian mechanism to measure these marginals with a uniform budget allocation and leveraged the Private-PGM package (McKenna et al., 2019) to estimate the data distribution. This baseline effectively imposes a Markovian assumption, where the value for a feature at time $t$ depends explicitly on its value at time $t-1$; we refer to it as "Direct (Mark.)" in our results.

We consider a second variant of the Direct Mechanism (referred to as "Direct (Across)") where, in addition to the temporal correlations, we attempt to measure the relationship between different features at the same time step (e.g., $(\texttt{heartrate}_t, \texttt{resprate}_t)$). Since measuring all $\binom{d}{2}$ attribute pairs across all $L$ time steps creates a measurement set too large for tractable estimation (causing GPU memory exhaustion during the PGM step), we randomly subselect up to 80 of these intra-step marginals to include in the mechanism. This variant tests whether its more important to capture instantaneous correlations over Markovian steps.

**AIM.** We further evaluated the **AIM** algorithm (McKenna et al., 2024), a leading workload-adaptive method. Unlike the Direct Mechanism, AIM automatically selects marginals using a greedy selection procedure that accounts for data, budget, and computational constraints (McKenna et al., 2024; Ponomareva et al., 2025). We provided AIM with the flattened dataset and allowed it to iteratively select the most informative marginals until the privacy budget was exhausted. This tests whether an automated selection metric can identify better temporal correlations than our manual heuristic in the high-dimensional flattened space, but we did not expect it to outperform the Direct mechanism, as Direct

encodes knowledge about the temporal structure which AIM presumably needs to use some privacy budget to learn.

**Foundation Model Prompting (Non-Private).** To establish a utility upper bound for LLM-based generation, we evaluated the capabilities of Gemini 2.5 Flash-Lite (FL) without private fine-tuning (Comanici et al., 2025). We utilized In-Context Learning (ICL) with varying numbers of demonstration examples ($k$). We used (1) Zero-shot ($k = 0$) where the model is provided only with the schema and a natural language instruction to "generate a realistic tabular dataset for a patient," (2) Few-shot ($k \in \{1, 5, 10\}$), where we construct a prompt containing $k$ real user tables randomly sampled from the training set, formatted using the same serialization scheme described in Section 5. The model is then tasked with generating a new, unique user table. See Appendix Section H for prompt text/structure.

# 6. Experimental Setup

**Data.** We evaluate our framework on three distinct datasets designed to test temporal coherence, generalization to unseen time periods, and ground-truth likelihoods. Details are provided in Appendix C.

**Vital signs subset of the MIMIC-IV dataset.** The privacy unit is a single patient (subject_id). To focus on the most common temporal trajectories, we filter the cohort to include only patients with sequence lengths i.e. charttimes of $4 \leq T \leq 50$, resulting in a final dataset of 102,864 unique patient trajectories (Johnson et al., 2020).

**NYC 311 calls.** To assess spatiotemporal fidelity and generalization, we use this data from October 1st, 2024, to August 1st, 2025 (NYC Dept. of Info Tech, 2026). This time range post-dates the training cutoff of the pre-trained Gemma 3 models, ensuring performance reflects generalization rather than training data memorization. The privacy unit is a unique property (Borough-Block-Building (BBL)). This dataset contains 342,959 longitudinal service histories.

**Synthetically generated data (HMM).** As a controlled check for our methods and to measure the preservation of latent temporal states against a known ground truth, we generated a synthetic dataset using an HMM with multivariate Gaussian emissions (Durbin et al., 1998) (see Appendix Section C). Unlike real data, where the true generative process is unknown, this allows us to rigorously evaluate the model's ability to recover the exact transition probabilities and emission distributions governing the data.

# 7. Experimental Results

**The Utility Limits of Flattening.** Our results on the MIMIC-IV dataset, summarized in Table 1, empirically demonstrate the issues with flattening longitudinal data.

When trajectories are concatenated into high-dimensional vectors, marginal-based methods struggle: at $\varepsilon = 2.0$, AIM achieves a TDCR of $0.736$ compared to e.g. $0.289$ for `PATH` (Gemma 4B). This gap indicates that AIM produces trajectories outside the true data manifold. Furthermore, flattening can dilute the privacy budget across the massive padded schema ($d \times L$), degrading univariate utility. For example, AIM has a univariate marginal divergence of 9.71 (`PATH` has 3.30) – a failure mode visualized in Figure 4.

**Temporal Fidelity vs. Marginal Accuracy.** The Synthetic dataset, constructed via a ground-truth HMM, reveals a trade-off between column-wise independence and sequential logic. As shown in Table 14, though AIM achieves a superior univariate marginal divergence score (0.28 at $\varepsilon = 10.0$) compared to `PATH` (Gemma 4B) (0.83), this marginal precision does not translate to preserved temporal dynamics. On the HMM Likelihood metric, which measures the probability of the generated sequences under the true latent process, `PATH` (both Gemma 1B and Gemma 4B) significantly outperforms AIM (50.99 vs. 319.43). While marginal methods accurately reproduce aggregate counts, they fail to capture the conditional probability $P(\mathbf{x}_t \mid \mathbf{x}_{<t})$. Conversely, `PATH` successfully learns the state transitions; this is qualitatively evident in the likelihood distributions shown in Figure 5 and quantitatively corroborated by the state transition divergence across all datasets (Tables 1-3), where Gemma 4B consistently yields lower Frobenius norms (e.g. 0.38 on MIMIC) compared to e.g. AIM (0.62).

**Distributional Manifolds and Generalization.** Beyond temporal metrics, we assess the overall distributional quality using MAUVE. On the complex, real-world datasets (MIMIC-IV and NYC 311), `PATH` (Gemma 4B) model consistently achieves the highest MAUVE scores among private methods. Specifically on MIMIC, `PATH` (Gemma

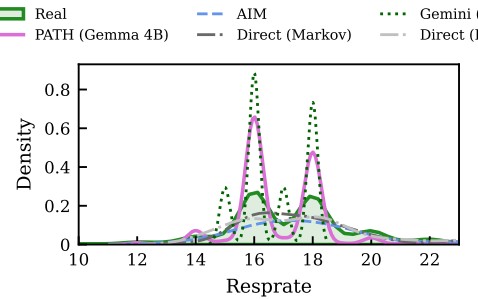

**Figure 4. Univariate Density Analysis (Respiratory Rate, MIMIC-IV).** We compare the marginal density of respiratory rate distributions across methods at $\varepsilon = 2.0$. While marginal-based baselines (AIM, Direct) achieve reasonable Wasserstein scores by broadly covering the support, they suffer from quantization smoothing and noise artifacts that obscure the data's natural shape. In contrast, `PATH` closely tracks the sharp modal peaks and specific skew of the real distribution (shaded green), demonstrating that `PATH` can preserve fine-grained univariate trends.

*Table 1.* Performance comparison on the MIMIC dataset ($\varepsilon \in \{2.0, 10.0\}$), evaluating privately fine-tuned LLMs against flattened marginal baselines. We include non-private Gemini 2.5 (10-shot or 10s) and Real Data subsamples as reference (upper bound) baselines. The results demonstrate that our autoregressive approach significantly outperforms marginal methods on temporal fidelity (State Trans.) and manifold coverage (MAUVE, TDCR) while also achieving superior marginal divergence. Notably, private fine-tuning achieves better distributional alignment than 10-shot prompting of the larger, non-private Gemini 2.5 Flash-Lite model. We follow the Olympic convention: gold represents the best metric values, silver the second best, and bronze third.

| $\varepsilon$ | Method | MAUVE ($\uparrow$) | TDCR (JSD $\downarrow$) | Marginal Div. (Avg Wass. $\downarrow$) | State Trans. (Avg Frob. $\downarrow$) |
|---|---|---|---|---|---|
| 2 | AIM | 0.564 | 0.7359 | 9.7080 | 0.7412 |
| | DIRECT (Acr.) | 0.493 | 0.7981 | 8.5031 | 0.6910 |
| | DIRECT (Mark.) | 0.573 | 0.7960 | 8.5788 | 0.5640 |
| | PATH (1B) | 0.596 | 0.4315 | 2.9174 | 0.3633 |
| | PATH (4B) | 0.661 | 0.2886 | 3.2980 | 0.3867 |
| 10 | AIM | 0.531 | 0.7349 | 5.7400 | 0.6243 |
| | DIRECT (Acr.) | 0.497 | 0.7665 | 8.3101 | 0.7076 |
| | DIRECT (Mark.) | 0.596 | 0.7791 | 8.4024 | 0.5497 |
| | PATH (1B) | 0.552 | 0.3001 | 2.6486 | 0.4454 |
| | PATH (4B) | 0.651 | 0.3784 | 3.4125 | 0.3747 |
| $\infty$ | Gemini 2.5 FL (10s) | 0.269 | 0.7894 | 5.9262 | 1.2283 |
| | REAL (10k) | 0.848 | 0.2322 | 0.3345 | 0.0161 |

*Table 2.* Performance comparison on the HMM dataset ($\varepsilon \in \{2.0, 10.0\}$). Our `PATH` (Gemma 4B) model achieves superior performance on temporal coherence metrics (HMM Likelihood, State Trans.) and distributional variety (TDCR).

| $\varepsilon$ | Method | MAUVE ($\uparrow$) | HMM Likelihood (Wass. $\downarrow$) | TDCR (JSD $\downarrow$) | Marginal Div. (Avg Wass. $\downarrow$) | State Trans. (Avg Frob. $\downarrow$) |
|---|---|---|---|---|---|---|
| 2 | AIM | 0.655 | 374.47 | 0.6352 | 0.4260 | 0.5204 |
| | DIRECT (Acr.) | 0.763 | 696.55 | 0.7384 | 1.5437 | 0.6014 |
| | DIRECT (Mark.) | 0.701 | 754.96 | 0.8059 | 1.8809 | 0.5953 |
| | PATH (1B) | 0.307 | 334.46 | 0.8073 | 3.0628 | 1.0329 |
| | PATH (4B) | 0.612 | 57.97 | 0.4150 | 0.9494 | 0.4363 |
| 10 | AIM | 0.705 | 319.43 | 0.5367 | 0.2757 | 0.4791 |
| | DIRECT (Acr.) | 0.704 | 409.11 | 0.6560 | 0.6496 | 0.5331 |
| | DIRECT (Mark.) | 0.710 | 653.24 | 0.6748 | 0.9022 | 0.4558 |
| | PATH (1B) | 0.359 | 184.90 | 0.5402 | 2.5169 | 0.7138 |
| | PATH (4B) | 0.690 | 50.99 | 0.3966 | 0.8250 | 0.4217 |
| $\infty$ | Gemini 2.5 FL (10s) | 0.268 | 9.02 | 0.5564 | 0.5596 | 0.6595 |
| | REAL (10k) | 0.856 | 0.33 | 0.1778 | 0.0325 | 0.0096 |

*Table 3.* Performance comparison on the NYC 311 dataset ($\varepsilon \in \{2.0, 10.0\}$). The `PATH` (Gemma 4B) model has high distributional fidelity (MAUVE) and temporal accuracy (Temporal Dist.) compared to the 1B model and the non-private baseline. Note that marginal baselines are excluded here as they failed to scale to the spatiotemporal schema of this dataset.

| $\varepsilon$ | Method | MAUVE (Gecko $\uparrow$) | Classifier AUC (Ideal 0.5) | Temporal Dist. (Wass. $\downarrow$) | Transition Div. (Frobenius $\downarrow$) |
|---|---|---|---|---|---|
| 2 | PATH (1B) | 0.311 | 0.955 | 1.400 | 0.605 |
| | PATH (4B) | 0.507 | 0.948 | 0.806 | 0.975 |
| 10 | PATH (1B) | 0.464 | 0.971 | 0.870 | 0.279 |
| | PATH (4B) | 0.634 | 0.948 | 0.870 | 0.405 |
| $\infty$ | Gemini 2.5 FL (10s) | 0.074 | 1.000 | 2.537 | - |
| | REAL (10k) | 0.876 | 0.470 | 0.312 | 0.165 |

4B) ($\varepsilon = 10.0$) reaches a MAUVE score of $0.651$, surpassing the Direct Mechanism ($0.596$). We note that while marginal baselines outperform Gemma on the simpler Synthetic dataset (Table 2), this is likely due to the fact that the Synthetic data is very well behaved and thus admits the quantization necessary for the PGM based-baselines.

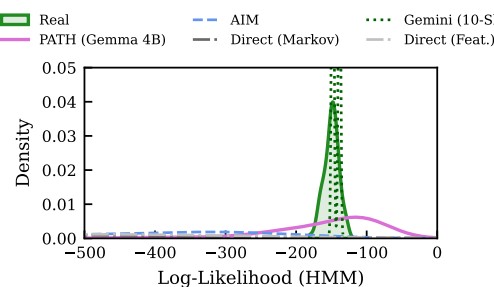

*Figure 5.* **HMM Log-Likelihood Distribution Analysis (Synthetic Dataset).** We evaluate long-range temporal coherence by scoring generated trajectories against the ground-truth HMM. The real data (shaded green) gives the variance in likelihood scores from our random process. The marginal-based baselines (AIM, Direct) produce a long tail of low-likelihood sequences (shifting left), indicating the generation of temporally incoherent trajectories that violate the underlying state transition logic. While non-private Gemini 2.5 FL (10-shot) produces a tight band of highly likely sequences, it fails to capture the full variance of the distribution (indicative of mode collapse). In contrast, PATH more successfully models the stochastic nature of the true process, producing a likelihood distribution closer to the support of the HMM data.

We note that the non-private, few-shot in-context learning (on Gemini 2.5 FL) often underperformed our private fine-tuning; for instance, achieving a MAUVE score of only 0.269 on MIMIC. While few-shot prompting can copy local structure, it struggles to generalize the full distribution from a limited context window ($k = 10$). PATH appears to internalize the distributional manifold over tables into the model weights, enabling the generation of a diverse table collection that better covers the real support. We note that classifier AUC scores remain near 1.0 for all methods on MIMIC (see Figure 14), indicating that synthetic data remains statistically distinguishable from real data; however, our TSTR results confirm this does not preclude downstream utility.

**Model Scaling and Privacy Robustness.** Comparing the 1B and 4B parameter variants of Gemma for PATH demonstrates the importance of model scale, particularly at lower privacy budgets. While both models perform comparably at $\varepsilon = 10.0$, the 4B model exhibits significantly higher robustness as $\varepsilon$ decreases. On MIMIC at $\varepsilon = 0.5$, the Gemma 1B model suffers a notable degradation in fidelity, with MAUVE scores dropping to 0.531 compared to 0.627 for the 4B model, and marginal divergence worsening from 3.00 (4B) to 3.93 (1B). We hypothesize that the larger model's stronger pre-trained semantic priors provide a more stable initialization, allowing it to retain structural coherence even when the DP-SGD updates are dominated by noise. This trend holds for the NYC 311 dataset as well, where Gemma 4B consistently outperforms the 1B variant. Larger models are likely better equipped to handle complex schemas involving geospatial and high-dimensional categorical data.

*Table 4.* **Variance analysis for PATH (Gemma 1B) at $\varepsilon = 2.0$.** We report mean $\pm$ standard deviation across 5 independent seeds on the MIMIC and Synthetic HMM datasets. We found low variance across seeds for DP-SGD. Variance analysis for the 4B model is omitted due to computational constraints.

| Dataset | Marginal Div. (Avg Wass. ↓) | State Trans. (Avg Frob. ↓) | TDCR (JSD ↓) |
|---|---|---|---|
| MIMIC | $3.01 \pm 0.16$ | $0.37 \pm 0.02$ | $0.48 \pm 0.04$ |
| HMM Dataset | $2.99 \pm 0.19$ | $0.84 \pm 0.34$ | $0.80 \pm 0.05$ |

**Varying Seeds.** To check consistency over different randomness, we ran PATH (Gemma 1B) with 5 independent seeds at $\varepsilon = 2.0$ on both the MIMIC and Synthetic HMM datasets. We focus on this representative setting as $\varepsilon = 2.0$ a common primary privacy regime for synthetic data (Ponomareva et al., 2025). Table 4 shows that the variance across seeds is small relative to the performance gaps reported in Tables 1-2; moreover, the consistency of our findings across multiple LLM backbones, LoRA ranks, and privacy budgets (see Appendix G) provides additional evidence that the results are robust.

**Comparison with Sequential DP Baselines.** We additionally compared PATH against a DP version of Doppel-GANger (Lin et al., 2020), a GAN-based sequential synthesizer (Table 17 in the Appendix). PATH outperforms DoppelGANger by $\sim$22$\times$ on marginal fidelity and $\sim$3$\times$ on transition divergence, consistent with prior findings that DP-trained GANs struggle on structured tabular data (Chen et al., 2025).

**Downstream Utility.** To confirm that distributional fidelity translates to practical utility, we conducted Train-on-Synthetic, Test-on-Real (TSTR) evaluations (Tables 22–23 in the Appendix). On NYC 311, PATH achieves 53.8% accuracy on next-complaint-type prediction (vs. 58.5% for real data and 32.1% for majority). On MIMIC, PATH shows positive lift (AUC 0.57 vs. 0.50 majority) on ED stay-length prediction, while AIM fails to exceed chance (AUC 0.47).

**Computational Cost and Practical Guidelines.** The primary trade-off of PATH is computational overhead: training takes $\sim$62 minutes for Gemma 1B on a single GPU, compared to $\sim$6 minutes for AIM (Table 18 in the Appendix). Generation is also slower (seconds per table vs. milliseconds) but is parallelizable. We recommend PATH when temporal dynamics matter (e.g., medical trajectories, event sequences). Conversely, marginal methods such as AIM should be preferred when the user primarily needs accurate marginal distributions and temporal ordering is not semantically meaningful, or when GPU resources are unavailable. Additional ablations on LoRA rank, LLM backbone, and budget splitting are provided in Appendix G.

## 8. Related Work

See Section B for an extended discussion of related work.

**Marginal-based Mechanisms.** Mechanisms based on low-order marginals, such as Private-PGM (McKenna et al., 2019) based methods like AIM (McKenna et al., 2024), currently represent the state-of-the-art for i.i.d. tabular synthesis (Ponomareva et al., 2025; Rosenblatt et al., 2024; Chen et al., 2025). However, as we discussed, applying these mechanisms to longitudinal data requires flattening and extensive preprocessing (e.g., discretization), which can discard important features of the raw data.

**Transformer-based Synthesis.** Following GReaT (Borisov et al., 2022), recent works serialize tabular data to fine-tune LLMs. To prevent the loss of structural coherence under DP-SGD (Yu et al., 2022), current methods often require complex two-stage training (learning syntax on public data, then distribution on private data) (Afonja et al., 2025; Tran & Xiong, 2024). While non-private methods like TabPF-Gen (Ma et al., 2024) leverage strong tabular foundation models like TabPFN (Hollmann et al., 2023) for generation, they lack DP adaptations and are potentially ill-suited for modeling long-range temporal dependencies.

**Longitudinal DPSD.** Research on synthetic DP time-series data is limited. GAN-based approaches like DoppelGANger (Lin et al., 2020) and NetShare (Yin et al., 2022) often suffer significant utility loss under DP-SGD. Marginal adaptations like NetDPSyn (Sun et al., 2024) use PrivSyn (Zhang et al., 2020) and focus on specific features (e.g., inter-arrival times); they assume event-level privacy rather than protecting full user trajectories, and thus target a different objective. In our experiments, we evaluate a DP-enhanced version of DoppelGANger on the MIMIC dataset and find that PATH substantially outperforms it across all metrics (see Appendix G).

**Comparison of PATH with the above.** We operationalize the multi-table privacy unit suggested by Ponomareva et al. (2025), and are the first to apply autoregressive LLMs. Unlike marginal methods, LLMs capture high-order dependencies (Castellon et al., 2023) essential for longitudinal logic. In contrast to prior Transformer approaches to DPSD, PATH targets a different privacy objective (protecting user owned tables instead of rows) and achieves high fidelity with a single-stage training process and no preprocessing.

## 9. Conclusion

Considering our findings across medical, civic, and synthetic settings, we arrive at four primary conclusions. **First**, the "unit of synthesis" matters; treating a user's trajectory as a single autoregressive sequence is a more efficient representation than flattening user histories. **Second**, there is a distinct dichotomy in utility; while marginal-based methods capture aggregate statistics well, LLMs dominate at capturing temporal dynamics and state transitions while also preserving

these statistics, offering a "best of both worlds" solution. **Third**, private fine-tuning is more effective than in-context learning for distribution matching; few-shot prompting often leads to mode collapse, whereas fine-tuning captures population variety. **Fourth**, larger is better; Gemma 3 4B was better than its 1B variant at maintaining high temporal fidelity even under strict privacy guarantees.

In this work, we introduced Private Autoregressive Trajectory Histories (PATH), a generative framework that redefines the unit of differential privacy from the individual row to the entire user table. PATH leverages the autoregressive capabilities of privately fine-tuned LLMs, and addresses the fundamental limitations of flattening longitudinal data. Our extensive empirical analysis on the MIMIC-IV, NYC 311, and synthetic HMM datasets demonstrates that PATH yields superior fidelity across a suite of metrics when compared to state-of-the-art marginal-based baselines and prompting foundation models. These results establish that LLMs can serve as effective, privacy-preserving synthesizers in sequential tabular domains, enabling robust data sharing of sensitive temporal data.

**Limitations.** We identify several limitations of our work. First, PATH's advantage is strongest for data with complex temporal structure; for simple tabular synthesis tasks where marginal distributions are the primary concern and temporal ordering is not semantically meaningful, marginal methods such as AIM may suffice at lower computational cost. Second, PATH requires GPU resources and is approximately $10\times$ slower to train than marginal methods (see Table 18), though training is a one-time cost and generation is parallelizable. Third, we do not include empirical privacy attack evaluations (e.g., membership inference); however, PATH satisfies formal $(\varepsilon, \delta)$-DP by construction via DP-SGD and the Gaussian mechanism during private selection, providing worst-case guarantees independent of any specific attack, and prior work has shown that membership inference attacks are ineffective against correctly implemented DP training (Stadler et al., 2022).

## Impact Statement

This paper presents work in the domain of privacy-preserving data synthesis. By enabling the generation of high-fidelity synthetic longitudinal data, PATH facilitates the sharing of sensitive information that would otherwise remain siloed due to privacy concerns. This has the potential to accelerate research in healthcare and public policy. However, we acknowledge that synthetic data, even when differentially private, has risks. Practitioners should rigorously evaluate these potential risks (to patient confidentiality, etc.) before deploying DPSD in decision-making systems.

## Acknowledgements

We thank Alex Bie and Weiwei Kong for their help with the experimental infrastructure for this paper.

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

# A. Architecture, Hyperparameters and Privacy Guarantees

We optimize the Gemma 1B and 4B models under `PATH` using DP-SGD (Abadi et al., 2016). We calibrate the noise multiplier $\sigma$ to satisfy a *total* privacy budget $\varepsilon_{total} \in \{0.5, 2.0, 4.0, 10.0\}$ with $\delta = 1/n^2$, where $n$ is the number of unique subjects in the training set. We reserve a small portion of the privacy budget for the selection step (described below), such that $\varepsilon_{total} = \varepsilon_{train} + \varepsilon_{select}$.

We perform hyperparameter tuning on the learning rate and batch size using the Synthetic dataset, converging on a batch size of 256 and a peak learning rate of $5 \times 10^{-4}$ with cosine decay. We utilize a fixed per-sample gradient clipping norm $C = 0.01$. These hyperparameter values are then used for all experiments on all datasets.

**DP Post-hoc Selection via Private Selection.**  To ensure the final dataset $\mathbf{D}^*$ has high utility, we employ a private selection strategy to post-process and filter a large, over-generated candidate pool (Ponomareva et al., 2025). To identify the most representative tables, we again map both real and synthetic data into a semantic vector space using the Gecko model (Lee et al., 2024). We execute a differentially private voting mechanism where each private table casts votes for its $k = 10$ nearest synthetic neighbors in this embedding space. The final dataset is composed of the candidates that receive the highest number of votes after the addition of privacy-preserving noise. For this selection step, we allocate a dedicated budget of $\varepsilon_{select} = 1.0$, scaling down to $\varepsilon_{select} = 0.5$ and $\varepsilon_{select} = 0.25$ for our tighter total budgets of $\varepsilon = 2.0$ and $\varepsilon = 0.5$, respectively.

**Privacy Guarantees and Parameters.**  Additionally, as suggested in (Ponomareva et al., 2023), we provide a standardized report with important privacy specs from our experimentation.

---

**Privacy Guarantees and Parameters**

1. **DP setting.** This work utilizes a *Central DP* model. The privacy guarantee applies to the release of the final synthetic dataset.
2. **Instantiating the DP Definition.**
   (a) *Data accesses covered.* The privacy guarantee covers the training of the specific Generative Model (Gemma) and the subsequent Private Selection of synthetic tables. *Note on Hyperparameters:* Hyperparameter tuning (learning rate, batch size) was performed on a separate *Synthetic HMM* dataset and transferred to the real datasets. Therefore, the privacy budget reported is consumed entirely by the final training run and selection, without additional cost for tuning on the sensitive data.
   (b) *What the final mechanism's output is.* The mechanism $\mathcal{M}$ is defined as the composition of the DP-SGD fine-tuning, the autoregressive generation, and the Private Selection step. The output is the collection of synthetic tables $\mathbf{D}^*$. While the model weights are differentially private, the primary release artifact is the synthetic data.
   (c) *Unit of privacy.* User-level Privacy: the unit of protection is the full table $D^{(i)}$ associated with a single user (e.g., a specific Patient ID in MIMIC or a specific Property BBL in NYC 311). This protects the entire longitudinal history of the user, not just individual rows.
   (d) *Adjacency definition.* We utilize the Add-or-Remove definition of adjacency. Two datasets $\mathbf{D}$ and $\mathbf{D}'$ are neighboring if $\mathbf{D}'$ can be obtained by adding or removing all records associated with exactly one user $u_i$ from $\mathbf{D}$.
3. **Privacy accounting details.**
   (a) *Type of accounting used.* We utilize *Privacy Loss Distribution (PLD)* accounting to compute tight bounds for the composition of subsampled Gaussian mechanisms.
   (b) *Accounting assumptions.* The training process utilizes random shuffling (sampling without replacement) rather than Poisson sampling, where each user contributes exactly one serialized sequence to the gradient update per epoch. The noise multiplier $\sigma$ is calibrated to satisfy the target $\varepsilon$ given the sampling rate $q = B/n$ and number of epochs.
   (c) *The formal DP statement.* The release satisfies $(\varepsilon_{total}, \delta)$-DP, where $\delta = 1/n^2$ (e.g. approx $10^{-10}$ for MIMIC). The total budget is composed of training and selection budgets: $\varepsilon_{total} = \varepsilon_{train} + \varepsilon_{select}$. Reported regimes are $\varepsilon_{total} \in \{0.5, 2.0, 4.0, 10.0\}$.
4. **Transparency and verifiability.** We will provide serialization logic and selection mechanism code upon acceptance.

---

# B. Extended Related Work

**Marginal-based Mechanisms.** Mechanisms based on low-order marginal measurements currently represent the state-of-the-art for standard tabular data synthesis (Ponomareva et al., 2025). Popular approaches parameterize the distribution (e.g. the Private-PGM distributional model (McKenna et al., 2019)), and cleverly update the distributional parameters by selecting and measuring a subset of column correlations. The most performant of these methods is AIM (McKenna et al., 2024), which dominates recent benchmarks on independent and identically distributed (i.i.d.) tabular data (Rosenblatt et al., 2024; Chen et al., 2025). However, applying these mechanisms requires extensive preprocessing, including discretization, imputation, and outlier removal. Furthermore, by treating the data as purely numerical abstractions, these methods discard the semantic context inherent in column names and values. Additionally, some recent theoretical work has addressed the bounds of longitudinal release for streaming data (Bun et al., 2024), though these methods focus on continual aggregation rather than the synthesis of coherent individual trajectories.

**Transformer based models for DP Tabular data synthesis.** End-to-end methods utilizing Large Language Models (LLMs) for tabular synthesis have recently gained prominence. In the non-private setting, GReaT (Borisov et al., 2022) serializes tabular data into textual strings (e.g., "Sex is Male, age is 50") to fine-tune pre-trained LLMs. GReaT first creates a textual encoding of the tabular data (in the format "column_name is cell_value,") (Borisov et al., 2022) and then fine-tunes a pretrained LLM on such textual encodings. Column names are permutated during the encoding phase to avoid relying on pseudo-ordering of the column names and to allow sampling using any subset of column names. Adapting GReaT to the differentially private setting presents challenges; naive application of DP-SGD often results in the loss of structural coherence. To mitigate this, Afonja et al. (2025) and (Tran & Xiong, 2024) concurrently proposed a two-stage training paradigm: a first stage learns "format compliance" (syntax) on public data, followed by a second stage of DP fine-tuning on private data. Both works also introduce loss reweighting for formatting tokens (e.g., "is", ",") to further stabilize training.

A parallel line of work leverages TabPFN (Hollmann et al., 2023), a transformer pre-trained to perform supervised classification via in-context learning. TabPFGen (Ma et al., 2024) is an energy-based model that harnesses TabPFN for (non-DP) data synthesis. TabPFGen generates data via stochastic gradient Langevin dynamics (SGLD), iteratively refining noise into synthetic samples by maximizing the AUC of the in-context classifier. We exclude TabPFGen from our evaluation for several reasons. First, the method requires designating a specific label column, limiting its generality. Second, the iterative energy-based update is ill-suited for capturing the long-range temporal dependencies central to our problem setting. Finally, no differentially private adaptation of TabPFGen currently exists; given that TabPFN's weights are immutable and fixed during pre-training, we hypothesize that the method would struggle to model private distributions that drift significantly from its pre-training prior.

**Longitudinal DPSD.** There is a noticeable lack of research activity on modeling multi-dimensional time series in differentially private settings, likely due to the difficulty of the problem. DoppelGANger (Lin et al., 2020) adapted GAN architectures by incorporating LSTMs to capture temporal dynamics. However, their experiments demonstrated that training with DP-SGD, even under moderate privacy budgets (e.g., $\varepsilon = 10.5$), significantly degrades temporal correlations. Subsequent works, such as NetShare (Yin et al., 2022), reported similar utility losses with GAN-based approaches. This led to a paradigm shift toward marginal-based methods. For instance, NetDPSyn (Sun et al., 2024) adapted PrivSyn (Zhang et al., 2020) to model network traces, capturing temporal dynamics by introducing inter-arrival times as an explicit feature and measuring marginals over these differences. These works typically define the privacy unit at the example (or event) level. In contrast, our framework adopts a stricter user-level definition, aiming to preserve the entire temporal trajectory of each synthetic user.

**Comparison of `PATH` with the above.** Our work distinguishes itself by operationalizing the multi-table privacy unit identified by Ponomareva et al. (2025). To the best of our knowledge, we are the first to leverage the autoregressive capabilities of LLMs to address the temporal modeling challenges inherent to this setting. While marginal-based methods excel at low-order statistics, recent literature suggests that LLMs are superior when capturing complex, high-order interactions (Castellon et al., 2023). Our results confirm that this capability is essential for modeling the conditional dependencies of longitudinal data. Furthermore, unlike prior transformer-based approaches that require complex two-stage training (Afonja et al., 2025; Tran & Xiong, 2024), our framework achieves high fidelity with a single-stage standard next-token prediction loss. Finally, by learning directly from serialized tokens, we bypass the extensive preprocessing (e.g., discretization, outlier removal) required by marginal mechanisms, preserving the raw semantic richness of the data.

## C. Data

### C.1. MIMIC IV

MIMIC vitalsigns data was preprocessed, first partitioning into users by $subject\_id$ to create patient time-series data. We filtered out subjects who had more than 50 and fewer than 4 $charttime$ (many patients, as is evidenced by Table 5, had fewer than 4 $charttime$, but we wanted patients with interesting temporal trajectories). Thus, we had a final cohort of 102,864 patients, reduced from the initial 198,131.

*Table 5.* Comparison of MIMIC vital signs data before and after filtering.

| Metric | Before Filtering | | | After Filtering | | |
|---|---|---|---|---|---|---|
| | **Record Counts** | **Unique 'stay_id'** | **Unique 'charttime'** | **Record Counts** | **Unique 'stay_id'** | **Unique 'charttime'** |
| count | 198131 | 198131 | 198131 | 102864 | 102864 | 102864 |
| mean | 7.90 | 2.06 | 7.90 | 10.58 | 2.43 | 10.58 |
| std | 15.00 | 3.37 | 15.00 | 8.45 | 2.09 | 8.45 |
| min | 1 | 1 | 1 | 4 | 1 | 4 |
| 25% | 2 | 2 | 2 | 5 | 1 | 5 |
| 50% | 4 | 4 | 4 | 7 | 2 | 7 |
| 75% | 8 | 8 | 8 | 13 | 3 | 13 |
| max | 1144 | 1144 | 1144 | 50 | 30 | 50 |

*Table 6.* Sample table from the MIMIC-IV Vital-sign public sample data.

| subject_id | stay_id | charttime | temperature | heartrate | resprate | o2sat | sbp | dbp | rhythm | pain |
|---|---|---|---|---|---|---|---|---|---|---|
| 10014729 | 37887480 | 2125-03-19 13:22:00 | | 124 | 24 | 100 | 93 | 65 | | |
| 10014729 | 37887480 | 2125-03-19 18:28:00 | 98.9 | 106 | 18 | 100 | 115 | 70 | Sinus Tachycardia | 5 |
| 10014729 | 37887480 | 2125-03-19 13:07:00 | | 128 | 18 | 100 | 132 | 96 | Sinus Tachycardia | |
| 10014729 | 37887480 | 2125-03-19 16:23:00 | 99.8 | 115 | 22 | 97 | 114 | 45 | Sinus Tachycardia | 0 |

### C.2. NYC 311 Calls/Requests

We use this data because (1) it has interesting geo-spatial *and* temporal trends for the model to pick up on, and (2) because it was released *after* the Gemma 3 class of model cutoff date (which was August 2024, according to Google). So, even though we can assume the Gemma models have been exposed to similar 311 call data (as its been public since 2010 on NYC Open Data), these specific calls are new, from a year distribution the model was not specifically trained on.

We use October 1st, 2024, through Aug 1st, 2025 of NYC 311 service requests (a dataset from NYC OpenData). This data was preprocessed, where "user-level" here is each unique "borough-block-building" code (a tax code that essentially identifies a single property with potentially multiple tenants). We removed columns that were redundant, empty, or had less precise information. We additionally filtered property-level such that properties (BBLs) with fewer than 2 total calls were excluded. Conversely, properties with more than 300 calls were removed. Thus, we went from an initial set of 381,066 unique properties to a final dataset of 118,510; then, each table is a a time-series of 311 service requests for a specific property.

*Table 7.* Sample of processed 311 data (column names changed for brevity).

| key | created | closed | agency | complaint | descriptor | location_type | zip | address | resolution | bbl | latitude | longitude |
|---|---|---|---|---|---|---|---|---|---|---|---|---|
| 62640682 | 2024-10-03 14:35:06 | 10/03/2024 02:53:31 PM | NYPD | Illegal Parking | Unauthorized Bus Layover | Street/Sidewalk | 10004.00 | 10 PETER MINUIT PLAZA | (desc) | 1000030003.00 | 40.70 | -74.01 |
| 62655723 | 2024-10-04 14:20:44 | 10/04/2024 02:38:27 PM | NYPD | Illegal Parking | Unauthorized Bus Layover | Street/Sidewalk | 10004.00 | 10 PETER MINUIT PLAZA | (desc) | 1000030003.00 | 40.70 | -74.01 |
| 62651993 | 2024-10-04 14:44:45 | 10/04/2024 03:04:03 PM | NYPD | Noise - Park | Loud Music/Party | Park/Playground | 10004.00 | 10 PETER MINUIT PLAZA | (desc) | 1000030003.00 | 40.70 | -74.01 |
| 62653125 | 2024-10-04 23:25:30 | NaN | DHS | Homeless Person Assistance | NaN | Street/Sidewalk | 10004.00 | 10 PETER MINUIT PLAZA | (desc) | 1000030003.00 | 40.70 | -74.01 |
| 62672879 | 2024-10-06 13:04:18 | 10/06/2024 03:01:12 PM | DSNY | Vendor Enforcement | Food Vendor | Street | 10004.00 | 10 PETER MINUIT PLAZA | NaN | 1000030003.00 | 40.70 | -74.01 |

### C.3. Synthetic Data

It's been known for quite some time that a Hidden Markov Model (HMM) is a good model for temporal dependencies in data (Jurafsky & Martin). Thus, using a multivariate Gaussian emission space, we specify an HMM instance as $(\mathcal{Q}, \mathcal{O}, \pi, A, B)$.

The finite set of $N_s$ unobserved states is,

$$\mathcal{Q} = \{q_1, q_2, \ldots, q_{N_s}\},$$

and the $N_f$-dimensional space of real-valued vectors from which observations are drawn is,

$$\mathcal{O} = \mathbb{R}^{N_f}.$$

We have initial state probabilities ($\pi$), which is a vector of probabilities for the system's starting state at time $t = 1$, given,

$$\pi = [\pi_1, \pi_2, \ldots, \pi_{N_s}], \quad \text{where } \pi_i = P(S_1 = q_i).$$

The dynamics of our model are governed by the transition probability matrix, denoted by $A$. This is an $N_s \times N_s$ matrix where each element $A_{ij}$ specifies the probability of transitioning from state $q_i$ to state $q_j$. This captures the first-order Markov property of the sequence of latent states, or

$$A = \{A_{ij}\},$$

where $A_{ij} = P(S_t = q_j | S_{t-1} = q_i)$.

Finally, the connection between the latent states and the observable data is defined by the emission probability distributions, denoted by $B$. This is a set of $N_s$ probability distributions, where each distribution $b_i(\mathbf{o})$ gives the probability density of emitting the observation vector $\mathbf{o} \in \mathcal{O}$ given that the system is in state $q_i$. In this model, each emission distribution is a multivariate normal distribution with a state-specific mean $\mu_i$ and covariance matrix $\Sigma_i$.

$$B = \{b_1(\mathbf{o}), b_2(\mathbf{o}), \ldots, b_{N_s}(\mathbf{o})\}$$

where $b_i(\mathbf{o}) = p(\mathbf{O}_t = \mathbf{o} | S_t = q_i) \sim \mathcal{N}(\mu_i, \Sigma_i)$.

So, to generate for each subject, a sequence of observations $\mathbf{O}_1, \ldots, \mathbf{O}_T$ of length $T$ is drawn.

DAG of HMM Data Generating Process

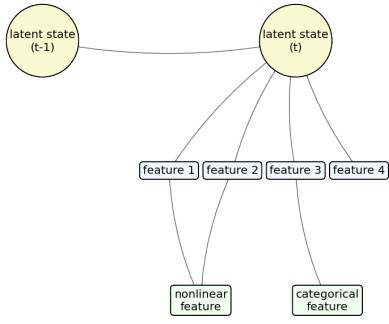

*Figure 6.* DAG for HMM.

*Table 8.* Sample of Synthetic data

| subject_id | timestep | Glomozole | Crirodex | Criphecor | Zolsidex | Zolphephine | Zolronide |
|---|---|---|---|---|---|---|---|
| 10 | 0 | -0.80 | -2.10 | -3.60 | 0.60 | 3.80 | Low |
| 10 | 1 | 2.70 | -1.80 | 3.80 | 3.10 | 3.90 | Very High |
| 10 | 2 | 0.70 | -2.70 | 2.70 | 3.40 | 8.00 | High |
| 10 | 3 | 0.60 | -0.60 | 2.30 | 3.70 | 1.30 | Medium |
| 10 | 4 | 1.00 | -1.50 | 4.10 | 2.40 | 2.00 | Very High |

## C.4. Example Real Data Tables

To provide qualitative context for interpreting our metrics, we present randomly selected real trajectory examples from each dataset below. These illustrate the diversity of temporal patterns, missingness structures, and value ranges that `PATH` must learn to reproduce.

*Table 9.* **Example MIMIC-IV patient trajectory (randomly selected).** This patient exhibits atrial fibrillation with a transition back to sinus rhythm. Note the natural missingness patterns (e.g., missing temperature and o2sat readings) and the temporal evolution of vital signs across multiple chart times.

| subject_id | stay_id | charttime | temperature | heartrate | resprate | o2sat | sbp | dbp | rhythm | pain |
|---|---|---|---|---|---|---|---|---|---|---|
| 10026255 | 34236274 | 2201-07-07 13:11:00 | 98.1 | 92 | 23 | 96 | 146 | 71 | Sinus Rhythm | 8 |
| 10026255 | 34236274 | 2201-07-07 13:52:00 | | 150 | 18 | 93 | 133 | 101 | Atrial Fibrillation | 8 |
| 10026255 | 34236274 | 2201-07-07 14:54:00 | | 136 | 22 | | 135 | 80 | Atrial Fibrillation | 8 |
| 10026255 | 34236274 | 2201-07-07 16:01:00 | | 130 | 24 | | 139 | 85 | Atrial Fibrillation | 7 |
| 10026255 | 34236274 | 2201-07-07 16:14:00 | 99 | 142 | 26 | 92 | 132 | 88 | Atrial Fibrillation | 7 |
| 10026255 | 34236274 | 2201-07-07 16:50:00 | | 92 | 23 | 95 | 140 | 75 | | 7 |
| 10026255 | 34236274 | 2201-07-07 17:50:00 | 98.3 | 102 | 27 | 96 | 140 | 74 | | 7 |
| 10026255 | 34236274 | 2201-07-07 18:55:00 | 97.7 | 98 | 28 | 98 | 124 | 90 | Sinus Rhythm | 7 |

*Table 10.* **Example Synthetic HMM trajectory (randomly selected).** A single subject's trajectory over 9 timesteps, showing the temporal evolution of the six observed features. The categorical column (Zolronide) transitions between discrete levels, reflecting the latent HMM state dynamics.

| subject_id | timestep | Glomozole | Crirodex | Criphecor | Zolsidex | Zolphephine | Zolronide |
|---|---|---|---|---|---|---|---|
| 0 | 0 | 0.4 | −2.3 | −4.0 | −0.7 | 6.2 | Medium |
| 0 | 1 | 0.6 | −3.3 | −5.0 | −1.4 | 11.8 | Low |
| 0 | 2 | −1.2 | −2.6 | −3.6 | 0.4 | 7.4 | High |
| 0 | 3 | −0.7 | −2.9 | −3.1 | −0.6 | 7.9 | Very High |
| 0 | 4 | 0.2 | −3.8 | −4.1 | −0.6 | 14.8 | Medium |
| 0 | 5 | −1.5 | −2.6 | −4.2 | −0.2 | 7.9 | Medium |
| 0 | 6 | 0.2 | −2.7 | −3.0 | −0.3 | 7.6 | Very High |
| 0 | 7 | −0.7 | −3.2 | −5.4 | −1.7 | 9.3 | Low |
| 0 | 8 | −1.6 | −4.1 | −3.3 | −1.1 | 17.6 | High |

*Table 11.* **Example NYC 311 service request trajectories (randomly selected).** Each row is a service request associated with a specific property (BBL). The data exhibits heterogeneous complaint types, timestamps, and geospatial coordinates, illustrating the complexity of the spatiotemporal schema. Resolution descriptions are abbreviated for space.

| key | created | closed | agency | complaint | descriptor | location_type | zip | address | resolution | bbl | latitude | longitude |
|---|---|---|---|---|---|---|---|---|---|---|---|---|
| 65200110 | 06/08/2025 08:30 PM | 06/08/2025 10:33 PM | NYPD | Noise - Commercial | Loud Music/Party | Store/Commercial | 11225 | 495 FLATBUSH AVE | (action taken) | 3011970006 | 40.66 | −73.96 |
| 65200207 | 06/08/2025 08:30 PM | 06/08/2025 09:37 PM | NYPD | Noise - Residential | Loud Music/Party | Residential | 10458 | 2402 WASHINGTON AVE | (no evidence) | 2030570011 | 40.86 | −73.89 |
| 65202804 | 06/08/2025 08:30 PM | 06/09/2025 02:30 AM | NYPD | Illegal Parking | Sign Violation | Street/Sidewalk | 11377 | 58-21 47 AVENUE | (summons issued) | 4023130047 | 40.74 | −73.91 |
| 65204515 | 06/08/2025 08:29 PM | 06/08/2025 10:08 PM | NYPD | Noise - Residential | Banging/Pounding | Residential | 11237 | 1333 DECATUR ST | (not necessary) | 3034310037 | 40.69 | −73.90 |
| 65201361 | 06/08/2025 08:29 PM | 06/08/2025 08:32 PM | NYPD | Noise - Street | Loud Music/Party | Street/Sidewalk | 10456 | 1465 WASHINGTON AVE | (not necessary) | 2029020036 | 40.84 | −73.90 |

# D. Extended Metrics

### D.1. Univariate Marginal Distance

As a fundamental consistency check, we ensure the global marginal distributions of individual columns are preserved. We do this by aggregating the values of a specific feature $c$ across all tables in the dataset into a single distribution, ignoring the temporal ordering. Let $\mathcal{X}_c = \bigcup_i \{v_{t,c}^{(i)} \mid \mathbf{x}_t^{(i)} \in D^{(i)}\}$ be the multiset of all observed values for feature $c$ in the real dataset, and let $\mathcal{Y}_c$ be the corresponding multiset for the synthetic dataset.

We quantify the distance between these empirical distributions using the Wasserstein-1 distance.

**Definition D.1** (Wasserstein-1 Distance). For probability distributions $\mu$ and $\nu$ on $\mathbb{R}$, the 1-Wasserstein distance is:

$$W_1(\mu, \nu) = \inf_{\gamma \in \Gamma(\mu,\nu)} \int_{\mathbb{R} \times \mathbb{R}} |x - y| \, \mathrm{d}\gamma(x, y) \, , \tag{4}$$

where $\Gamma(\mu, \nu)$ denotes the set of all couplings (joint distributions) with marginals $\mu$ and $\nu$.

In practice, we estimate this by computing the distance between the empirical distributions of $\mathcal{X}_c$ and $\mathcal{Y}_c$. We privilege $W_1$ over metrics like KL-Divergence because it respects the underlying geometry of the metric space (e.g., penalizing a generated heart rate of 180 bpm significantly more than 85 bpm if the true value is 80 bpm, whereas statistical metrics like TVD treat both errors identically).

### D.2. Table-wise Distance to Closest Record (TDCR)

Let the complete collection of real user tables be $\mathbf{D}$, which we split into disjoint training and test sets, $\mathbf{D}_{\text{train}}$ and $\mathbf{D}_{\text{test}}$. Let $\mathbf{D}^*$ be the collection of synthetic tables generated by our model. The first step is to formally define the distance $\Delta(D^{(a)}, D^{(b)})$ between any two tables $D^{(a)}, D^{(b)} \in \mathbf{D} \cup \mathbf{D}^*$.

Recalling our problem formulation, a table $D^{(i)}$ consists of $d$ attributes $\mathcal{A} = \{A_1, \ldots, A_d\}$. We define the trajectory for a specific attribute $A_j$ in table $D^{(i)}$ as the sequence of values $\mathbf{v}_{\cdot,j}^{(i)} = (v_{1,j}^{(i)}, \ldots, v_{T_i,j}^{(i)})$.

The distance $\Delta$ is a weighted sum of per-attribute time series distances, $\delta_j$. For a numerical attribute $A_j$, we define its distance as the normalized Dynamic Time Warping (DTW) distance:

$$\delta_j(\mathbf{v}_{\cdot,j}^{(a)}, \mathbf{v}_{\cdot,j}^{(b)}) = \frac{1}{|K|} \mathrm{DTW}(\mathbf{v}_{\cdot,j}^{(a)}, \mathbf{v}_{\cdot,j}^{(b)}) \tag{5}$$

where $K$ is the optimal warping path found by the DTW algorithm. This normalization accounts for varying sequence lengths $T_a$ and $T_b$. The total inter-table distance is then the weighted $L_1$ norm across all attributes:

$$\Delta(D^{(a)}, D^{(b)}) = \sum_{j=1}^{d} w_j \cdot \delta_j(\mathbf{v}_{\cdot,j}^{(a)}, \mathbf{v}_{\cdot,j}^{(b)}) \tag{6}$$

where weights $w_j$ can be set to balance the contribution of each attribute. In our experiments, we set uniform weights $w_j = 1$.

Next, using this distance function, we compute the TDCR score for each table in our evaluation sets by finding its minimum distance to any table in the training set $\mathbf{D}_{\text{train}}$. For a synthetic table $D^* \in \mathbf{D}^*$, its score is:

$$\mathrm{TDCR}(D^*) = \min_{D \in \mathbf{D}_{\text{train}}} \Delta(D^*, D) \tag{7}$$

And for a real test table $D' \in \mathbf{D}_{\text{test}}$, its benchmark score is:

$$\mathrm{TDCR}(D') = \min_{D \in \mathbf{D}_{\text{train}}} \Delta(D', D) \tag{8}$$

This procedure yields two empirical distributions of minimum distances: $P_{\text{synth}} = \{\mathrm{TDCR}(D^*) \mid D^* \in \mathbf{D}^*\}$ and a benchmark distribution $P_{\text{test}} = \{\mathrm{TDCR}(D') \mid D' \in \mathbf{D}_{\text{test}}\}$.

To quantify the fidelity of the synthetic data, we measure the divergence between these two distributions. Unlike transport-based metrics (e.g., Wasserstein) which focus on magnitude, we prioritize distributional shape. We discretize both $P_{\text{synth}}$ and $P_{\text{test}}$ into $k$ histograms bins over their joint support, yielding discrete probability distributions $P$ and $Q$. The final score is the Jensen-Shannon Distance (JSD), bounded in $[0, 1]$:

$$\text{TDCR}(\mathbf{D}^*, \mathbf{D}) = \sqrt{\frac{D_{KL}(P||M) + D_{KL}(Q||M)}{2}} \tag{9}$$

where $M = \frac{1}{2}(P + Q)$ is the mixture distribution and $D_{KL}$ is the Kullback-Leibler divergence. A lower JSD indicates that the privacy-utility trade-off mechanism has preserved the "distance-to-real" manifold of the held-out data.

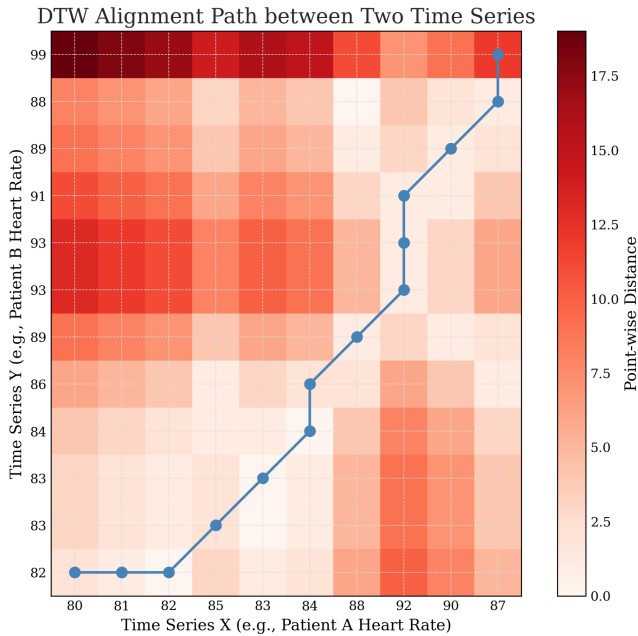

*Figure 7.* Dynamic Time Warping (DTW) alignment between two heart rate time series. The background heatmap represents the point-wise distance cost matrix, where darker red indicates a higher distance. The blue line traces the optimal warping path that minimizes the cumulative cost between Time Series X (Patient A: $x = [80, 81, 82, 85, 83, 84, 88, 92, 90, 87]$) and Time Series Y (Patient B: $y = [82, 83, 83, 84, 86, 89, 93, 93, 91, 89, 88, 99]$). The resulting Total DTW Distance for this example path is 25.0.

### D.3. MAUVE

Standard pairwise similarity approaches assume that comparing the average similarity of each table in set $\mathbf{A}$ directly to its closest neighbor in set $\mathbf{B}$ implicitly captures the closeness of the two families of tables. We adopt an alternative approach that measures the similarity of the distributions over the table embeddings using a divergence measure called MAUVE (Liu et al., 2021a; Pillutla et al., 2021). We utilize the Gecko embedding model (Lee et al., 2024).

**Intuition.** First, we employ a lower-dimensional subspace transformation (via Gecko embeddings) to capture variance while mitigating computational cost and noise. Subsequently, k-means clustering is applied to partition the feature space into discrete clusters, providing a means to quantize the continuous data space into a finite set of representative bins. We use the cluster assignments to construct histograms, which give a discrete representation of the underlying data distributions. To handle data sparsity and zero-probability bins, we apply standard smoothing techniques.

The divergence curve is then generated by assessing the Kullback-Leibler (KL) divergence between the two distributions, and a mixture of both, across a range of mixing parameters. We distill this curve into a single scalar value, the MAUVE score, which corresponds to the area under the divergence curve. A score closer to 1 indicates high similarity. We normalize sample sizes between real and synthetic sets to ensure fair comparison.

**Formal Definition.** Calculating the MAUVE metric requires discretizing the continuous embedding space and measuring the KL-divergence between the two resulting distributions along a curve of their mixtures. Let $\mathcal{P} = \{\mathbf{p}_1, \ldots, \mathbf{p}_m\}$ and

$\mathcal{Q} = \{\mathbf{q}_1, \ldots, \mathbf{q}_n\}$ be the sets of embeddings from the real and synthetic data, respectively, where each embedding vector lives in $\mathbb{R}^d$.

We combine the two sets into a single dataset and apply k-means clustering to partition the embedding space into $k$ discrete clusters. This defines two multinomial probability distributions, $P$ and $Q$, where for each cluster $i \in \{1, \ldots, k\}$, the probabilities $P(i)$ and $Q(i)$ are the fractions of embeddings from $\mathcal{P}$ and $\mathcal{Q}$ assigned to that cluster.

We analyze their relationship by constructing a divergence curve. We define a mixture distribution $R_\lambda = (1 - \lambda)P + \lambda Q$, where $\lambda \in [0, 1]$ is a mixing parameter. The divergence curve is a parametric plot of the KL divergences from the mixture to each original distribution. Specifically, the curve is traced by the points:

$$(D_{KL}(Q||R_\lambda), D_{KL}(P||R_\lambda)) \quad \forall \lambda \in [0, 1] \tag{10}$$

Here, $D_{KL}(P||R_\lambda)$ quantifies Type I error (generating unrealistic samples), while $D_{KL}(Q||R_\lambda)$ quantifies Type II error (mode collapse).

To bound the metric, we map the KL-divergence values to $[0, 1]$ via an exponential scaling function, $f(x) = e^{-c \cdot x}$, where $c$ is a scaling constant. The final MAUVE score is the area under this *scaled* parametric curve.

## D.4. Classifier Discriminator

This metric employs a binary classifier to distinguish between "real" (original) tables and "synthetic" tables generated by the model. If the model produces high-quality synthetic data, the classifier should struggle to tell it apart from the real data (i.e., achieving an accuracy near 0.5 on a balanced set).

We utilize the Gecko embedding strategy described in Section 3, where each table $D^{(i)}$ is converted to a vector $\mathbf{v}_i$. We construct a labeled dataset $\mathcal{D}_{clf} = \{(\mathbf{v}, y) \mid \mathbf{v} \in \mathcal{P} \cup \mathcal{Q}\}$, where $y = 0$ for real tables and $y = 1$ for synthetic tables. We split $\mathcal{D}_{clf}$ into training (70%) and test (30%) sets. We train three classifiers: Logistic Regression, Random Forest, and XGBoost. We report the AUC-ROC on the held-out test set. We additionally inspect the separation of the predicted probability distributions $P(\hat{y} = 1 \mid y = 0)$ and $P(\hat{y} = 1 \mid y = 1)$ using Kernel Density Estimation plots to visualize the margin of separability.

## D.5. State Transition Divergence

To evaluate the dynamics of how a system moves between different states, we discretize continuous features into a finite set of $S$ states (using quantiles derived from the real data). From the sequence of states for all users, we estimate a state transition probability matrix, $\mathbf{M} \in \mathbb{R}^{S \times S}$, where an element $M_{ij}$ represents the empirical probability of transitioning from state $s_i$ to state $s_j$.

Let $\mathbf{M}_R$ and $\mathbf{M}_S$ be the transition matrices estimated from the real and synthetic datasets, respectively. The divergence between them is measured using the Frobenius norm of their difference:

$$D_{\text{trans}} = ||\mathbf{M}_R - \mathbf{M}_S||_F = \sqrt{\sum_{i=1}^{S} \sum_{j=1}^{S} (M_{R,ij} - M_{S,ij})^2} \tag{11}$$

A divergence value closer to zero indicates that the fundamental dynamics of the state changes are well-captured in the synthetic data.

# E. Limits of Flattening, Extended

In this section, we demonstrate the limitations of the flattening approach discussed in Section 4. When longitudinal data is flattened into high-dimensional vectors, the privacy cost of measuring global correlations typically becomes prohibitive, forcing marginal-based mechanisms to restrict their measurements to local neighborhoods (e.g., adjacent time steps). We show below that this restriction imposes a conditional independence structure on the synthetic distribution. Consequently, if the true data contains long-range dependencies, such as distinct user profiles that persist over time, a locally consistent mechanism will inevitably "mix" these profiles, assigning probability mass to spurious trajectories that are locally plausible but globally invalid.

**Definition E.1** ($k$-Local Marginal Consistency). Let $P_\Phi$ be the true distribution over flattened vectors in $\mathcal{Y}$. A synthetic distribution $P^*$ satisfies $k$-*local marginal consistency* with $P_\Phi$ if for all time steps $1 \leq t \leq L - k + 1$, the marginal distribution on contiguous temporal windows matches the source:

$$\mathbb{P}_{P^*}(\mathbf{x}_t, \ldots, \mathbf{x}_{t+k-1}) = \mathbb{P}_{P_\Phi}(\mathbf{x}_t, \ldots, \mathbf{x}_{t+k-1}) \,. \tag{12}$$

Standard marginal-based approaches generate the distribution with Maximum Entropy among those satisfying the measured marginals (McKenna et al., 2024). We show that if a mechanism is restricted to $k$-local marginals (due to budget constraints preventing the measurement of global correlations), it *must* hallucinate invalid data for certain datasets.

**Proposition E.2** (Spurious Trajectories in Local MaxEnt Distributions). *Let $K_{\max} \geq 3$ and $k = 2$. There exists a source distribution supported on a valid set of trajectories $\mathcal{S} \subset \mathcal{Y}$, such that the Maximum Entropy distribution $P^*$ satisfying 2-local marginal consistency assigns non-zero probability to invalid trajectories $\mathbf{y} \notin \mathcal{S}$.*

*Proof.* Let the row domain be $\{\alpha, \beta, \gamma\}$ with fixed trajectory length $L = 3$. We construct the source distribution as an equiprobable mixture of two user types: Type A, defined by $\mathbf{y}_A = (\alpha, \gamma, \alpha)$, and Type B, defined by $\mathbf{y}_B = (\beta, \gamma, \beta)$. The valid support is therefore $\mathcal{S} = \{\mathbf{y}_A, \mathbf{y}_B\}$. Examining the pairwise adjacent marginals, we observe that $\mathbb{P}[\mathbf{x}_1, \mathbf{x}_2]$ is uniform on $\{(\alpha, \gamma), (\beta, \gamma)\}$, implying a deterministic transition, i.e. $\mathbb{P}[\mathbf{x}_2 = \gamma \mid \mathbf{x}_1] = 1$. Similarly, $\mathbb{P}[\mathbf{x}_2, \mathbf{x}_3]$ is uniform on $\{(\gamma, \alpha), (\gamma, \beta)\}$, yielding $\mathbb{P}[\mathbf{x}_3 = \alpha \mid \mathbf{x}_2 = \gamma] = 0.5$. The Maximum Entropy distribution $P^*$ consistent with these local marginals necessarily satisfies the conditional independence $\mathbf{x}_1 \perp \mathbf{x}_3 \mid \mathbf{x}_2$ (i.e. a Markov chain structure). Under this independence, we compute the likelihood of the "mixed" trajectory $\mathbf{z} = (\alpha, \gamma, \beta)$ as,

$$\mathbb{P}_{P^*}[\mathbf{z}] = \mathbb{P}[\mathbf{x}_1 = \alpha] \cdot \mathbb{P}[\mathbf{x}_2 = \gamma \mid \mathbf{x}_1 = \alpha] \cdot \mathbb{P}[\mathbf{x}_3 = \beta \mid \mathbf{x}_2 = \gamma] = 0.5 \cdot 1.0 \cdot 0.5 = 0.25 \,. \tag{13}$$

Since $\mathbf{z} \notin \mathcal{S}$, we conclude that $P^*$ assigns probability mass to spurious records. $\square$

**Remark E.3.** Proposition E.2 highlights that flattening fails to preserve long-range temporal dependencies (e.g., $T_1$ predicting $T_3$) when the synthesizer relies on local dependency structures to save privacy budget. While global marginals *could* capture this, they are often too costly to measure in high dimensions. This motivates an alternate approach, one specifically designed to model sequential data with long-range dependencies.

# F. All Results

We include extensive summary and detail tables here with all our metrics, alongside an example of an NYC 311 set of plots in Section F.1. **Many additional plots are available in the supplement (excluded here for brevity of this document)**.

*Table 12.* **Comprehensive Summary of Metrics on MIMIC-IV.** We report the performance of `PATH` (Gemma 1B/4B) against marginal baselines and non-private reference points across four privacy regimes. Beyond the results already reported in the main body of the paper, we observe a distinct degradation pattern in the marginal baselines: AIM's marginal divergence explodes at $\varepsilon = 0.5$ (18.48), whereas `PATH` remains remarkably stable ($\approx 3.0$), suggesting the LLM's pre-trained priors provide robustness when the privacy signal is weak. Additionally, comparing the "Real (100)" subsample to our synthetic results reveals that for distributional metrics like TDCR, our privately fine-tuned models ($\approx 0.28 - 0.40$) are approaching the natural variance observed between small subsamples of real data ($\approx 0.19$).

| Method | MAUVE (Gecko ↑) | HMM Likelihood (Wass. ↓) | TDCR (Privacy) (JSD ↓) | Marginal Div. (Avg Wass. ↓) | State Trans. Div. (Avg Frobenius ↓) |
|---|---|---|---|---|---|
| AIM (Clipped, $\varepsilon = 0.5$) | 0.587 | - | 0.8326 | 18.4848 | 0.9067 |
| DIRECT (Across, Clipped, $\varepsilon = 0.5$) | 0.608 | - | 0.7937 | 12.8130 | 0.7607 |
| DIRECT (Mark., Clipped, $\varepsilon = 0.5$) | 0.595 | - | 0.8218 | 12.2337 | 0.7299 |
| Gemma 1B ($\varepsilon = 0.5$) | 0.531 | - | 0.3470 | 3.9333 | 0.4819 |
| Gemma 4B ($\varepsilon = 0.5$) | 0.627 | - | 0.3376 | 3.0028 | 0.3817 |
| AIM (Clipped, $\varepsilon = 2.0$) | 0.564 | - | 0.7359 | 9.7080 | 0.7412 |
| DIRECT (Across, Clipped, $\varepsilon = 2.0$) | 0.493 | - | 0.7981 | 8.5031 | 0.6910 |
| DIRECT (Mark., Clipped, $\varepsilon = 2.0$) | 0.573 | - | 0.7960 | 8.5788 | 0.5640 |
| Gemma 1B ($\varepsilon = 2.0$) | 0.596 | - | 0.4315 | 2.9174 | 0.3633 |
| Gemma 4B ($\varepsilon = 2.0$) | 0.661 | - | 0.2886 | 3.2980 | 0.3867 |
| AIM (Clipped, $\varepsilon = 4.0$) | 0.501 | - | 0.7820 | 7.2091 | 0.6189 |
| DIRECT (Across, Clipped, $\varepsilon = 4.0$) | 0.541 | - | 0.7573 | 8.2982 | 0.6722 |
| DIRECT (Mark., Clipped, $\varepsilon = 4.0$) | 0.532 | - | 0.7615 | 8.0687 | 0.5746 |
| Gemma 1B ($\varepsilon = 4.0$) | 0.554 | - | 0.3365 | 3.4431 | 0.3855 |
| Gemma 4B ($\varepsilon = 4.0$) | 0.600 | - | 0.4112 | 4.0971 | 0.3816 |
| AIM (Clipped, $\varepsilon = 10.0$) | 0.531 | - | 0.7349 | 5.7400 | 0.6243 |
| DIRECT (Across, Clipped, $\varepsilon = 10.0$) | 0.497 | - | 0.7665 | 8.3101 | 0.7076 |
| DIRECT (Mark., Clipped, $\varepsilon = 10.0$) | 0.596 | - | 0.7791 | 8.4024 | 0.5497 |
| Gemma 1B ($\varepsilon = 10.0$) | 0.552 | - | 0.3001 | 2.6486 | 0.4454 |
| Gemma 4B ($\varepsilon = 10.0$) | 0.651 | - | 0.3784 | 3.4125 | 0.3747 |
| Gemini 2.5 FL (0-Shot) ($\varepsilon = \infty$) | 0.130 | - | 0.8143 | 14.5522 | 1.2082 |
| Gemini 2.5 FL (1-Shot) ($\varepsilon = \infty$) | 0.265 | - | 0.7858 | 5.4584 | 1.0173 |
| Gemini 2.5 FL (5-Shot) ($\varepsilon = \infty$) | 0.226 | - | 0.5023 | 5.0768 | 1.0841 |
| Gemini 2.5 FL (10-Shot) ($\varepsilon = \infty$) | 0.269 | - | 0.7894 | 5.9262 | 1.2283 |
| REAL (10k) ($\varepsilon = \infty$) | 0.848 | - | 0.2322 | 0.3345 | 0.0161 |
| REAL (1k) ($\varepsilon = \infty$) | 0.866 | - | 0.2202 | 0.7633 | 0.0430 |
| REAL (100) ($\varepsilon = \infty$) | 0.866 | - | 0.1918 | 1.5563 | 0.1332 |

*Table 13.* **Feature-wise Breakdown of Marginal and Temporal Fidelity (MIMIC-IV).** This table decomposes the aggregate scores into per-feature Wasserstein and Frobenius errors. PATH maintains low transition errors across *all* features. Note that while AIM struggles with transition dynamics generally, it is particularly poor at capturing the stability of ordinal features like pain, where PATH (Gemma 4B) reduces the error by over 50%.

| Method | Marginal Distribution (Wasserstein ↓) | | | | | | | State Transition (Frobenius ↓) | | | | | | |
| --- | --- | --- | --- | --- | --- | --- | --- | --- | --- | --- | --- | --- | --- | --- |
| | temp | hear | resp | o2sa | sbp | dbp | pain | temp | hear | resp | o2sa | sbp | dbp | pain |
| AIM (Clipped, $\varepsilon = 0.5$) | 27.187 | 16.368 | 7.145 | 23.720 | 28.687 | 22.992 | 3.294 | 0.971 | 0.918 | 0.693 | 1.261 | 0.855 | 0.919 | 0.730 |
| DIRECT (Across, Clipped, $\varepsilon = 0.5$) | 37.486 | 9.597 | 2.021 | 12.354 | 12.490 | 15.073 | 0.669 | 1.294 | 0.790 | 0.522 | 0.761 | 0.754 | 0.826 | 0.378 |
| DIRECT (Mark., Clipped, $\varepsilon = 0.5$) | 29.550 | 9.710 | 2.753 | 10.727 | 12.956 | 19.438 | 0.501 | 1.151 | 0.616 | 0.611 | 0.696 | 0.561 | 1.114 | 0.361 |
| Gemma 1B ($\varepsilon = 0.5$) | 0.669 | 1.342 | 12.766 | 0.523 | 6.171 | 4.701 | 1.362 | 0.610 | 0.527 | 0.506 | 0.501 | 0.534 | 0.449 | 0.246 |
| Gemma 4B ($\varepsilon = 0.5$) | 0.860 | 3.689 | 1.054 | 0.852 | 7.863 | 4.950 | 1.750 | 0.343 | 0.360 | 0.282 | 0.417 | 0.501 | 0.480 | 0.289 |
| AIM (Clipped, $\varepsilon = 2.0$) | 10.531 | 11.638 | 3.162 | 10.536 | 13.362 | 18.003 | 0.723 | 0.762 | 0.907 | 0.458 | 0.807 | 0.749 | 0.907 | 0.598 |
| DIRECT (Across, Clipped, $\varepsilon = 2.0$) | 23.545 | 6.687 | 1.654 | 7.918 | 10.209 | 8.398 | 1.110 | 0.905 | 0.848 | 0.628 | 0.506 | 0.709 | 0.625 | 0.615 |
| DIRECT (Mark., Clipped, $\varepsilon = 2.0$) | 24.664 | 6.938 | 1.700 | 6.924 | 9.868 | 8.934 | 1.023 | 0.875 | 0.597 | 0.379 | 0.673 | 0.489 | 0.430 | 0.505 |
| Gemma 1B ($\varepsilon = 2.0$) | 0.461 | 5.498 | 1.220 | 0.697 | 8.195 | 3.368 | 0.983 | 0.403 | 0.378 | 0.358 | 0.273 | 0.438 | 0.454 | 0.239 |
| Gemma 4B ($\varepsilon = 2.0$) | 0.976 | 5.987 | 1.102 | 0.821 | 9.248 | 4.143 | 0.810 | 0.414 | 0.368 | 0.362 | 0.405 | 0.457 | 0.453 | 0.247 |
| AIM (Clipped, $\varepsilon = 4.0$) | 9.638 | 7.611 | 2.579 | 6.912 | 10.767 | 12.110 | 0.847 | 0.572 | 0.828 | 0.428 | 0.509 | 0.714 | 0.684 | 0.598 |
| DIRECT (Across, Clipped, $\varepsilon = 4.0$) | 22.738 | 7.300 | 1.796 | 7.488 | 10.187 | 7.386 | 1.192 | 0.898 | 0.837 | 0.482 | 0.538 | 0.705 | 0.595 | 0.650 |
| DIRECT (Mark., Clipped, $\varepsilon = 4.0$) | 22.386 | 6.670 | 1.775 | 6.820 | 10.766 | 7.107 | 0.956 | 0.818 | 0.572 | 0.463 | 0.748 | 0.496 | 0.432 | 0.492 |
| Gemma 1B ($\varepsilon = 4.0$) | 0.574 | 6.669 | 1.125 | 0.926 | 8.634 | 5.065 | 1.109 | 0.432 | 0.369 | 0.386 | 0.445 | 0.461 | 0.419 | 0.187 |
| Gemma 4B ($\varepsilon = 4.0$) | 0.698 | 7.136 | 1.198 | 0.848 | 8.493 | 9.324 | 0.982 | 0.430 | 0.354 | 0.384 | 0.439 | 0.454 | 0.421 | 0.189 |
| AIM (Clipped, $\varepsilon = 10.0$) | 6.255 | 7.275 | 2.019 | 6.217 | 9.583 | 8.037 | 0.793 | 0.447 | 0.814 | 0.425 | 0.798 | 0.703 | 0.586 | 0.597 |
| DIRECT (Across, Clipped, $\varepsilon = 10.0$) | 21.799 | 7.049 | 1.842 | 7.825 | 11.005 | 7.740 | 0.911 | 0.924 | 0.817 | 0.490 | 0.802 | 0.707 | 0.600 | 0.613 |
| DIRECT (Mark., Clipped, $\varepsilon = 10.0$) | 21.276 | 7.196 | 1.849 | 8.859 | 11.124 | 7.533 | 0.979 | 0.814 | 0.560 | 0.373 | 0.713 | 0.483 | 0.405 | 0.499 |
| Gemma 1B ($\varepsilon = 10.0$) | 0.765 | 4.108 | 1.355 | 0.453 | 5.537 | 4.421 | 1.902 | 0.465 | 0.505 | 0.509 | 0.398 | 0.525 | 0.474 | 0.241 |
| Gemma 4B ($\varepsilon = 10.0$) | 0.546 | 5.201 | 1.224 | 1.946 | 9.174 | 4.732 | 1.064 | 0.372 | 0.388 | 0.366 | 0.377 | 0.447 | 0.455 | 0.219 |
| Gemini 2.5 FL (0-Shot) ($\varepsilon = \infty$) | 60.837 | 10.647 | 1.288 | 1.041 | 14.121 | 12.641 | 1.290 | 1.209 | 1.416 | 1.126 | 1.530 | 1.338 | 1.295 | 0.543 |
| Gemini 2.5 FL (1-Shot) ($\varepsilon = \infty$) | 0.966 | 8.754 | 1.385 | 1.198 | 12.342 | 12.133 | 1.432 | 0.911 | 1.224 | 0.869 | 0.904 | 1.219 | 1.143 | 0.851 |
| Gemini 2.5 FL (5-Shot) ($\varepsilon = \infty$) | 0.531 | 8.241 | 1.076 | 1.469 | 12.287 | 10.865 | 1.069 | 1.368 | 0.884 | 1.257 | 0.931 | 1.268 | 1.056 | 0.825 |
| Gemini 2.5 FL (10-Shot) ($\varepsilon = \infty$) | 0.627 | 10.024 | 1.393 | 1.158 | 12.776 | 13.755 | 1.750 | 1.550 | 1.342 | 1.281 | 1.364 | 1.097 | 1.221 | 0.742 |
| REAL (10k) ($\varepsilon = \infty$) | 0.062 | 0.760 | 0.137 | 0.060 | 0.225 | 1.050 | 0.048 | 0.015 | 0.023 | 0.005 | 0.024 | 0.018 | 0.018 | 0.009 |
| REAL (1k) ($\varepsilon = \infty$) | 0.369 | 0.894 | 0.103 | 0.084 | 0.488 | 3.270 | 0.136 | 0.075 | 0.034 | 0.034 | 0.055 | 0.028 | 0.054 | 0.021 |
| REAL (100) ($\varepsilon = \infty$) | 0.205 | 3.349 | 0.416 | 0.315 | 4.874 | 1.283 | 0.452 | 0.124 | 0.174 | 0.117 | 0.179 | 0.167 | 0.129 | 0.042 |

*Table 14.* **Comprehensive Summary of Metrics on Synthetic HMM Data.** This table highlights the utility trade-off between marginal precision and temporal logic. Unsurprisingly, AIM achieves the lowest univariate marginal divergence scores ($\approx 0.25$ at $\varepsilon = 4.0$). However, AIM's HMM Likelihood scores remain prohibitively high ($> 300$), indicating the generation of impossible trajectories. We also observe a significant performance gap between Gemma 1B and 4B here that is wider than in the MIMIC experiments; the 1B model struggles to capture the latent HMM states (Likelihood 261.9 vs. 122.8 at $\varepsilon = 4.0$). This is likely because, for the MIMIC and 311 type data, the 1B model has analogous examples from its training data. As this data is fully synthetic with nonsense column names, processing it relies solely on the models reasoning ability coupled with its ability to generalize a prior over tabular data to unseen/unfamiliar domains.

| Method | MAUVE (Gecko ↑) | HMM Likelihood (Wass. ↓) | TDCR (Privacy) (JSD ↓) | Marginal Div. (Avg Wass. ↓) | State Trans. Div. (Avg Frobenius ↓) |
|---|---|---|---|---|---|
| AIM ($\varepsilon = 0.5$) | 0.727 | 1029.1500 | 0.8326 | 2.0781 | 0.6649 |
| DIRECT (Across, $\varepsilon = 0.5$) | 0.765 | 1455.6246 | 0.8326 | 2.7925 | 0.7234 |
| DIRECT (Mark., $\varepsilon = 0.5$) | 0.705 | 1540.8679 | 0.8268 | 3.0734 | 0.7419 |
| Gemma 4B ($\varepsilon = 0.5$) | 0.237 | 77.3683 | 0.2861 | 1.1648 | 0.4640 |
| AIM ($\varepsilon = 2.0$) | 0.655 | 374.4686 | 0.6352 | 0.4260 | 0.5204 |
| DIRECT (Across, $\varepsilon = 2.0$) | 0.763 | 696.5508 | 0.7384 | 1.5437 | 0.6014 |
| DIRECT (Mark., $\varepsilon = 2.0$) | 0.701 | 754.9578 | 0.8059 | 1.8809 | 0.5953 |
| Gemma 1B ($\varepsilon = 2.0$) | 0.307 | 334.4557 | 0.8073 | 3.0628 | 1.0329 |
| Gemma 4B ($\varepsilon = 2.0$) | 0.612 | 57.9726 | 0.4150 | 0.9494 | 0.4363 |
| AIM ($\varepsilon = 4.0$) | 0.804 | 321.6139 | 0.6262 | 0.2532 | 0.4994 |
| DIRECT (Across, $\varepsilon = 4.0$) | 0.802 | 740.3704 | 0.7305 | 0.9890 | 0.5525 |
| DIRECT (Mark., $\varepsilon = 4.0$) | 0.728 | 944.7643 | 0.7624 | 1.3724 | 0.5306 |
| Gemma 1B ($\varepsilon = 4.0$) | 0.268 | 261.9771 | 0.6806 | 2.6619 | 0.7937 |
| Gemma 4B ($\varepsilon = 4.0$) | 0.660 | 122.8850 | 0.3880 | 1.0726 | 0.4670 |
| AIM ($\varepsilon = 10.0$) | 0.705 | 319.4292 | 0.5367 | 0.2757 | 0.4791 |
| DIRECT (Across, $\varepsilon = 10.0$) | 0.704 | 409.1063 | 0.6560 | 0.6496 | 0.5331 |
| DIRECT (Mark., $\varepsilon = 10.0$) | 0.710 | 653.2374 | 0.6748 | 0.9022 | 0.4558 |
| Gemma 1B ($\varepsilon = 10.0$) | 0.359 | 184.9042 | 0.5402 | 2.5169 | 0.7138 |
| Gemma 4B ($\varepsilon = 10.0$) | 0.690 | 50.9899 | 0.3966 | 0.8250 | 0.4217 |
| Gemini 2.5 FL (0-Shot) ($\varepsilon = \infty$) | 0.270 | 318770.3493 | 0.8326 | 53.5924 | 1.1601 |
| Gemini 2.5 FL (1-Shot) ($\varepsilon = \infty$) | 0.262 | 41.8997 | 0.7800 | 0.9118 | 0.7483 |
| Gemini 2.5 FL (5-Shot) ($\varepsilon = \infty$) | 0.255 | 96.9679 | 0.6787 | 1.0024 | 0.7310 |
| Gemini 2.5 FL (10-Shot) ($\varepsilon = \infty$) | 0.268 | 9.0193 | 0.5564 | 0.5596 | 0.6595 |
| REAL (10k) ($\varepsilon = \infty$) | 0.856 | 0.3254 | 0.1778 | 0.0325 | 0.0096 |
| REAL (1k) ($\varepsilon = \infty$) | 0.892 | 0.8304 | 0.1618 | 0.0517 | 0.0316 |
| REAL (100) ($\varepsilon = \infty$) | 0.899 | 1.5302 | 0.2069 | 0.1468 | 0.0882 |

*Table 15.* **Feature-wise Breakdown on Synthetic Data.** Detailed analysis reveals specific modes of failure for the baselines. For instance, marginal baselines have spiky errors across features; some columns were or were not selected for measurement, and thus the privacy budget is unevenly spread (e.g., DIRECT (Mark.) reaches 0.996 at $\epsilon = 0.5$, but is then clearly measured at $\epsilon = 10$), while PATH (Gemma 4B) has reasonable results across the board.

| Method | Marginal Distribution (Wasserstein ↓) | | | | | State Transition (Frobenius ↓) | | | | |
|---|---|---|---|---|---|---|---|---|---|---|
| | Glom | Crir | Crip | Zols | Zolp | Glom | Crir | Crip | Zols | Zolp |
| AIM ($\varepsilon = 0.5$) | 1.325 | 1.213 | 1.401 | 1.443 | 5.009 | 0.725 | 0.662 | 0.895 | 0.725 | 0.318 |
| DIRECT (Across, $\varepsilon = 0.5$) | 0.691 | 0.736 | 0.863 | 0.927 | 10.747 | 0.640 | 0.611 | 0.827 | 0.638 | 0.901 |
| DIRECT (Mark., $\varepsilon = 0.5$) | 0.742 | 0.736 | 0.915 | 1.042 | 11.932 | 0.632 | 0.596 | 0.820 | 0.665 | 0.996 |
| Gemma 4B ($\varepsilon = 0.5$) | 0.848 | 0.494 | 2.593 | 0.592 | 1.298 | 0.586 | 0.531 | 0.426 | 0.325 | 0.451 |
| AIM ($\varepsilon = 2.0$) | 0.365 | 0.424 | 0.374 | 0.271 | 0.696 | 0.600 | 0.534 | 0.776 | 0.531 | 0.161 |
| DIRECT (Across, $\varepsilon = 2.0$) | 0.295 | 0.399 | 0.549 | 0.416 | 6.060 | 0.612 | 0.541 | 0.790 | 0.566 | 0.499 |
| DIRECT (Mark., $\varepsilon = 2.0$) | 0.484 | 0.472 | 0.580 | 0.553 | 7.316 | 0.583 | 0.505 | 0.740 | 0.533 | 0.616 |
| Gemma 1B ($\varepsilon = 2.0$) | 3.597 | 4.102 | 3.812 | 1.373 | 2.429 | 1.301 | 1.438 | 1.064 | 0.557 | 0.804 |
| Gemma 4B ($\varepsilon = 2.0$) | 0.913 | 0.762 | 0.996 | 0.570 | 1.505 | 0.507 | 0.447 | 0.498 | 0.259 | 0.471 |
| AIM ($\varepsilon = 4.0$) | 0.261 | 0.298 | 0.179 | 0.147 | 0.382 | 0.612 | 0.530 | 0.698 | 0.504 | 0.153 |
| DIRECT (Across, $\varepsilon = 4.0$) | 0.253 | 0.354 | 0.392 | 0.236 | 3.709 | 0.607 | 0.533 | 0.779 | 0.546 | 0.297 |
| DIRECT (Mark., $\varepsilon = 4.0$) | 0.378 | 0.417 | 0.463 | 0.350 | 5.254 | 0.548 | 0.481 | 0.700 | 0.481 | 0.443 |
| Gemma 1B ($\varepsilon = 4.0$) | 2.899 | 3.629 | 3.238 | 0.846 | 2.698 | 0.568 | 0.878 | 1.213 | 0.577 | 0.732 |
| Gemma 4B ($\varepsilon = 4.0$) | 0.668 | 1.176 | 0.729 | 0.855 | 1.935 | 0.511 | 0.399 | 0.588 | 0.287 | 0.550 |
| AIM ($\varepsilon = 10.0$) | 0.247 | 0.327 | 0.198 | 0.175 | 0.432 | 0.609 | 0.533 | 0.614 | 0.483 | 0.157 |
| DIRECT (Across, $\varepsilon = 10.0$) | 0.212 | 0.326 | 0.261 | 0.246 | 2.203 | 0.603 | 0.532 | 0.765 | 0.543 | 0.221 |
| DIRECT (Mark., $\varepsilon = 10.0$) | 0.375 | 0.415 | 0.370 | 0.255 | 3.096 | 0.495 | 0.441 | 0.628 | 0.428 | 0.286 |
| Gemma 1B ($\varepsilon = 10.0$) | 2.763 | 3.119 | 3.166 | 0.610 | 2.926 | 0.623 | 0.724 | 0.941 | 0.566 | 0.715 |
| Gemma 4B ($\varepsilon = 10.0$) | 0.576 | 0.961 | 0.660 | 0.686 | 1.242 | 0.528 | 0.388 | 0.518 | 0.284 | 0.390 |
| Gemini 2.5 FL (0-Shot) ($\varepsilon = \infty$) | 186.225 | 56.715 | 16.987 | 5.976 | 2.059 | 1.178 | 0.996 | 1.142 | 1.130 | 1.354 |
| Gemini 2.5 FL (1-Shot) ($\varepsilon = \infty$) | 0.257 | 0.494 | 0.314 | 0.639 | 2.855 | 0.518 | 0.652 | 0.596 | 0.914 | 1.061 |
| Gemini 2.5 FL (5-Shot) ($\varepsilon = \infty$) | 1.150 | 0.736 | 0.445 | 0.649 | 2.033 | 0.942 | 0.663 | 0.598 | 0.576 | 0.875 |
| Gemini 2.5 FL (10-Shot) ($\varepsilon = \infty$) | 0.526 | 0.303 | 0.444 | 0.284 | 1.241 | 0.641 | 0.488 | 0.532 | 0.725 | 0.912 |
| REAL (10k) ($\varepsilon = \infty$) | 0.028 | 0.017 | 0.044 | 0.025 | 0.049 | 0.010 | 0.007 | 0.011 | 0.012 | 0.008 |
| REAL (1k) ($\varepsilon = \infty$) | 0.063 | 0.064 | 0.061 | 0.020 | 0.050 | 0.031 | 0.035 | 0.031 | 0.031 | 0.030 |
| REAL (100) ($\varepsilon = \infty$) | 0.062 | 0.087 | 0.221 | 0.148 | 0.216 | 0.094 | 0.089 | 0.081 | 0.080 | 0.096 |

*Table 16.* **Summary Metrics for NYC 311 Service Requests.** Unlike the tabular MIMIC dataset, the NYC 311 data contains complex, heterogeneous schema elements (geospatial coordinates, categorical descriptors). Here, the disparity between private fine-tuning and non-private prompting is most pronounced: Gemini 2.5 FL (even with 10-shot) has extremely poor MAUVE scores. PATH (Gemma 4B) demonstrates strong scaling behavior, improving MAUVE from 0.375 to 0.634 as the budget increases from $\varepsilon = 0.5$ to $\varepsilon = 10.0$. The high Classifier AUC scores across all methods ($\approx 0.95$) show how real 311 requests are hard to perfectly simulate.

| Method | MAUVE (Gecko ↑) | Classifier AUC (Ideal 0.5) | Temporal Dist. (Wass. ↓) | Transition Div. (Frobenius ↓) |
|---|---|---|---|---|
| Gemma 1B ($\varepsilon = 0.5$) | 0.410 | 0.982 | 1.922 | 0.357 |
| Gemma 4B ($\varepsilon = 0.5$) | 0.375 | 0.952 | 1.327 | 0.230 |
| Gemma 1B ($\varepsilon = 2.0$) | 0.353 | 0.967 | 1.487 | 0.239 |
| Gemma 4B ($\varepsilon = 2.0$) | 0.573 | 0.940 | 0.823 | 0.408 |
| Gemma 1B ($\varepsilon = 4.0$) | 0.500 | 0.951 | 0.566 | 0.240 |
| Gemma 4B ($\varepsilon = 4.0$) | 0.566 | 0.972 | 1.020 | 0.295 |
| Gemma 1B ($\varepsilon = 10.0$) | 0.464 | 0.971 | 0.870 | 0.279 |
| Gemma 4B ($\varepsilon = 10.0$) | 0.634 | 0.948 | 0.870 | 0.405 |
| Gemini 2.5 FL (0-Shot) ($\epsilon = \infty$) | 0.040 | 1.000 | 5.468 | - |
| Gemini 2.5 FL (1-Shot) ($\epsilon = \infty$) | 0.062 | 1.000 | 4.275 | 1.927 |
| Gemini 2.5 FL (5-Shot) ($\epsilon = \infty$) | 0.056 | 1.000 | 2.389 | - |
| Gemini 2.5 FL (10-Shot) ($\epsilon = \infty$) | 0.066 | 1.000 | 2.131 | - |
| REAL (10k) ($\epsilon = \infty$) | 0.876 | 0.470 | 0.312 | 0.165 |
| REAL (1k) ($\epsilon = \infty$) | 0.863 | 0.529 | 0.343 | 0.087 |
| REAL (100) ($\epsilon = \infty$) | 0.850 | 0.699 | 0.881 | 0.379 |

### F.1. Detailed Analysis: NYC 311 (Gemma 4B, $\varepsilon = 10.0$)

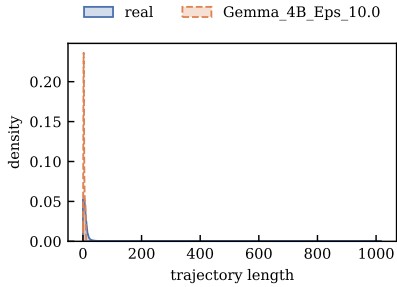

*(a)* **Trajectory Lengths:** Distribution of total service requests per user history.

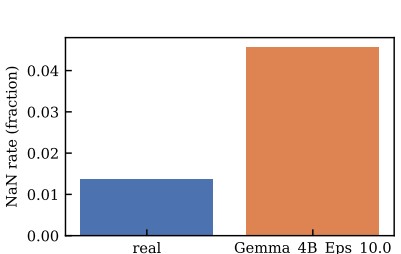

*(b)* **Missingness:** Comparison of 'NaN' value proportions across schema columns.

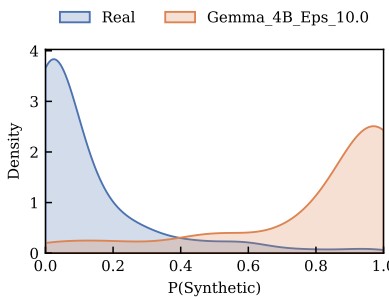

*(c)* **Indistinguishability:** Classifier separation score (AUC) between Real and Synthetic embeddings.

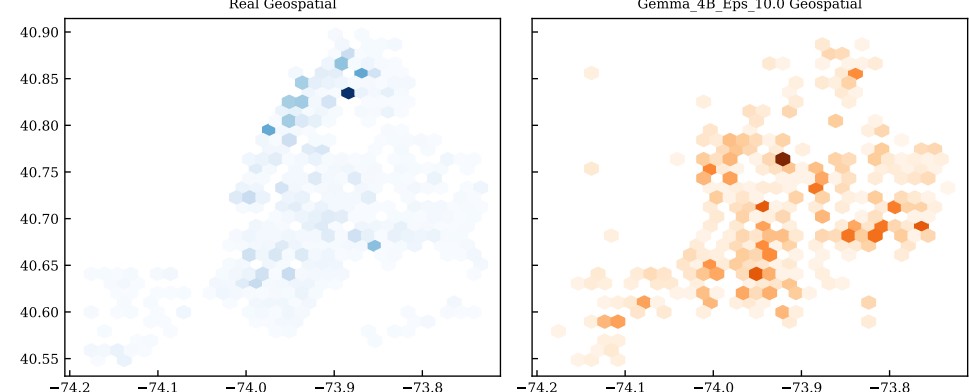

*(d)* **Geospatial Fidelity:** Hexbin density maps comparing the spatial distribution of 311 calls. The synthetic data (right) accurately reconstructs the complex topology of the NYC boroughs visible in the real data (left), preserving high-density clusters in Manhattan and Brooklyn.

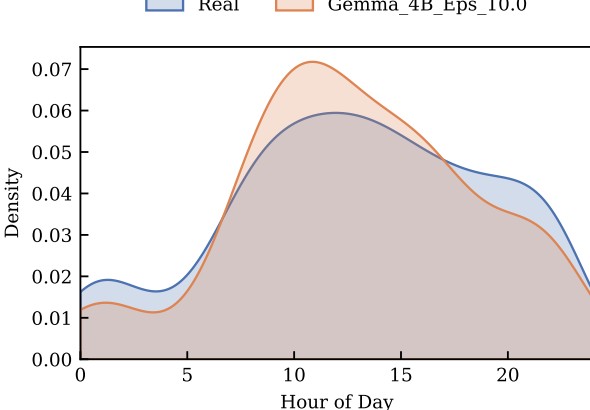

*(e)* **Temporal Patterns:** Distribution of service requests by Hour of Day, showing preservation of diurnal rhythms.

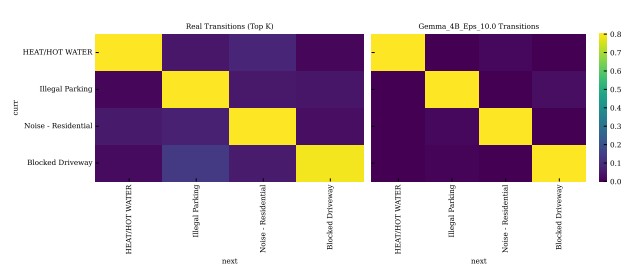

*(f)* **Complaint Dynamics:** State transition matrix visualizing the conditional probability of the next complaint type given the previous one.

*Figure 8.* **Qualitative Evaluation on NYC 311 Data (Gemma 4B, $\varepsilon = 10.0$).** The privately fine-tuned model successfully captures multi-modal distributions, including long-tail trajectory lengths (a), complex geospatial densities (d), and daily temporal seasonality (e).

## F.2. Privacy as Regularization

Counter-intuitively, our evaluation of the Synthetic dataset reveals that a larger privacy budget (higher $\varepsilon$) does not necessarily guarantee better distributional matching when measured by e.g. TDCR. The PATH (Gemma 4B) model achieves its lowest (best) Jensen-Shannon Distance (JSD) for the Table-wise Distance to Closest Record (TDCR) metric at the strictest privacy setting of $\varepsilon = 0.5$, scoring a JSD of 0.2861. As the privacy budget is relaxed to $\varepsilon = 2.0$ and $\varepsilon = 10.0$, the error surprisingly increases to 0.4150 and 0.3966, respectively. This non-monotonic behavior is confusing.

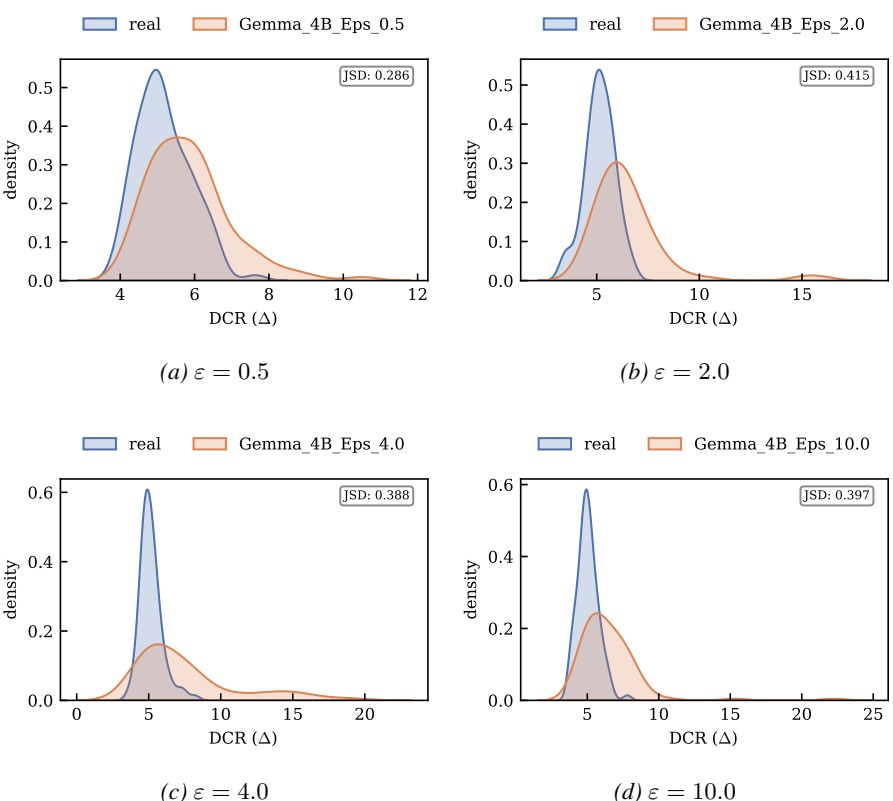

*Figure 9.* **Evolution of TDCR Density with Privacy Budget.** At low $\varepsilon$ (0.5), the distribution of distances is tight and well-aligned with the baseline. As $\varepsilon$ increases (decreasing noise), the model begins to overfit to outliers, creating a heavy tail of generated tables that are topologically distant from the typical data manifold.

We hypothesize that in the high-$\varepsilon$ (low noise) regime, the model retains sufficient capacity to memorize or overfit to outliers and out-of-distribution examples present in the training set. This results in the generation of a "long tail" of synthetic records that drift significantly from the typical data manifold. Additionally, the private selection step may contribute to this non-monotonic behavior: at higher total $\varepsilon$, a larger $\varepsilon_{\text{select}}$ allows the selection mechanism to more precisely filter the over-generated pool, but this precision may paradoxically favor outlier-like tables that happen to be nearest neighbors of outlier training points, thereby amplifying rather than dampening the overfitting effect.

This phenomenon is visually evident in Figure 9. At $\varepsilon = 0.5$ (Figure 9a), the distribution of nearest-neighbor distances is tight; the x-axis extends only to approximately 12, and the curve closely overlaps the reference baseline. However, as we relax the budget to $\varepsilon = 2.0$ (Figure 9b), a tail begins to emerge, extending the support to 15. By $\varepsilon = 10.0$ (Figure 9d), the distribution exhibits a long tail.

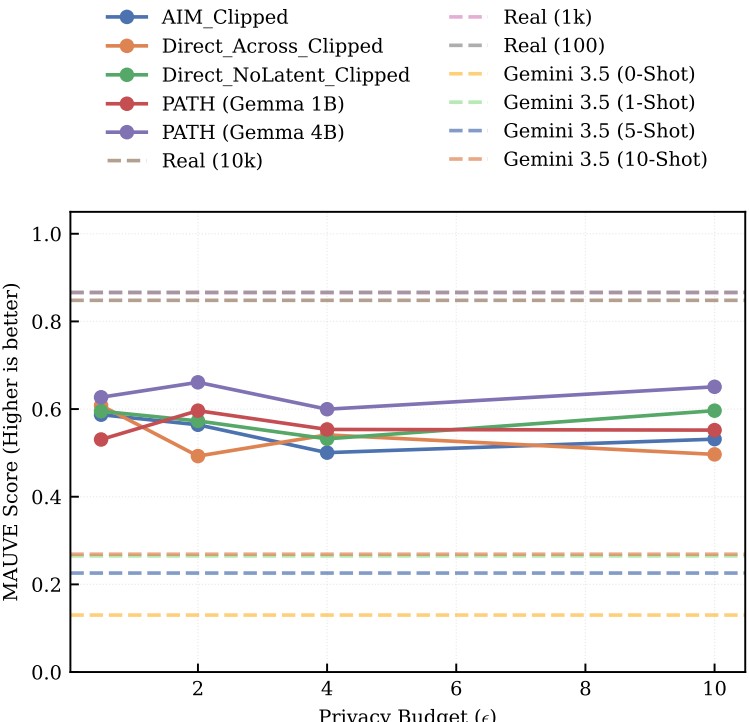

*Figure 10.* **Distributional fidelity on MIMIC-IV.** The Gemma 4B model (top blue line) consistently achieves the highest MAUVE scores across all privacy budgets, peaking at 0.690 for $\varepsilon = 10$. Notably, private fine-tuning significantly outperforms the non-private Gemini 2.5 FL baselines (dashed horizontal lines), which fail to exceed 0.30 even with 10-shot prompting. This illustrates that parameter-efficient fine-tuning is far more effective than in-context learning for capturing the complex manifold of clinical trajectories.

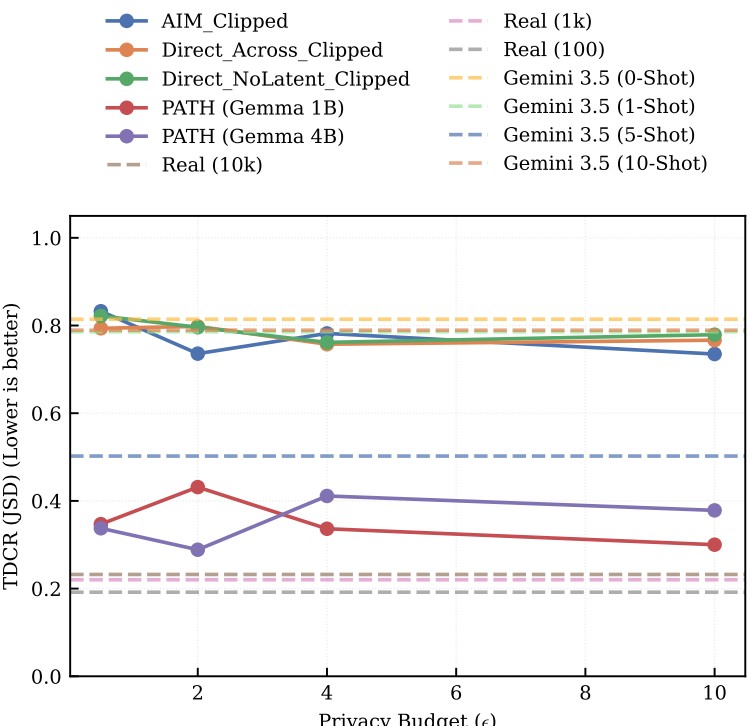

*Figure 11.* **Distance to closest record analysis.** A lower TDCR score indicates generated tables are structurally similar to real user trajectories without being identical. The marginal-based baselines (AIM and Direct) exhibit high error rates ($> 14.0$) that degrade sharply as $\varepsilon$ decreases, reflecting the failure of "flattening" strategies to model user-level coherence. In contrast, Gemma 4B maintains a low, stable distance ($\approx 3.9$ to $7.1$) even at $\varepsilon = 0.5$, demonstrating superior robustness to noise.

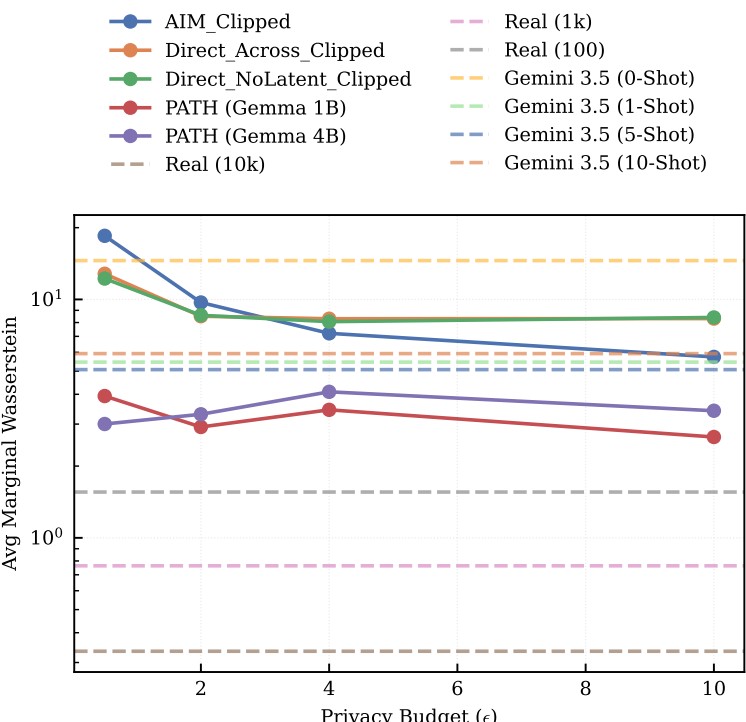

*Figure 12.* **Preservation of univariate marginals.** Despite being an autoregressive model, Gemma 4B (bottom lines) surprisingly outperforms marginal-based mechanisms on column-wise fidelity, maintaining Wasserstein distances below 4.0 across all budgets. The flattened marginal baselines struggle here, likely due to the high dimensionality introduced by padding all trajectories to a fixed length, which dilutes the privacy budget available for individual column measurements.

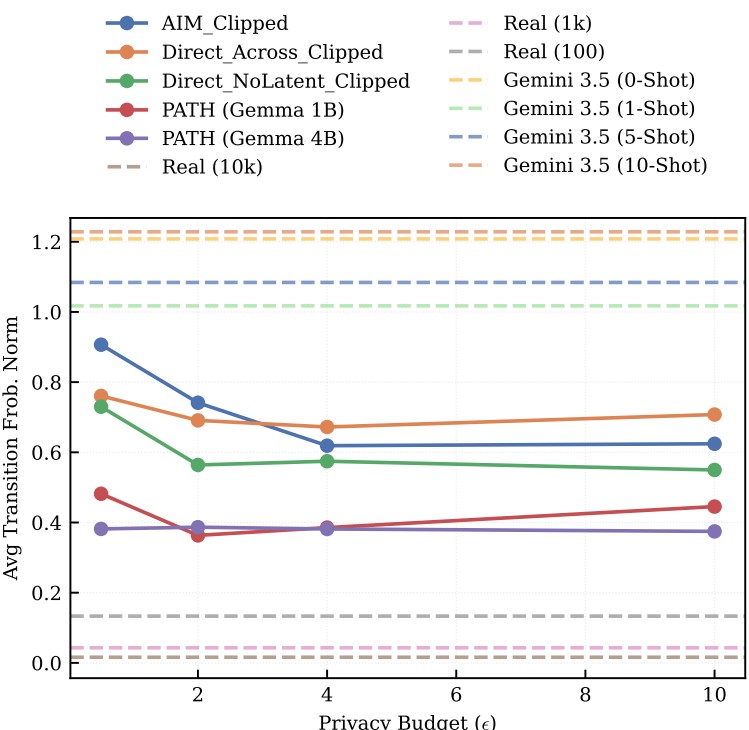

*Figure 13.* **Temporal dynamics preservation.** This metric measures the error in the conditional probability of state transitions (e.g., $P(\text{vital}_t \mid \text{vital}_{t-1})$). The Gemma family (1B and 4B) consistently yields lower error rates ($\approx 0.38 - 0.48$) compared to AIM and Direct mechanisms ($> 0.60$). This confirms that the LLM's autoregressive objective naturally aligns with the task of preserving sequential dependencies, whereas marginal methods struggle to capture these time-variant correlations.

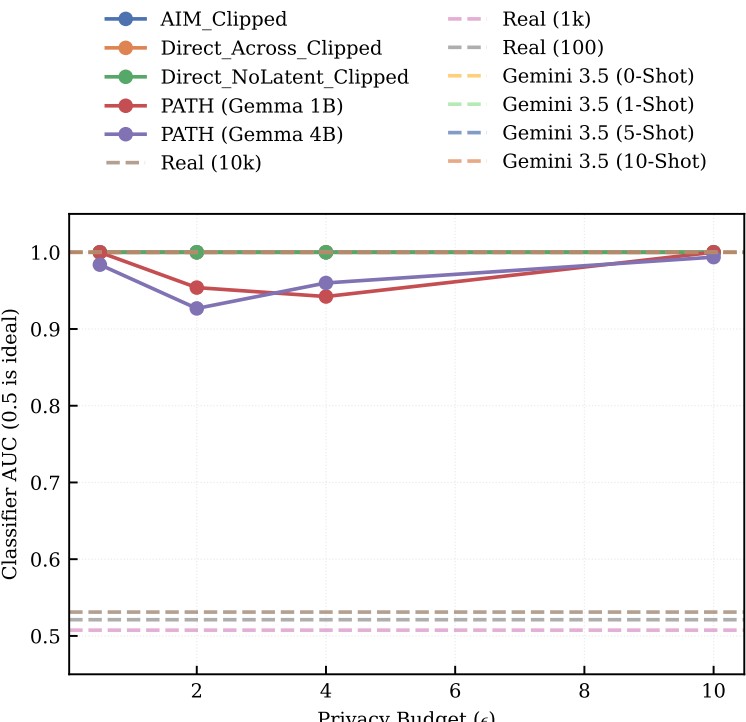

*Figure 14.* **Discriminability of synthetic data.** While most methods yield an AUC near 1.0 (indicating the synthetic data is distinguishable from real data), the Gemma 1B model shows a slight improvement in indistinguishability at lower privacy budgets (AUC $\approx 0.95$ at $\varepsilon = 2.0$). However, the generally high AUC scores across all methods suggest that while the generated trajectories are statistically useful, they remain distinct enough for a classifier to separate from the original training data.

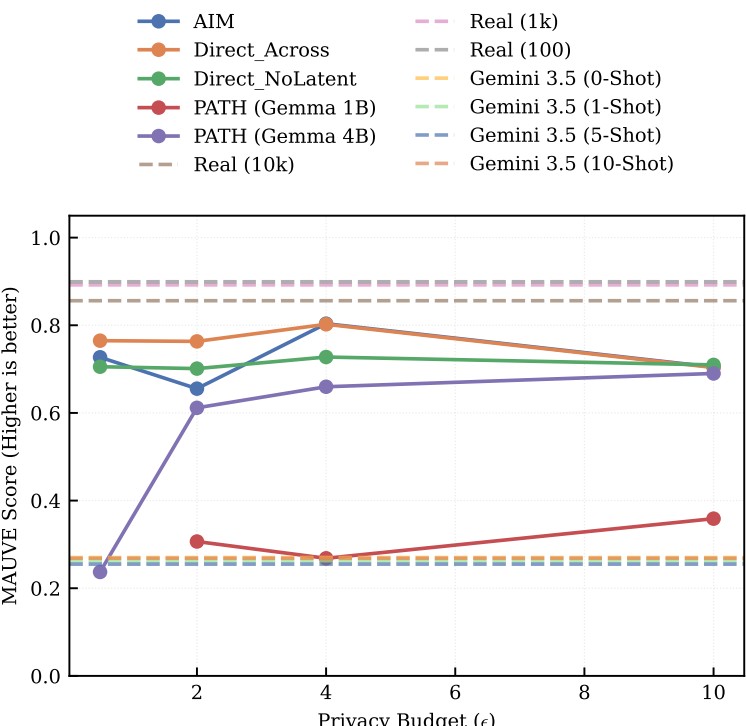

*Figure 15.* **Distributional fidelity on Synthetic data.** In this controlled setting, the marginal-based AIM algorithm (top lines) outperforms the LLM-based approaches, achieving MAUVE scores between 0.72 and 0.79. This reversal suggests that for simpler, strictly structured data where the manifold is well-defined by independent components, marginal measurements can be more efficient than learning the distribution via next-token prediction.

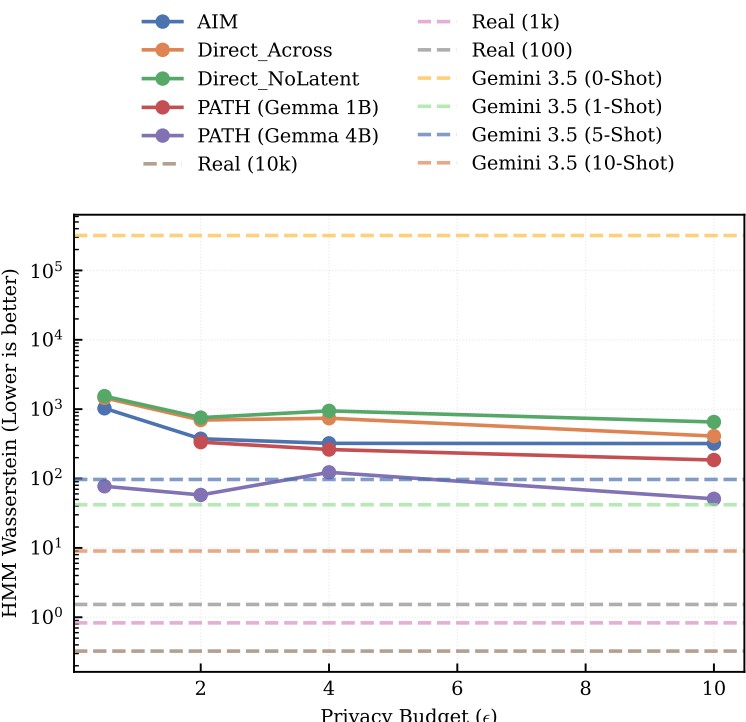

*Figure 16.* **Likelihood under true latent process.** While AIM captures the static manifold well (see MAUVE), it fails to capture the temporal logic. Gemma 4B achieves significantly lower Wasserstein distances to the true HMM likelihoods ($\approx 52 - 122$) compared to AIM ($\approx 300 - 1000$). This highlights the critical trade-off: marginal methods preserve independent statistics, but autoregressive models are required to preserve the validity of the sequence under the generative process.

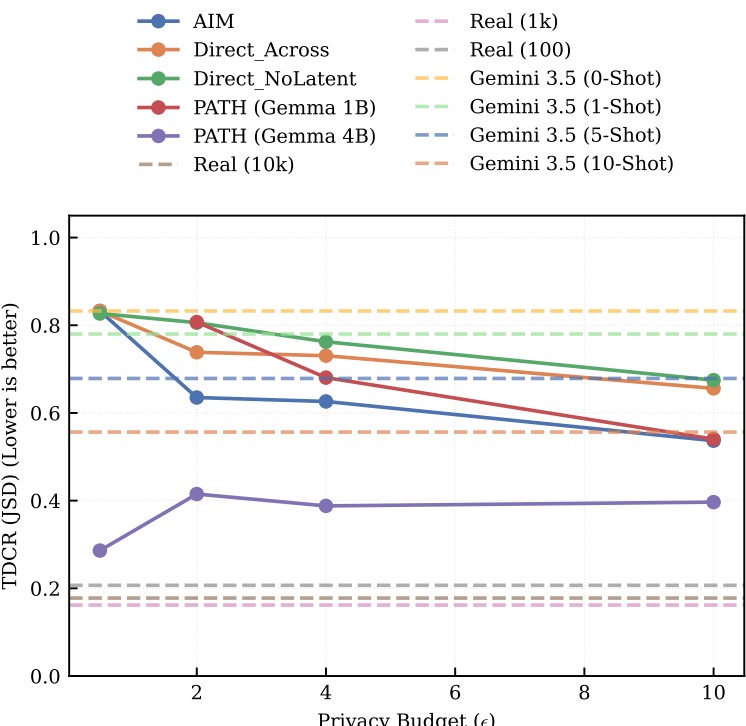

*Figure 17.* **Structural fidelity on Synthetic data.** Gemma 4B outperforms all baselines with the lowest TDCR scores, particularly at $\varepsilon = 10$ (0.686), indicating it generates trajectories that sit closest to the support of the real data. Notably, the non-private Gemini 2.5 FL 0-shot baseline exhibits a massive error (72.5), further proving that without specific fine-tuning or demonstrations, foundation models cannot zero-shot complex tabular distributions.

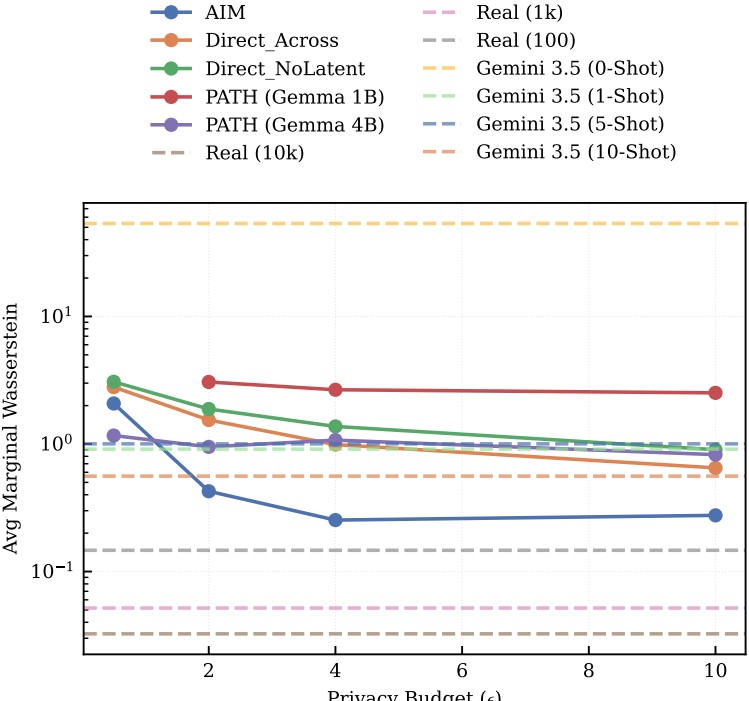

*Figure 18.* **Column-wise accuracy.** Consistent with the MAUVE results for this dataset, AIM achieves the lowest Marginal Distance ($\approx 0.23 - 0.26$ for $\varepsilon \geq 4$), beating Gemma 4B ($\approx 0.8 - 1.1$). This confirms that when the data generation process is simple and columns have strong independent signals, marginal-based mechanisms utilize the privacy budget more efficiently for univariate statistics.

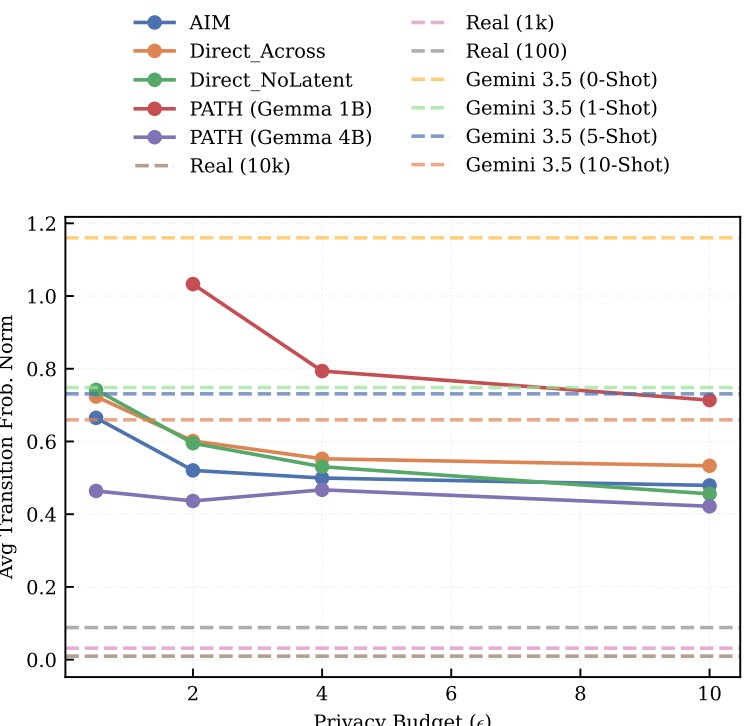

*Figure 19.* **Transition matrix recovery.** The Gemma 4B model achieves the lowest error in recovering the transition matrix (≈ 0.42 − 0.47), slightly outperforming AIM. This reinforces the finding that even when marginal methods excel at static distributions (as seen in the Marginal Distance plot), they are less capable of capturing the derivative, time-dependent structure of the data.

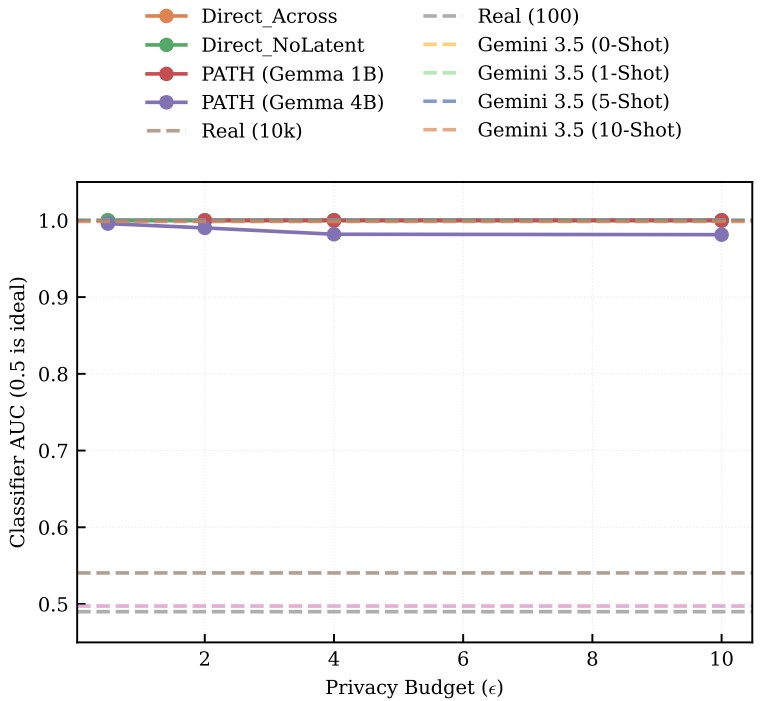

*Figure 20.* **Discriminability of Synthetic data.**

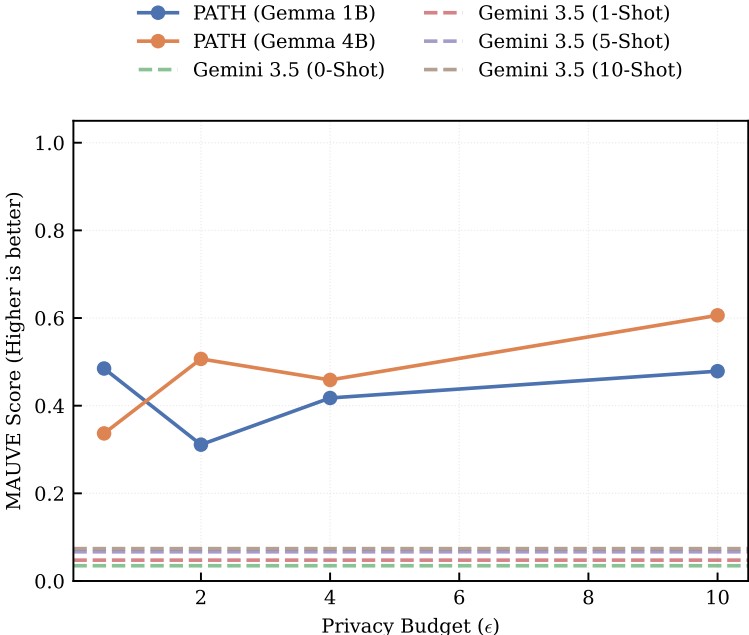

*Figure 21.* **Fidelity on spatiotemporal data.** The private Gemma models demonstrate a clear advantage on this complex, heterogeneous dataset. Gemma 4B ($\varepsilon = 2.0$) achieves a MAUVE score of $0.546$, vastly outperforming the non-private Gemini 2.5 FL few-shot baselines, which plateau below $0.10$. This indicates that for rich, semantic data like service requests, private fine-tuning is essential for learning the underlying distribution, whereas prompting is insufficient.

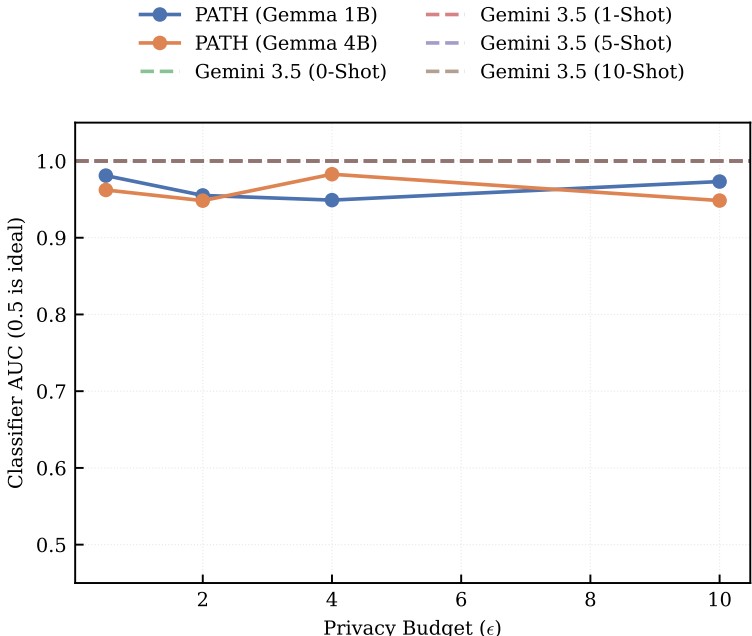

*Figure 22.* **Discriminability on NYC 311.** Similar to the MIMIC results, the synthetic data remains distinguishable from real data (AUC $> 0.95$). However, the Gemma 1B model shows slightly better resistance to classification than the 4B model at lower epsilons ($0.94$ at $\varepsilon = 4.0$), suggesting a slight regularization effect in the smaller model that prevents it from overfitting to identifiable artifacts.

# G. Additional Baselines and Ablations

We conducted several additional experiments to further characterize PATH's performance, including comparisons with sequential DP baselines, non-private fine-tuning, downstream task evaluations, and ablations on key design choices.

## G.1. Comparison with DP-DoppelGANger

To compare against a sequential DP baseline beyond flattened marginal methods, we evaluated a DP-enhanced version of DoppelGANger (Lin et al., 2020) on the MIMIC dataset at $\varepsilon = 2.0$. We strengthened the original implementation by incorporating DP-MERF (Harder et al., 2021) (kernel mean matching via random Fourier features with Gaussian mechanism) and tuned hyperparameters non-privately to give DoppelGANger the best possible chance.

*Table 17.* **Comparison with DP-DoppelGANger on MIMIC ($\varepsilon = 2.0$).** PATH substantially outperforms DoppelGANger ($\sim 22\times$ on marginals, $\sim 3\times$ on transitions), consistent with prior findings that GAN-based methods struggle under DP constraints on structured tabular data (Chen et al., 2025).

| Method | Marginal Div. (Avg Wass. ↓) | State Trans. (Avg Frob. ↓) | TDCR (JSD ↓) |
|---|---|---|---|
| PATH (Gemma 1B) | **2.85** | **0.38** | **0.45** |
| DP-DoppelGANger | 64.17 | 1.20 | 0.83 |

As shown in Table 17, PATH substantially outperforms DoppelGANger across all metrics. The GAN-based training is sensitive to high-dimensional, mixed-type tabular data under DP constraints; several MIMIC vital-sign columns showed limited diversity (mode collapse) despite non-private hyperparameter tuning.

## G.2. Computational Cost

*Table 18.* **Computational cost comparison on MIMIC (single NVIDIA RTX PRO 6000).** PATH training is approximately $10\times$ slower than marginal methods, which is expected given LLM fine-tuning overhead. Generation is slower per table but is parallelizable across GPUs.

| Method | Training Time | Generation Time |
|---|---|---|
| DP-DoppelGANger | $\sim 10$ min | $\sim$ms / table |
| AIM / Direct | $\sim 6$ min | $\sim$ms / table |
| PATH (Gemma 1B) | $\sim 62$ min | $\sim$seconds / table |

Table 18 summarizes the training and generation costs for all methods on the MIMIC dataset. PATH training is approximately $10\times$ slower than marginal methods, which is expected given the overhead of LLM fine-tuning with DP-SGD. However, training is a one-time cost, and the quality gains demonstrated throughout our experiments justify the additional compute. Generation with PATH takes seconds per table (compared to milliseconds for marginal methods) but is highly parallelizable across GPUs.

## G.3. Non-Private Baseline and LoRA Rank Ablation

Table 19 compares DP-trained PATH against non-private fine-tuning across LoRA ranks $r \in \{32, 64, 128, 256\}$. Interestingly, DP-trained PATH with private selection matches or outperforms non-DP without selection on marginal fidelity (e.g., $r = 128$: DP Marg. = 2.93 vs. Non-DP Marg. = 3.74). This suggests the private selection step acts as an effective quality filter over the over-generated pool, compensating for DP noise. Transition divergence is comparable across DP and non-DP settings, indicating that DP noise primarily affects marginal accuracy rather than temporal structure. Under DP, higher LoRA rank improves marginals ($r = 256$: 2.82 vs. $r = 32$: 3.89), whereas rank matters less without DP.

## G.4. Choice of LLM Backbone

To confirm that PATH is model-agnostic, we evaluated two additional LLM backbones: Llama 3.2 1B and Qwen 3.5 0.8B (Table 20). Gemma is the most consistent performer overall; however, Qwen achieves the best single result at $r = 128$

*Table 19.* **Non-private vs. DP fine-tuning across LoRA ranks (Gemma 1B, MIMIC).** Non-DP uses standard fine-tuning (no noise, no clipping). DP runs use $\varepsilon_{train} = 1.5$, $\varepsilon_{select} = 0.5$. DP-trained `PATH` with private selection matches or outperforms non-DP without selection on marginal fidelity, demonstrating that the selection step compensates for DP noise.

| | **Non-DP** | | **DP** ($\varepsilon = 2.0$) | |
|---|---|---|---|---|
| $r$ | **Marg.** (Avg Wass. ↓) | **Trans.** (Avg Frob. ↓) | **Marg.** (Avg Wass. ↓) | **Trans.** (Avg Frob. ↓) |
| 256 | 3.96 | 0.40 | **2.82** | 0.41 |
| 128 | 3.74 | 0.41 | 2.93 | 0.43 |
| 64 | 3.62 | **0.40** | 3.78 | 0.41 |
| 32 | 3.79 | 0.42 | 3.89 | 0.42 |

*Table 20.* **Choice of LLM backbone (MIMIC, $\varepsilon = 2.0$).** Gemma is most consistent; Qwen achieves the best single result at $r = 128$. `PATH` works across all three model families, confirming the framework is model-agnostic.

| **Model** | **Marg.** ($r = 128$) (Avg Wass. ↓) | **Marg.** ($r = 256$) (Avg Wass. ↓) |
|---|---|---|
| Gemma 3 1B | 2.93 | **2.82** |
| Llama 3.2 1B | 3.61 | 3.73 |
| Qwen 3.5 0.8B | **2.70** | 3.20 |

(Marg. $= 2.70$). All three model families produce competitive results, confirming that `PATH`'s effectiveness stems from the framework design rather than a specific model architecture.

### G.5. Privacy Budget Splitting

*Table 21.* **Privacy budget splitting ablation (Gemma 1B, $r = 128$, MIMIC, $\varepsilon_{total} = 2.0$).** Marginals are stable across splits, but temporal metrics differ substantially: the default $(1.5, 0.5)$ achieves the best temporal fidelity.

| $(\varepsilon_{train}, \varepsilon_{sel})$ | **Marg.** (Avg Wass. ↓) | **Trans.** (Avg Frob. ↓) | **TDCR** (JSD ↓) |
|---|---|---|---|
| (0.5, 1.5) | 2.92 | 1.13 | 0.55 |
| (1.0, 1.0) | 2.85 | 1.12 | 0.52 |
| **(1.5, 0.5)** | **2.93** | **0.43** | **0.36** |
| (2.0, −) | 2.85 | 1.13 | 0.50 |

Table 21 ablates the allocation of the total privacy budget $\varepsilon = 2.0$ between training and private selection on MIMIC with Gemma 1B at $r = 128$. Marginal fidelity is stable across splits, but temporal metrics differ substantially. The default allocation $(\varepsilon_{train}, \varepsilon_{sel}) = (1.5, 0.5)$ achieves the best temporal fidelity (Trans. $= 0.43$, TDCR $= 0.36$), outperforming both the no-selection baseline $(2.0, −)$ and allocations that devote more budget to selection. The model needs sufficient training budget to learn temporal dynamics, after which even a modest $\varepsilon_{sel} = 0.5$ provides effective filtering.

### G.6. Downstream Task Evaluation (TSTR)

To validate that distributional fidelity translates to practical downstream utility, we conducted Train-on-Synthetic, Test-on-Real (TSTR) evaluations on both MIMIC and NYC 311.

For MIMIC (Table 22), we trained an XGBoost classifier to predict whether a patient's emergency department stay exceeds 8 hours, using 20 temporal vital-sign features. `PATH` achieves positive lift above the majority baseline (AUC 0.57 vs. 0.50), while AIM fails to exceed chance (AUC 0.47).

For NYC 311 (Table 23), we trained an XGBoost classifier to predict the next complaint type (10 classes) using temporally-focused features. `PATH` achieves 53.8% accuracy, close to the real-data accuracy of 58.5% and far above the majority baseline of 32.1%. AIM was excluded as it cannot scale to the 311 schema due to dimensionality constraints.

*Table 22.* **TSTR: MIMIC ED stay prediction** ($\varepsilon = 2.0$, $n = 1000$)**.** We predict whether a patient's ED stay exceeds 8 hours using 20 temporal vital-sign features with XGBoost. PATH shows positive lift above the majority baseline (AUC 0.57 vs. 0.50), while AIM fails to exceed chance (AUC 0.47).

| Training Data | Accuracy | AUC |
|---|---|---|
| Real | 0.84 | 0.89 |
| PATH (Gemma 1B) | 0.55 | 0.57 |
| AIM | 0.46 | 0.47 |
| Majority Baseline | 0.52 | 0.50 |

*Table 23.* **TSTR: NYC 311 next complaint type prediction** ($\varepsilon = 10.0$, $n = 1000$)**.** PATH achieves near real-data accuracy (53.8% vs. 58.5%), far above the majority baseline (32.1%). AIM cannot scale to 311 due to dimensionality constraints.

| Training Data | Accuracy | Weighted F1 |
|---|---|---|
| Real | 0.585 | 0.578 |
| PATH (Gemma 4B) | 0.538 | 0.518 |
| Majority Baseline | 0.321 | – |

### G.7. Classifier AUC Discussion

The Classifier AUC scores approach 1.0 across all methods (see Appendix Tables and Figures), indicating that a strong classifier can distinguish synthetic from real data. We verified (via extensive automated and manual inspection) that formatting, decimal precision, rounding behavior, and serialization conventions are identical between real and synthetic tables, so the classifier cannot distinguish them based on surface-level formatting artifacts. The high AUC therefore reflects distributional differences in data content introduced by DP noise.

Importantly, our TSTR results (Tables 22–23) provide a direct test of whether these distinguishable features corrupt task-relevant signal. On NYC 311, PATH achieves near real-data accuracy despite the high classifier AUC, suggesting the distinguishable features do not harm downstream utility. On MIMIC, while all methods struggle on this inherently difficult task, PATH still shows positive lift while AIM does not.

# H. LLM Zero/Few Shot Prompts

**Zero-Shot Prompt Structure**

Consider the following domain over patient records (the column names):

<column_name_1>, <column_name_2>, ..., <column_name_n>

Can you now generate a table for another patient, in the same domain as this data?  Please give me a csv surrounded by ````csv' and ````' tags.

**Few-Shot Prompt Structure (for $k$ examples)**

Consider the following set of vital sign time series records, each a single table representing an individual patient (denoted by 'subject_id'), where each row is a chart time and set of vitalsign features (with some missingness):

Example Patient 1:
``<CSV String for Patient 1>''

Example Patient 2:
``<CSV String for Patient 2>''

...

Example Patient $k$:
``<CSV String for Patient $k$>''

Can you now generate a table for another patient, in the same domain as this data?  Please give me a csv surrounded by ````csv' and ````' tags.

# I. Gradient Signal-to-Noise Analysis in High Dimensions

During the fine-tuning of our LLMs with DP-SGD, we observed an interesting phenomenon. Specifically, the $L_2$-norm of the noise vector $\mathbf{z}$ added to the gradients typically exceeded the norm of the true clipped gradient signal $\bar{\mathbf{g}}$ by several orders of magnitude. Under a standard signal-to-noise Ratio (SNR) interpretation based on raw magnitudes ($\|\bar{\mathbf{g}}\|_2/\|\mathbf{z}\|_2$), successful learning appeared unlikely.

However, recent work by Li et al. (2022) suggests that $L_2$-based SNR is a fundamentally misleading metric for understanding learning dynamics in high-dimensional spaces (in our case, $d \approx 210$ million trainable parameters for e.g. the 4 billion parameter Gemma model with LoRA). In this section, we present an empirical analysis performed during our experimentation to validate that learning is driven by the *alignment* of the noisy update with a low-dimensional signal subspace, rather than the raw magnitude of the noise.

**Orthogonality and Alignment** Let's build some intuition. Let $\bar{\mathbf{g}}$ denote the aggregated, clipped gradient computed over a batch, and let $\mathbf{z} \sim \mathcal{N}(0, \sigma^2 \mathbf{I})$ be the noise vector. The update vector is $\tilde{\mathbf{g}} = \bar{\mathbf{g}} + \mathbf{z}$.

In high-dimensional spaces, isotropic Gaussian noise is approximately orthogonal to any fixed low-dimensional subspace with high probability (Livan et al., 2018). Assuming the true learning signal lies within a low-dimensional subspace $\mathcal{S} \subset \mathbb{R}^d$, the component of the noisy update that drives learning is its projection onto $\bar{\mathbf{g}}$. The alignment can be characterized by the dot product, as,

$$\tilde{\mathbf{g}} \cdot \bar{\mathbf{g}} = (\bar{\mathbf{g}} + \mathbf{z}) \cdot \bar{\mathbf{g}} = \|\bar{\mathbf{g}}\|_2^2 + \mathbf{z} \cdot \bar{\mathbf{g}} \,. \tag{14}$$

Since $\mathbb{E}[\mathbf{z}] = \mathbf{0}$, we have $\mathbb{E}[\mathbf{z} \cdot \bar{\mathbf{g}}] = 0$. Consequently, in expectation, there is always a positive component of the update in the direction of the signal, provided the optimizer can accumulate this weak signal over many steps without the noise variance causing divergence.

**Some Empirical Metrics** To verify this hypothesis, we consider two metrics during training: expected cosine similarity and the participation ratio.

We measure the alignment between the clean signal $\bar{\mathbf{g}}$ and the noisy gradient $\tilde{\mathbf{g}}$. While the direct cosine similarity is non-linear, we approximate its expectation as,

$$\mathbb{E}\left[\text{sim}(\bar{\mathbf{g}}, \tilde{\mathbf{g}})\right] \approx \frac{\|\bar{\mathbf{g}}\|_2}{\mathbb{E}[\|\tilde{\mathbf{g}}\|_2]} \,. \tag{15}$$

Empirically, we found that while this value is small (on the order of $10^{-4}$ for $\sigma = 0.4$), it remains consistently positive, confirming that the gradient updates maintain a stable, albeit weak, directional alignment with the loss landscape.

To validate the assumption that the gradient signal is effectively low-rank (sparse in some basis), we compute the participation ratio (PR) (Livan et al., 2018) of the gradient updates, or,

$$PR(\bar{\mathbf{g}}) = \frac{\left(\sum_{j=1}^d \bar{g}_j^2\right)^2}{\sum_{j=1}^d \bar{g}_j^4} = \frac{\|\bar{\mathbf{g}}\|_2^4}{\|\bar{\mathbf{g}}\|_4^4} \,. \tag{16}$$

The PR provides a continuous measure of the effective number of active dimensions, ranging from 1 (maximal localization) to $d$ (maximal delocalization).

**Observations** We compared training runs with a standard noise multiplier ($\sigma = 0.4$, corresponding to $\varepsilon \approx 10$) against ablations with high noise ($\sigma = 10.0$).

As shown in Figure 24, we observed $PR(\bar{\mathbf{g}}) \ll d$ (typically orders of magnitude lower than the parameter count $2.1 \times 10^8$), providing strong evidence that the gradients occupy a low-dimensional manifold. This sparsity allows DP-SGD to navigate the optimization landscape despite the overwhelming ambient noise.

Conversely, at $\sigma = 10.0$, we reach a critical threshold $\sigma_{\text{crit}}$ where the utility bound becomes vacuous, and the alignment (Figure 23b) becomes dominated by stochastic variance, preventing effective learning. These findings validate the theoretical bounds proposed by Song et al. (2021); Li et al. (2022) and justify our choice of hyperparameters.

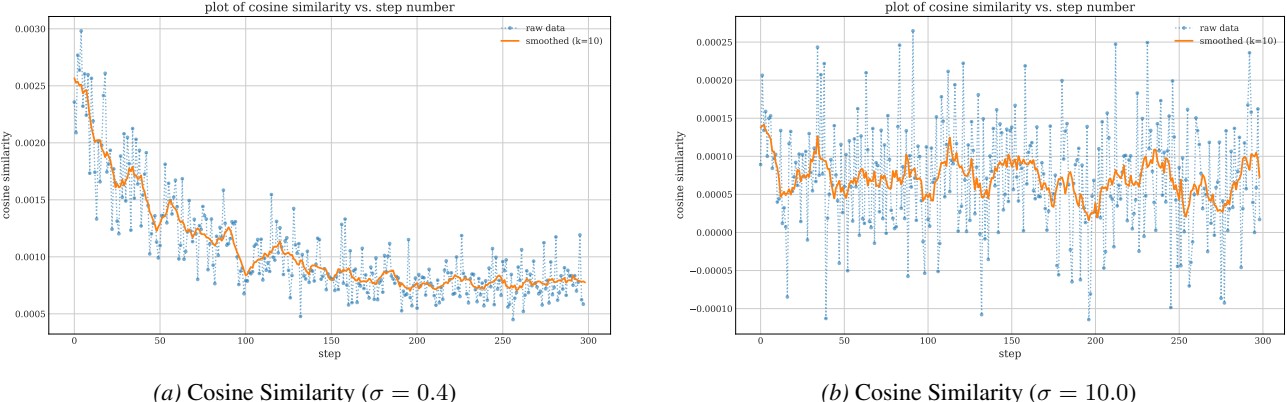

*(a)* Cosine Similarity ($\sigma = 0.4$)     *(b)* Cosine Similarity ($\sigma = 10.0$)

*Figure 23.* Cosine similarity between the signal and the noisy gradient. At operational noise levels (Left), a consistent positive alignment is observed. At critical noise levels (Right), alignment degrades significantly.

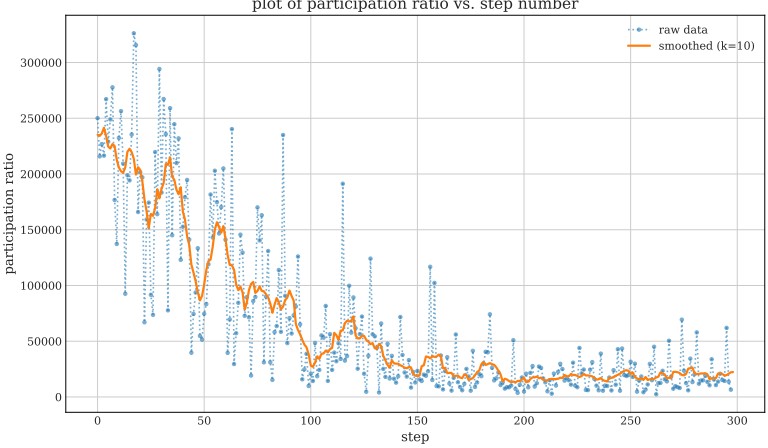

*Figure 24.* The Participation Ratio (PR) of the gradients (log scale) during training. The PR remains orders of magnitude below the ambient dimension $d$, indicating the effective gradients are confined to a low-dimensional subspace.

