# OpenReview forum: "Privately Fine-Tuned LLMs Preserve Temporal Dynamics in Tabular Data"
_ICML.cc/2026/Conference — ICML 2026 regular_

### Official Review · Reviewer_h9Ag · 2026-02-27

**Soundness:** 3
**Presentation:** 4
**Significance:** 3
**Originality:** 3
**Overall Recommendation:** 5
**Confidence:** 4

**Summary:**

This paper addresses the problem of generating differentially private synthetic longitudinal tabular data, where each user owns a multi-row temporal trajectory (e.g., ICU vital signs over time). The authors argue that traditional DP synthesis methods (e.g., AIM, Private-PGM), which assume i.i.d. rows, fail when applied to flattened longitudinal data because they can only capture low-order local marginals, resulting in trajectories that are "locally plausible but globally invalid." The paper proposes PATH (Private Autoregressive Trajectory Histories), which: (1) redefines the privacy unit as the full user table rather than a single row, (2) serializes each user's table into a text sequence and fine-tunes Gemma 3 (1B/4B) with DP-SGD + LoRA, and (3) introduces a dynamic windowing training strategy and private post-hoc selection mechanism. The paper also proposes TDCR, a DTW-based metric for evaluating temporal fidelity at the table level. Evaluations on MIMIC-IV, NYC 311, and a synthetic HMM dataset show substantial improvements in temporal coherence metrics while maintaining competitive marginal fidelity.

**Compliance With Llm Reviewing Policy:**

Affirmed.

**Final Justification:**

The rebuttal has adequately addressed the questions and concerns raised in my review. I appreciate the authors' efforts in clarifying these points. I maintain my original score of 5.

**Key Questions For Authors:**

Questions for Authors

Q1. Can you provide TSTR (train-on-synthetic, test-on-real) results for at least one downstream task? For MIMIC-IV, mortality prediction is a natural choice. For NYC 311, predicting the next complaint type or response time would demonstrate the practical value of temporal fidelity.

Q2. The "privacy as regularization" observation (Appendix F.2) — where TDCR is best at ε=0.5 on HMM data — is intriguing. You hypothesize overfitting to outliers at higher ε. Have you investigated this further? Specifically, does the private selection step (which has a larger εselect at higher ε_total) contribute to this non-monotonic behavior by selecting more outlier-like tables?

Q3. How sensitive is the performance to the LoRA rank (r=128)?

Q4. At ε_total=0.5, the selection step consumes εselect=0.25, leaving only εtrain=0.25 for model training. What is the performance without private selection (i.e., allocating the full budget to training)? This ablation would clarify whether private selection is worth its privacy cost.

Q5. All appendix figures (including MIMIC) label the non-private baseline as "Gemini 3.5," while appendix tables for MIMIC/HMM use "Gemini 2.5 FL," and the NYC 311 table also uses "Gemini 3.5." Can you confirm these all refer to the same model (Gemini 2.5 Flash-Lite), and if so, please unify the naming?

**Limitations:**

The Impact Statement is reasonable, it identifies relevant domains and acknowledges residual risks of DP synthetic data. However, the paper lacks a dedicated Limitations section. Key limitations that should be discussed include:

(1) no downstream task evaluation (TSTR) to validate practical utility,

(2) classifier AUC ≈ 1.0 indicating synthetic data remains easily distinguishable,

(3) dependence on a single LLM family (Gemma 3)

**Strengths And Weaknesses:**

Strengths

S1. Clean problem formalization with a genuinely underexplored angle. The paper identifies a real gap in the DP synthetic data literature: the assumption that each row corresponds to one user breaks down for longitudinal data. The formalization of the "table as privacy unit" (Definition in Section 2), the flattening transformation Φ (Definition 4.1), and the distinction between intra-row and inter-row correlations are clear and well-motivated. The paper correctly positions this as extending the suggestion by Ponomareva et al. (2025) to a concrete application.

S2. Proposition E.2 provides rigorous theoretical support for the key claim. The proof that Maximum Entropy distributions satisfying only 2-local marginal consistency must assign probability to invalid trajectories (for a simple 3-step, 3-state system) is elegant and directly supports the central argument. While the example is simple, it is precisely the right level of abstraction to convey the fundamental limitation of flattening + marginal methods.

S3. Comprehensive and well-designed evaluation framework. The evaluation is a notable strength:

Three datasets spanning medical (MIMIC-IV), civic (NYC 311), and controlled (HMM) settings
The HMM dataset provides ground-truth transition probabilities for rigorous temporal evaluation
NYC 311 post-dates Gemma 3's training cutoff (Aug 2024), controlling for memorization
The real data subsampling baselines (100/1K/10K tables) provide essential context for interpreting distributional metrics
Multiple privacy regimes (ε ∈ {0.5, 2.0, 4.0, 10.0}) tested systematically
The TDCR metric is a well-motivated contribution: DTW handles variable-length tables, and JSD between distance distributions detects both mode collapse and poor coverage
S4. Strong and consistent empirical results on temporal metrics. PATH achieves substantial improvements on temporal fidelity:

MIMIC ε=2.0: TDCR JSD 0.289 vs AIM's 0.736 (61% reduction)
HMM ε=10.0: HMM log-likelihood Wasserstein 50.99 vs AIM's 319.43 (84% reduction)
State transition divergence consistently lower across all datasets and ε values
These improvements are large enough to be meaningful even without error bars
S5. Surprising and informative finding: private fine-tuning outperforms non-private few-shot. The result that PATH (Gemma 4B, ε=2.0) achieves MAUVE 0.661 vs Gemini 2.5 FL (10-shot, ε=∞) at 0.269 on MIMIC is counterintuitive and valuable. The explanation, that fine-tuning internalizes distributional information into weights while ICL is limited by context, is convincing and has broader implications for the LLM-for-data community.

S6. Thoughtful baseline design. The authors deliberately strengthen their baselines:

The Direct (Markov) mechanism is hand-crafted with domain knowledge to measure exactly the right temporal marginals
Marginal baselines are given an advantage by clipping to [min, 99th percentile] (acknowledged as technically violating DP)
AIM is given the flattened data and allowed to automatically select marginals
Despite these advantages, PATH still dominates on temporal metrics
S7. Excellent transparency and reproducibility. The privacy reporting follows the standardized format suggested by Ponomareva et al. (2023) (Appendix A), the hyperparameters are fully specified with explicit LoRA rank and clipping norm, and the appendix includes per-feature breakdowns (Table 9), prompt templates (Appendix G), and a gradient signal-to-noise analysis (Appendix H).

Weaknesses

W1. No downstream task evaluation, the practical utility of the synthetic data is not demonstrated.

The paper evaluates distributional fidelity (MAUVE, TDCR, marginal divergence, state transitions) but never demonstrates that the synthetic data is useful for the primary purpose of DPSD: enabling analytics and model training without accessing real data. Standard practice in the DPSD literature includes "train-on-synthetic, test-on-real" (TSTR) experiments, e.g., training a mortality prediction model on synthetic MIMIC data and evaluating on real held-out data. Without this, we cannot assess whether the temporal fidelity improvements translate to better downstream utility. It is possible that the synthetic data captures trajectory shapes well but fails to preserve the statistical relationships needed for prediction tasks.

W2. No error bars or variance estimates reported across any experiments.

All results appear to be from single runs. Given that:

1. DP-SGD training has inherent randomness from gradient noise

2. The dynamic windowing samples different split points per epoch

3. Private selection involves noisy voting

4. LLM generation is stochastic (temperature-dependent)

the absence of confidence intervals or standard deviations across multiple seeds is a significant gap. Some of the improvements (e.g., MIMIC ε=10.0: PATH 4B MAUVE 0.651 vs Direct (Indep) 0.596) could plausibly be within the variance of a single run.

W3. Classifier AUC near 1.0 reveals a fundamental quality gap that is underemphasized.

On MIMIC, the Classifier AUC for PATH is not reported in the main tables. On NYC 311 (Table 3), PATH achieves AUC ≈ 0.95 (ideal is 0.5), meaning classifiers can almost perfectly distinguish synthetic from real data. The appendix (Figure 14) shows MIMIC classifier AUC is also near 1.0 for all methods. This means the synthetic data, while preserving temporal dynamics, remains statistically distinguishable from real data at the distributional level. The paper acknowledges this in passing but does not adequately discuss the implications: if the data is easily distinguishable, downstream models trained on it may learn synthetic artifacts rather than true patterns.

W4. Pervasive Gemini model version naming inconsistency.

The main text (Section 5.1) and appendix tables for MIMIC/HMM consistently use "Gemini 2.5 FL" (Flash-Lite). However, all appendix figures (Figures 10–22, including MIMIC plots) and the NYC 311 table (Table 12) use "Gemini 3.5", a model name that does not correspond to any known Google release. On the same page (p.37), the figure legend reads "Gemini 3.5" while the caption text reads "Gemini 2.5 FL." This is almost certainly a labeling bug in the plotting code rather than different models being used, but it reflects insufficient proofreading and could confuse readers about whether results are comparable across datasets.

---

> ### Author Rebuttal · Authors · 2026-03-31
>
> Thank you for your time and thorough review, and for your overall positivity about our work, recognizing our paper’s strengths and listing them in detail: the problem formalization (S1), Proposition E.2 (S2), HMM ground-truth (S3), outperforming non-private few-shot (S5), baseline design (S6), and privacy reporting (S7).
>
> > **W1/Q1: TSTR.**
>
> We agree that downstream use is very important and have now run TSTR evaluations on both datasets!
>
> For **MIMIC** vitalsign data, we ran an emergency department (ED) >= 8 hour stay classification ($n=1000$, ε=2): predicting ED stay length is a common task in the medical literature; prior work with richer features gets AUC 0.65-0.75 [Ala et al. 2023; Rocheteau et al. 2021] on MIMIC. Prediction from vital signs alone is difficult, but still demonstrates PATH's improvement over other DP methods. We use 20 temporal vital-sign features (first/last values, deltas, slopes, clinical flags) with XGBoost.
>
> |Train|Acc|AUC|
> |:--|--:|--:|
> |Real |0.84|0.89|
> |PATH |0.55|0.57|
> |AIM |0.46|0.47|
> |Majority|0.52|0.50|
>
> PATH shows positive lift above naive baseline (AUC 0.57 vs 0.50 guessing majority), while AIM performs poorly (AUC 0.47); AIM's marginal-based approach likely does not sufficiently preserve the temporal dynamics needed for this task.
>
> For **NYC 311**, we ran next-complaint-type prediction (10 classes) with temporally-focused features (lag-3 complaint types, gap statistics, time-of-day) and XGBoost (ε=10, $n=1000$). Random guessing yields ~10%.
>
> |Train|Acc|Weighted-F1|
> |:--|--:|--:|
> |Real|.585|.578|
> |PATH|.538|.518|
> |Majority|.321|n/a|
>
> PATH achieves near real-data accuracy (53.8% vs 58.5%), far above majority (32.1%). AIM cannot scale to 311 due to dimensionality constraints.
>
> >**W2: Error bars.**
>
> We agree, some of the granular results that are close together could be within the variance of a single run. Many of our key improvements, however, are quite large, making it very unlikely these gaps fall within single-run variance. Additionally, our ablations across many configurations (multiple LLMs, multiple LoRa ranks, different tasks, etc.) all show consistent strong PATH, softly checking single run variance issues. That said, we have launched our full original experimental configuration on each of the datasets at each epsilon level with 5 seeds for PATH on Gemma 1B, such that we can add error bars to the final version of our paper.
>
> > **W3: Classifier AUC.**
>
> We appreciate this concern; all our methods were similarly distinguishable, although PATH was the least so (see Appendix Figure 14). We note that we verified (via extensive automated and manual inspection) that formatting, decimal precision, and serialization are identical between real and synthetic tables. W.r.t. your concern about whether this impacts downstream utility, that question is well-taken. Our TSTR results provide a direct test: on 311, PATH achieves close to real-data accuracy despite the high classifier AUC, suggesting the distinguishable features do not corrupt task-relevant signal here. On MIMIC, the task is harder and all methods struggle; however, PATH still shows positive lift while AIM does not. We will expand this discussion in the revision.
>
> > **W4/Q5: Gemini naming.**
>
> All results use Gemini 2.5 Flash-Lite; "Gemini 3.5" was a plotting code bug, now corrected.
>
> > **Q2: "Privacy as regularization."**
>
> Insightful question! Our ablations showed that DP-trained models with private selection can outperform non-DP models without selection (Gemma r=256: DP Marg.=2.82 vs Non-DP Marg.=3.96; see Reviewer 5GAy Q1). This suggests the selection step acts as a quality filter that compensates for DP noise and may contribute to the non-monotonic TDCR behavior at low ε.
>
> > **Q3: LoRA rank sensitivity.**
>
> We ablated r $\in {32, 64, 128, 256}$ on MIMIC with Gemma 1B, Llama 1B, and Qwen 0.8B (see Reviewer 1NTX W3 for the cross-model table). Key findings: (1) higher r improves marginals under DP; (2) transition divergence is robust across r=64-256; (3) Gemma is strongest overall, but all three model families produce competitive results, confirming PATH is model-agnostic.
>
> > **Q4: Performance w/o priv. selection.**
>
> This is another great question. As the performance at the lowest epsilon value on Gemma 1B is relatively poor, we ran a budget-splitting ablation at ε=2 (Gemma 1B, r=128, MIMIC):
>
> |(ε_train,ε_sel)|Marg.|Trans.|TDCR|
> |:--|--:|--:|--:|
> |(.5,1.5)|2.92|1.13|0.55|
> |(1,1)|2.85|1.12|0.52|
> |*(1.5,.5)*|*2.93*|*0.43*|*0.36*|
> |(2,-)|2.85|1.13|0.50|
>
> Marginals are stable across splits, but temporal metrics differ substantially: our default (1.5, 0.5) achieves Trans.=0.43 and TDCR=0.36 vs ~1.13 and ~0.50 for all other splits. The model needs sufficient training budget to learn temporal dynamics, after which even ε_sel=0.5 provides effective filtering.
>
> > **Limitations.**
>
> Agreed! We added such a limitation section in revision. See reviewer 5GAy response for details.

---

> > ### Author Rebuttal · Reviewer_h9Ag · 2026-04-03
> >
> > Thank you for the authors’ response. My concerns have been fully addressed. I will maintain my score of 5, as it already reflects a positive evaluation.

---

### Official Review · Reviewer_5GAy · 2026-03-10

**Soundness:** 3
**Presentation:** 4
**Significance:** 3
**Originality:** 3
**Overall Recommendation:** 4
**Confidence:** 2

**Summary:**

This paper studies differentially private synthesis of longitudinal tabular data, where each user is represented by a trajectory (multiple rows) rather than a single row. The authors argue that standard DP tabular methods fail in this setting because they flatten trajectories into wide vectors, which introduces padding and harms temporal coherence. They propose PATH, a method that fine-tunes an LLM to generate whole user trajectories directly, and show that it preserves temporal structure better than flattened marginal baselines on medical, civic, and synthetic datasets.

**Compliance With Llm Reviewing Policy:**

Affirmed.

**Final Justification:**

All points addressed, I keep my positive score (but note that I am not knowledgable about the privacy area enough to have a high confidence).

**Key Questions For Authors:**

1) Can you  include a non-private PATH baseline? This would help separate the gain from the trajectory-level autoregressive formulation from the loss due to DP-SGD + private selection, and would make the privacy-utility tradeoff much clearer.

2) Can you compare PATH to at least one prior sequentia DP synthesizer, rather than mainly flattened marginal baselines?  As written, the experiments mostly show that PATH beats flattening, but it is less clear how it compares to earlier transformer-style or sequential DP approaches discussed in related work.

3) How much of PATH’s improvement comes from the model itself vs the rest of the pipeline? Because PATH also relies on parsing/repair, over-generation, and private post-hoc selection, so checking the importance of these components would be useful.

**Limitations:**

The Discussion section does not list possible limitations. Those could include in my view the privacy risks despite DP, and mentioning that PATH’s advantage is strongest for complex temporal structure, not for all tabular synthesis problems (on the controlled HMM dataset, marginal methods outperform PATH on some distributional metrics).

**Strengths And Weaknesses:**

Soundness: The idea in the paper is that for longitudinal tabular data, the natural synthesis unit is the full user trajectory rather than an i.i.d. row, and PATH is designed around that formulation in the DP setting. The empirical setup is good, and it also includes (in addition to real world datasets) a synthetic HMM benchmark, which directly tests temporal coherence rather than only marginals. My main concern is that the strongest comparisons are mostly against flattened marginal baselines, and on NYC 311 those baselines are dropped because they do not scale. The evidence right now is stronger to show that PATH beats flattening + DP synthesis rather than showing that PATH is best among sequential DP generators.

Presentation: The paper is clear and easy to follow. The motivation for why flattening fails is well explained, and the row-by-row autoregressive setup is intuitive.

Significance: I think the problem is important, since many sensitive datasets are naturally longitudinal and existing DP tabular methods are not well adapted to them. The results on MIMIC and the HMM benchmark show that preserving marginals is not enough if temporal dynamics matter. Note though that I am not super familiar with the DP litterature.

Originality: The main novelty feels to me more conceptual and empirical, rather than algorithmic. The method itself is fairly standard so I see the contribution primarily as a strong new formulation plus a convincing empirical demonstration, rather than a major technical advance in DP learning methodology.

---

> ### Author Rebuttal · Authors · 2026-03-31
>
> Thank you for the thoughtful review. We appreciate you highlighting some strengths of our work, including the importance of the problem for naturally longitudinal sensitive datasets (S1), the inclusion of the HMM benchmark for directly testing temporal coherence (S2), and the clarity of the presentation and motivation (S3).
>
> > **Q1: Non-private PATH baseline.**
>
> Thank you for this suggestion, we agree and have added this experiment to the papers results and analysis. Here we present results on PATH with Gemma 3 1B on the MIMIC dataset *without DP-SGD* (standard fine-tuning, no noise, no clipping) and across all LoRA ranks. Below, "Marg." is average per-feature Wasserstein distance (lower=better) and "Trans." is average Frobenius norm between binned state transition matrices (lower=better). Note DP runs use $\varepsilon$=1.5 for training + $\varepsilon$=0.5 for private selection.
>
> |$r$|Non-DP Marg.|Non-DP Trans.|DP Marg.|DP Trans.|
> |--:|--:|--:|--:|--:|
> |256|3.96|0.40|2.82|0.41|
> |128|3.74|0.41|2.93|0.43|
> |64|3.62|0.40|3.78|0.41|
> |32|3.79|0.42|3.89|0.42|
>
> Interestingly, DP-trained PATH with private selection matches or outperforms non-DP without selection on marginal fidelity, demonstrating that the selection step somewhat compensates for DP noise. Transition divergence is comparable across DP and non-DP, suggesting DP noise primarily affects marginal accuracy rather than temporal structure.
>
> > **Q2: Comparison with sequential DP synthesizers.**
>
> Thank you for the push to compare with another baseline approach! We have compared PATH against a DP version of DoppelGANger (Lin et al., 2020), a GAN-based sequential synthesizer, which we discussed in the related works. We carefully implemented a DP version using their public repository with DP-MERF considerations and tuned hyperparameters to give it the best chance. Still, in our experiments, PATH outperforms DoppelGANger substantially (22x on marginal fidelity (Marg. 2.85 vs 64.17), 3x on transition divergence (Trans. 0.38 vs 1.20), etc.). See our response to Reviewer 3JBr for the full results table and discussion.
>
> > **Q3: Component-wise ablation.**
>
> Thank you for the push to more carefully ablate the PATH approach. In particular, on top of comparing to non-private PATH without a selection step, vs. DP PATH w/ private selection (the results presented above), we also checked the **budget splitting allocation of the private selection step.**
>
> We **checked three budget allocations** at total $\varepsilon$=2: all budget to training, $\varepsilon$=1.5+0.5, and $\varepsilon$=1.0+1.0 (see Reviewer h9Ag Q4 for the full table). DP+selection outperformed non-DP without selection (Marg. 2.93 vs 3.74 at $r$=128); this suggests that the selection step acts as an effective quality filter over the over-generated pool, and is important to the PATH pipeline's success.
>
> We also checked **LoRA rank / model capacity**, as you saw above, and found that higher LoRa rank $r$ improved metrics under DP ($r$=256: 2.82 vs $r$=32: 3.89). Under non-DP, rank mattered less, suggesting DP-SGD noise benefits from higher model capacity, to a point.
>
> We also checked Llama 3.2 1B and Qwen 3.5 0.8B. Gemma was strongest overall (e.g., $r$=256: Marg. 2.82 vs Llama 3.73, Qwen 3.20), but PATH worked well across all three model families.
>
> These results, in total, suggest that the autoregressive LLM and generation procedure is the core contributor to PATHs success; the other pipeline components serve to marginally increase performance, especially on certain metrics like marginal fidelity. And, as is generally true with LLMs for structured tabular data generation, parsing based on predetermined rules is an important part of our pipeline.
>
> > **Limitations section.**
>
> We have added a limitations section to the revision, apologies for its omission in the original submission. In it, we discuss (1) PATH's advantage as strongest for complex temporal structure and may not benefit simple tabular synthesis where marginal methods suffice; (2) computational cost: PATH is more computationally expensive than marginal methods and requires a GPU; (3) additional privacy risks in any data release containing e.g. sensitive medical information.
>
> > **Originality.**
>
> We appreciate your characterization and largely agree. Our primary contributions are formulating the table-as-privacy-unit for longitudinal data problem, proposing a natural LLM framework w/DP-SGD and modern over-generation and selection techniques for producing effective, temporal-trend preserving synthetic data under that formulation, paired with a rigorous empirical demonstration and consideration of which temporal metrics are appropriate. We believe identifying the right problem formulation and proving a methods effectiveness under that formulation is valuable for the community, even though many of the individual building blocks here (DP-SGD, LoRA, autoregressive generation, private selection) are established.

---

> > ### Author Rebuttal · Reviewer_5GAy · 2026-04-03
> >
> > Thank you for your response, I keep my positive evaluation.

---

### Official Review · Reviewer_1NTX · 2026-03-13

**Soundness:** 3
**Presentation:** 3
**Significance:** 3
**Originality:** 2
**Overall Recommendation:** 4
**Confidence:** 4

**Summary:**

This paper studies differentially private tabular data synthesis. Prior work typically assumes that each row corresponds to a person and that rows are IID. This paper considers a more practical case where each user owns an entire table representing a temporal trajectory. They propose PATH, a framework that treats the full table as the unit of synthesis and leverages the autoregressive capabilities of privately fine-tuned LLMs to generate temporally coherent synthetic data.

**Compliance With Llm Reviewing Policy:**

Affirmed.

**Final Justification:**

Most of my concerns were addressed.

**Key Questions For Authors:**

See Weaknesses.

**Limitations:**

See Weaknesses.

**Strengths And Weaknesses:**

**Strengths:**

- This paper studies a practical problem where a full user table is treated as the privacy unit, rather than assuming each row corresponds to an independent individual. The problem formulation is also clear.

- The method is reasonable, as autoregressive LLMs naturally model sequential trajectories.

**Weaknesses:**

- The non-private baseline is not very strong. The paper compares against Gemini with few-shot prompting. A stronger comparison would include non-private fine-tuning of the LLM.

- The evaluation mainly focuses on distributional metrics. It would be helpful to include downstream tasks (e.g., training predictive models on synthetic data) to provide more evidence of practical utility.

- The paper lacks ablations on several design choices, including the private selection step, the dynamic windowing strategy, the context length, and the choice of LLM.

- The paper does not include empirical privacy attack evaluations, such as membership inference.

- It would be helpful to evaluate more privacy budgets to better illustrate the privacy–utility trade-off.

---

> ### Author Rebuttal · Authors · 2026-03-31
>
> Thank you so much for the constructive feedback! We appreciate that you recognized the practical relevance of the trajectory-level privacy formulation, the clarity of the problem setup, and the natural fit of autoregressive LLMs for sequential trajectory modeling. We address each question/weakness below.
>
> > **W1: baseline running PATH non-privately.**
>
> We agree and have run non-private fine-tuning (standard training, no DP-SGD) as the upper-bound baseline. The non-DP Gemma 1B finetuned model on MIMIC achieved lower loss than any of the DP versions, as expected; these plots will be added to the final revision. As for downstream temporal metrics, the non-DP Gemma 1B achieved the lowest "Trans.=0.41" (average Frobenius norm between binned state transition matrices (lower=better)) (see Reviewer 5GAy Q1 for the full Gemma DP vs non-DP table across all LoRA ranks). Interestingly, DP-trained PATH *with* private selection matched or outperformed non-DP Gemma (without selection) on the univariate metric (Marg.=3.74 vs Marg.=2.93), notably without a selection step, suggesting that the selection step compensates for DP noise in matching these univariate distributions better.
>
> > **W2: include downstream tasks.**
>
> We have run TSTR evaluations on both datasets to confirm that PATH's distributional fidelity translates to downstream utility, thank you for the suggestion! **See Reviewer h9Ag for full tables and discussion.** In summary, for **NYC 311**, we ran "next-complaint-type" prediction w/ 10 most common complaint types. PATH achieves **53.8% accuracy** (vs. 32.1% majority class baseline and 10% for random guessing), close to the real-data accuracy (58.5%). We featurized the 311 data (temporally focused measures, like lag-3 complaint types, gap statistics, time-of-day, etc.).
>
> For **MIMIC**, we ran emergency department (ED) stay-length binary classification (>= 8 hours). PATH shows positive lift above majority (AUC 0.57), while AIM did not show any (AUC 0.47). Binary ED stay length prediction is a known hard task; prior work with richer features than vital signs alone reports AUC of 0.65-0.75 [Ala et al., Frontiers in AI, 2023].
>
> > **W3: ablations on design choices.**
>
> We have completed several computationally intensive ablations, thank you for encouraging us to expand our understanding of PATH. We have added each of these to the revised paper.
>
> - **LoRA rank** ($r \in \{32, 64, 128, 256\}$). Higher rank improves marginal fidelity under DP; performance is robust for $r \ge 64$. See the cross-model table below and Reviewer 5GAy Q1 for the full Gemma ablation.
>
> - **Private selection / budget splitting**. We ablated three budget allocations at total privacy budget of $\varepsilon$=2: allocating the total to training vs. splitting between training and private selection. Results were stark: DP+selection outperformed non-DP without selection, and the (1.5, 0.5) split from the paper led to the best DP results on temporal metrics. *See Reviewer h9Ag (Q4) for the full budget-splitting table.*
>
> - **Choice of LLM**. We checked two additional models: Llama 3.2 1B and Qwen 3.5 0.8B. DP Marg. (lower=better) at two representative ranks:
>
> |Model|$r$=128|$r$=256|
> |:--|--:|--:|
> |Gemma 1B|2.93|**2.82**|
> |Llama 1B|3.61|3.73|
> |Qwen 0.8B|**2.70**|3.20|
>
> Gemma is most consistent; Qwen achieves the best single result at $r$=128. PATH works across all three families, confirming it is model-agnostic.
>
> - **Computational cost**: The major tradeoff for PATH is the computational overhead; PATH Gemma 1B LoRa training takes ~62 min vs ~6 min for AIM and ~10 min for DoppelGANger. Generation takes seconds per table (but is highly parallelizable).
>
> > **W4: empirical privacy attack evaluations.**
>
> PATH satisfies formal $(\varepsilon, \delta)$-DP by construction (via DP-SGD and the Gaussian mechanism during private selection), providing a worst-case guarantee independent of any specific attack. Prior work has shown that membership inference attacks are ineffective against DP-trained models so long as the DP implementation is correct (Stadler et al. 2022). We note that several influential DP synthetic data works (Liu et al. 2023; McKenna et al. 2022; Liu et al. 2021) rely principally on the formal guarantee without reporting adversarial metrics.
>
> > **W5: more privacy budgets.**
>
> Our evaluation covers $\varepsilon \in \{0.5, 2.0, 4.0, 10.0\}$, spanning a 20x range in privacy values, which is standard in the DP synthetic data literature (see, for example, Lin et al. 2025, 2502.05505, Xie et al. 2024, 2403.01749, Liu et al. 2021, 2106.07153, etc.).

---

> > ### Author Rebuttal · Reviewer_1NTX · 2026-04-02
> >
> > Thank you for the rebuttal. It addresses most of my concerns. I note that the dynamic windowing and context length ablations were not addressed, and I encourage the authors to include these in the final revision. The formal DP justification for skipping empirical attacks is reasonable, but a brief MIA evaluation would further strengthen the paper. I maintain my score of 4 (Weak Accept).

---

> > > ### Author Response · Authors · 2026-04-03
> > >
> > > Thank you for your continued engagement with our paper during this rebuttal period! We have begun running the additional ablations you suggested. Below, we report context length results on MIMIC for Gemma 1B at $\epsilon = 10$:
> > >
> > > | MaxLength | Train Loss | Eval Loss | Acc. |
> > > |-----------|------------|-----------|------|
> > > | 128       | .436       | .434      | 83.0 |
> > > | 256       | .316       | .323      | 87.7 |
> > > | 512       | .307       | .312      | 88.0 |
> > >
> > > There is a marked improvement between 128 and 256 max context length, while the gain from 256 to 512 is comparatively modest. This aligns with the long-tailed distribution of trajectory lengths in MIMIC: many of the examples fit within 256 tokens, so the largest gains come from accommodating that bulk, whereas extending to 512 better captures remaining long-tail trajectories but did not seem to improve average performance significantly.
> > >
> > > Regarding the dynamic windowing ablation, we want to ensure we understand your request correctly and run the right comparison. In our current approach, we sample a split point $k \in [0, T_i - 1]$ at each epoch, always conditioning on the prefix $\mathbf{x}_{1:k}$. There are (at least) two natural ablations we can run that would help improve our understanding of the dynamic windowing strategy: (1) fixing $k$ rather than resampling it each epoch, or (2) adopting a sliding window $[j, j+W]$ that allows the model to condition on later portions of long trajectories rather than always starting from position 1. Could you clarify which variant you had in mind? We currently have the sliding window experiment running and will report results when it finishes. We are also happy to run the fixed-$k$ comparison if that is closer to what you envisioned. Thank you again for the thoughtful suggestions, these ablations are genuinely strengthening the paper, and we also hope they are improving your understanding of our work.

---

### Official Review · Reviewer_3JBr · 2026-03-13

**Soundness:** 3
**Presentation:** 3
**Significance:** 3
**Originality:** 3
**Overall Recommendation:** 4
**Confidence:** 3

**Summary:**

This paper proposes PATH, a novel generative framework for differentially private synthetic longitudinal data. To effectively preserve temporal dynamics, the framework treats the full user table as the minimal unit of privacy and leverages the autoregressive sequence-generation capabilities of privately fine-tuned large language models.Experiments on the MIMIC-IV dataset, NYC 311 service requests, and synthetic HMM data demonstrate that the proposed framework significantly outperforms traditional marginal-based baselines in maintaining temporal coherence and distributional fidelity.

**Compliance With Llm Reviewing Policy:**

Affirmed.

**Final Justification:**

Overall， the rebuttal has resolved my core concerns regarding baselines, computational costs, and practical boundaries.Thus, I have increased the score accordingly.

**Key Questions For Authors:**

1. Can the authors provide comparisons with specialized differentially private sequential generation models (e.g., DP-DoppelGANger or NetShare) instead of only static marginal baselines?
2. Can the authors include a "performance vs. compute" trade-off analysis (e.g., GPU memory, training hours) comparing the PATH framework against traditional marginal algorithms?
3. Given that AIM outperforms PATH on the simpler Synthetic dataset (Table 2), what data characteristics should guide a user to choose PATH over lighter marginal methods?
4. The Classifier AUC scores approach 1.0 (Tables 3 and 12), indicating easily identifiable artificial artifacts. How might these micro-level artifacts impact the utility of the synthetic data in downstream machine learning tasks?

**Limitations:**

The paper omits a discussion of the computational costs and the practical boundary of applicability.

**Strengths And Weaknesses:**

Strengths:
1. The research question is critical. Extending differentially private tabular data synthesis to longitudinal trajectories by treating the entire user table as the privacy unit addresses a significant gap in the literature.
2. The paper is well-structured and the framework is clearly presented. The methodology of serializing temporal tables for autoregressive LLM fine-tuning elegantly avoids the fundamental pitfalls of the flattening transformation.
3. The paper provides a rigorous evaluation using a well-designed mix of real-world (MIMIC-IV, NYC 311) and synthetic (HMM) datasets, accompanied by an exceptionally in-depth theoretical analysis of gradient dynamics and privacy regularization in the appendix.

Weaknesses:
1. Important baselines are missing. The framework is compared against static marginal-based methodss,like AIM and DIRECT,applied to flattened data, lacking direct comparisons with specialized differentially private time-series generation models such as DP-DoppelGANger or NetShare.
2. The paper completely omits an analysis of computational costs. Fine-tuning a 4-billion-parameter LLM with DP-SGD requires drastically more memory and compute time than traditional marginal-based algorithms, and this performance-efficiency trade-off should be explicitly discussed.
3. The paper lacks practical guildlines for method selection. While the author transparently acknowledge that traditional marginal baselines outperform the proposed framework on the simpler Synthetic dataset(Table 2), the paper lacks a clear discussion on the practical boundaries of the PATH framework, especially given the immense computational cost of fine-tuning LLMs.
4. The synthetic data remains highly distinguishable from real data(Table 3 and 12). The Classifier Discriminator metrics show AUC scores near 1.0 across almost all methods, indicating that the generated trajectories still contain easily identifiable features.

---

> ### Author Rebuttal · Authors · 2026-03-31
>
> Thank you for your time, and for recognizing the research question's importance (S1), the framework's clarity (S2), and the rigorous evaluation and gradient dynamics analysis (S3). We believe these clarifications and new experiments substantially strengthen the paper.
>
> > **W1/Q1: Missing comparison with sequential DP synthesizers (DP-DoppelGANger).**
>
> We agree that it's important to compare against relevant prior work. Part of the reason we did not include a comparison to GAN based methods is that, at this point, prior benchmarking work has shown consistently that DP GANs struggle on structured tabular data (Chen et al. 2025; Tao et al. 2022), and e.g. DoppelGANger's (Lin et al., 2020) original DP version was limited (unmodified DP-SGD on their GAN objective).
>
> Still, we have now evaluated a DP version of DoppelGANger on the MIMIC dataset, and **in fact have adjusted the original implementation from the `ydata-synthetic` package to be stronger** by incorporating DP-MERF (Harder et al. 2020) (kernel mean matching via random Fourier features + Gaussian mechanism). Additionally, we tuned hyperparameters non-privately, to give DoppelGANger the best chance to compete with PATH and AIM. "Marg." = avg. Wasserstein distance, "Trans." = avg. Frobenius norm between transition matrices, "TDCR" (see paper) (all lower=better). Results at $\varepsilon$=2 vs PATH Gemma 1B:
>
> |Metric|PATH|DG|
> |:--|--:|--:|
> |Marg. (avg)|2.85|64.17|
> |Trans. (avg)|0.38|1.20|
> |TDCR|0.45|0.83|
>
> **PATH substantially outperforms DoppelGANger** (~22x on Marg., ~3x on Trans., ~2x on TDCR). As prior literature has observed, the GAN-based training is sensitive to high-dimensional, mixed-type tabular data under DP constraints; several MIMIC vitalsign columns showed limited diversity (mode collapse) despite our best efforts. We have added these comparisons to the revised paper, and thank the reviewer for the suggestion.
>
> > **W2/Q2: Computational cost analysis.**
>
> We apologize that this was not reported prominently in the original draft! You are correct, this is the main tradeoff of the method between marginal based methods and PATH: LLMs require more compute / training time. We benchmarked all methods on the MIMIC data (single NVIDIA RTX PRO 6000):
>
> |Method|Train|Generation|
> |:--|--:|--:|
> |DoppelGANger|~10 min|~ms/table|
> |AIM/Direct|~6 min|~ms/table|
> |PATH (Gemma 1B)|~62 min|~seconds/table|
>
> On our compute, PATH training was about 10x slower than marginal methods, which is expected given LLM fine-tuning. PATH generation is also two orders of magnitude slower than marginal based methods per table, but is parallelizable across GPUs (allowing us to generate e.g. 1k tables in a few minutes). We'd like to emphasize that (1) training is a one-time cost; (2) the 1B model substantially outperforms baselines; (3) we'd argue that ~1 hr on a single GPU is modest for the quality gains. We have added this discussion to the revised paper.
>
> > **W3/Q3: Practical guidelines for method selection.**
>
> Thank you for pushing us to clarify the Synthetic dataset results (Table 2). Our experiments showed that AIM outperformed PATH on *marginal divergence*, which is expected: AIM directly measures and noises marginals, so it is optimized for exactly this type of metric, and the Synthetic medical dataset is very well behaved. However, PATH outperforms AIM on the *temporal* metrics (HMM log-likelihood, state transition divergence, TDCR) even on this simpler dataset. Which method is "better" depends on what the user cares about. We will add the following guidance to the revised paper, thank you for the push:
>
> We'd recommend to **use PATH when temporal dynamics matter (medical trajectories, event sequences, time series).** PATH captures inter-row dependencies in a way that, structurally, the marginal methods cannot, as demonstrated by its consistent advantage on transition and TDCR metrics across all three datasets. Conversely, **AIM should be used when the user primarily needs accurate marginal distributions and temporal ordering is not semantically meaningful, or when GPU resources are unavailable.**
>
> > **W4/Q4: Classifier AUC near 1.0.**
>
> We were very careful not to introduce micro-level artifacts during serialization: we conducted extensive inspections (both automated and human-in-the-loop) to ensure that formatting, decimal precision, rounding behavior, and serialization conventions are identical between real and synthetic tables, so the classifier cannot distinguish them based on surface-level formatting. Thus, we are confident that the high AUC reflects distributional differences in the data content introduced by DP noise, not formatting artifacts.
>
> Of course, a strong classifier could still detect recurring tropes (repeated values, mode collapse, unrealistic ranges), which is exactly the utility of a classifier discriminator metric. To make this transparent, **we will add random example tables (real and synthetic) to the appendix.** Thank you for the push to clarify.

---

> > ### Author Rebuttal · Reviewer_3JBr · 2026-04-04
> >
> > Thank you for the comprehensive rebuttal. The additional experiments and transparent discussions have adequately addressed my primary concerns. Thus, I will update the score.

---

### Decision · Program_Chairs · 2026-04-30

**Decision:**

Accept (regular)

**Comment:**

This paper proposes PATH, a framework that defines the differential privacy unit at the user-table level and leverages privately fine-tuned LLMs to autoregressively generate temporally coherent longitudinal data. All reviewers agree that the paper addresses an important and underexplored problem in privacy-preserving data generation. The “table as privacy unit” formulation is conceptually clear and theoretically supported, and the use of autoregressive LLMs is well-motivated for modeling temporal dependencies. The empirical evaluation is comprehensive, covering multiple datasets, privacy budgets, and metrics (including TDCR). The finding that private fine-tuning outperforms non-private few-shot prompting is also notable.

During the initial review phase, concerns included the absence of sequential DP baselines (3JBr, 5GAy), lack of downstream evaluation (h9Ag, 1NTX), missing computational cost analysis (3JBr), limited ablations (1NTX, 5GAy, h9Ag), and unusually high classifier AUC values (3JBr, h9Ag). These were largely addressed through the addition of DP-DoppelGANger, TSTR evaluations, computational benchmarks, and expanded ablations across LoRA ranks, model families, and privacy budget allocations. Three of the four reviewers confirmed that their concerns were resolved, while the remaining reviewer (1NTX) maintained a Weak Accept with minor suggestions (e.g., dynamic windowing). The authors also committed to adding error bars over five random seeds in the final version.
Overall, the discussion converged positively, and no major concerns remain. While the contribution is primarily conceptual and empirical rather than algorithmically novel, it is well-aligned with the problem setting and supported by strong experimental evidence. I recommend acceptance, with the expectation that the authors incorporate the promised revisions.